# DGSolver: Diffusion Generalist Solver with Universal Posterior Sampling for Image Restoration

**Hebaixu Wang**[1,3]**, Jing Zhang**[2†]**, Haonan Guo**[3,4]**, Di Wang**[2,3]**, Jiayi Ma**[1,3†]**, and Bo Du**[2,3†]

[1]School of Electronic Information, Wuhan University, Wuhan, China
[2]School of Computer Science, Wuhan University, Wuhan, China
[3]Zhongguancun Academy, Beijing, China
[4]State Key Laboratory of Information Engineering in Surveying, Mapping and Remote Sensing, Wuhan University, Wuhan, China
{wanghebaixu,jingzhang.cv,jyma2010}@gmail.com;{haonan.guo,d_wang,dubo}@whu.edu.cn

## Abstract

Diffusion models have achieved remarkable progress in universal image restoration. However, existing methods perform naive inference in the reverse process, which leads to cumulative errors under limited sampling steps and large step intervals. Moreover, they struggle to balance the commonality of degradation representations with restoration quality, often depending on complex compensation mechanisms that enhance fidelity at the expense of efficiency. To address these challenges, we introduce **DGSolver**, a diffusion generalist solver with universal posterior sampling. We first derive the exact ordinary differential equations for generalist diffusion models to unify degradation representations and design tailored high-order solvers with a queue-based accelerated sampling strategy to improve both accuracy and efficiency. We then integrate universal posterior sampling to better approximate manifold-constrained gradients, yielding a more accurate noise estimation and correcting errors in inverse inference. Extensive experiments demonstrate that DGSolver outperforms state-of-the-art methods in restoration accuracy, stability, and scalability, both qualitatively and quantitatively. Code and models are publicly available at https://github.com/MiliLab/DGSolver.

## 1 Introduction

Universal image restoration (UIR) strives to achieve a versatile model capable of discerning and mitigating various degradation types. Capitalizing on the superior attributes of the restored details with high fidelity, it has emerged as a promising approach for enabling applications such as autonomous driving [37], remote sensing [44], and other related fields [73, 26, 58, 59, 71, 68, 54, 55, 56].

Owing to the substantial advancements in deep learning, a surge of universal image restoration frameworks has emerged. Typically, the existing methods can be delineated into two categories based on the dependency of representations across varied degradation distributions [64, 74, 9], namely blend-representation-based methods and distinct-representation-based methods, as displayed in Fig. 1. The former approaches promote the commonality among degradation representations by projecting diverse degradation distributions into a shared representation space, where the commonality measures the degradation-agnostic extent of representations within the shared representation space [34]. However, the complexity of restoration escalates as the commonality of degradation representations intensifies [10, 28, 29]. DiffUIR [75] reaches a compromise by projecting the disparate distributions onto an impure Gaussian distribution, retaining partial degradation conditions. FoundIR [23] further

---

[†]Corresponding author.

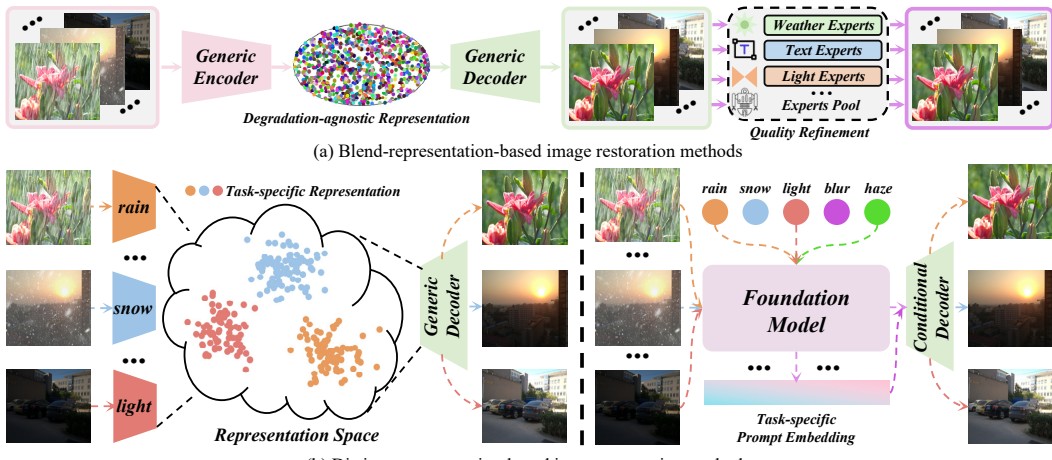

(a) Blend-representation-based image restoration methods

(b) Distinct-representation-based image restoration methods

Figure 1: Overview of mainstream UIR methods. (a) These methods typically train generic models to learn a shared distribution with compensation mechanisms for quality refinement. (b) Conversely, distinct-representation-based methods integrate task-specific modules into generic models to handle diverse degradations, forcing the generic model to learn different distributions.

implements a mixture-of-experts system to mitigate the diminished performance resulting from the pursuit of commonality. Consequently, the compensation systems that are incurred significantly reduce efficiency. The latter approaches utilize multi-encoder architectures or prompts from foundation models to acquire distinct representations tailored for specific degradation distributions [27, 42, 13]. Afterwards, customized conditional guidance is incorporated into a shared decoder to reconstruct the distorted information [7, 11]. Nevertheless, their performance is constrained by the limited exploration of degradation correlations, as different tasks may potentially share complementary information that could augment the efficacy of individual tasks [25]. For instance, low light conditions and noise frequently coexist, while rain and raindrops are often interlinked.

In this paper, we aim to develop a precise and efficient method for universal image restoration. It is observed that the existing methods are plagued by two major drawbacks: One is the dilemma between the commonality of degradation representations and restoration quality, and the other is the lack of a versatile and effective compensatory mechanism for quality refinement. To this end, we propose **DGSolver**, a diffusion generalist solver with universal posterior sampling for image restoration, as presented in Fig. 2. It amalgamates the merits of ordinary differential equation (ODEs) solvers for high-precision inference and leverages diffusion posterior sampling as universal accuracy compensation for refinement under high commonality conditions. Specifically, we adopt the diffusion generalist models (DGMs) [75, 29] to unify diverse degradation types into a purely degradation-agnostic latent distribution. Subsequently, we tackle the alternative sampling from DGMs by solving the corresponding semi-linear ODEs to alleviate the discretization error accumulated in the multi-step inference process. Benefiting from the analytical structure, the solutions can be equivalently simplified to a linearly weighted integral of the neural networks with an accelerated sampling strategy. Furthermore, we seamlessly integrate the underlying mechanism of generalist modeling with Bayesian posterior sampling, thereby universally serving as manifold-constrained gradients for restoration guidance without incurring computational expenditures. Consequently, high-order ODEs numerical approximation and universal posterior sampling are combined to effectively improve the solution accuracy and stability in a training-free manner. Extensive experiments on natural and remote sensing scenes are conducted to verify the superior performance of our method, showcasing its remarkable restoration capability across different application scenes. Our key contributions are as follows:

1. We reformulate the generalist diffusion process using ODEs and derive its analytical solution form. Besides, high-order solvers equipped with an accelerated sampling strategy are customized to mitigate accumulated discretization errors and boost efficiency during inverse inference.
2. To further enhance the accuracy and stability of restoration, we introduce a universal posterior sampling strategy to offer manifold-constrained gradient guidance for noise estimation. It is seamlessly integrated into generalist diffusion solvers requiring no training.

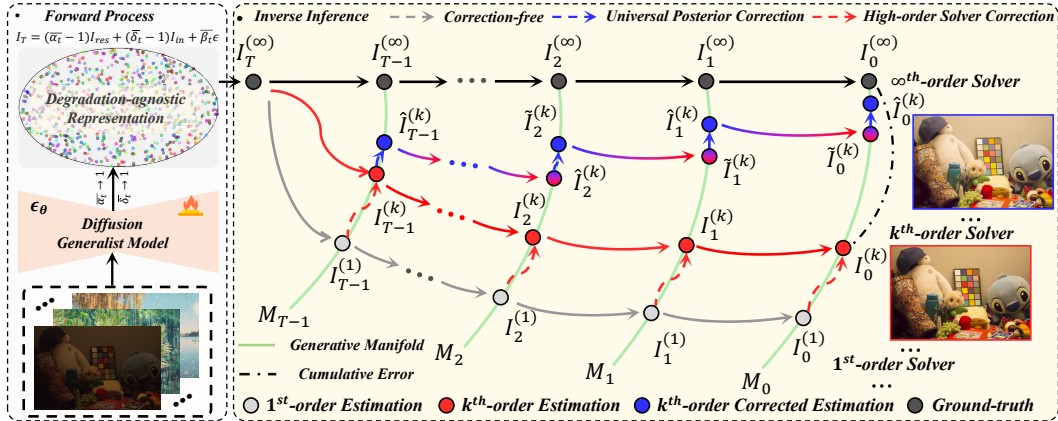

Figure 2: Illustration of the DGSolver. In the forward process, we utilize a diffusion generalist model to uniformly represent diverse degradation categories into degradation-agnostic distributions. In the inverse inference, we employ a $k^{th}$-order solver to alleviate the discretization error accumulated in the multi-step sampling process. Simultaneously, universal posterior sampling is used to stabilize the solution and further minimize the discrepancies with the ideal one generated by a $\infty^{th}$-order solver.

## 2 Background

### 2.1 Diffusion-based Image Restoration Model

Grounded in solid mathematical foundations, diffusion models [14, 50, 51] have emerged as powerful frameworks for image restoration (see Appendix A). Initially, lots of works focus on structural improvements to facilitate the restoration of details [62, 40, 49] by simply treating degraded images as conditional inputs of the denoising network. Afterwards, InDI [10] and I²SB [28] pioneer profound theoretical and practical improvements in the reverse process, opting for estimating the target image or its linear transformation term to replace the noise estimation. RDDM [29] extends beyond the single diffusion paradigm to independent double diffusion processes corresponding to residual and noise estimations, respectively. Consequently, the residual prioritizes certainty to restore the major structures while the noise emphasizes diversity to retrieve the dismissed details. Furthermore, DiffUIR [75] incorporates partially degraded information into the forward process and constructs a non-pure Gaussian distribution representation of different degradation types. Nevertheless, the commonality of degradation representations and the performance of restoration are mutually constrained, with no versatile and effective approach available to simultaneously enhance both.

### 2.2 Diffusion Posterior Sampling

To tackle the ill-posed attributes of linear inverse problem [19], diffusion posterior sampling has been developed as a novel approximation tool to reduce the solution uncertainty [11, 45]. Specifically, it leverages the Bayesian framework to accurately estimate the clear image $I_0$ from its degraded observation $I_{in} = \mathcal{A}I_0 + n$, where $\mathcal{A}$ is a measurement operator and $n$ is usually the Gaussian noise. The stochastic differential equations (SDEs) of diffusion probabilistic models [51] with diffusion term $f(I_t, t)$ and drift term $g(t)$ can be formulated as:

$$dI_t = f(I_t, t)dt + g(t)d\omega_t, \tag{1}$$

where $\omega_t$ is the standard Wiener process. The reverse SDE of this process can be:

$$dI_t = [f(I_t, t) - g^2(t)\nabla_{I_t} \log q_t(I_t)]dt + g(t)d\overline{\omega}_t, \tag{2}$$

where $\nabla_{I_t} \log q_t(I_t)$ defines the predicted score functions. In the case of inverse problems, we want to generate the posterior distribution $q_t(I_t | I_{in}, \mathcal{A})$. Leveraging Bayesian rule, Eq. (2) becomes:

$$dI_t = [f(I_t, t) - g^2(t)(\nabla_{I_t} \log q_t(I_t) + \nabla_{I_t} \log q(I_{in} | I_t))]dt + g(t)d\overline{\omega}_t. \tag{3}$$

Nevertheless, $\nabla_{I_t} \log q(I_{in} | I_t))$ can be computationally intractable. Chung *et al.* [7] first approximate gradient of the log likelihood by assuming $p(I_{in} | I_t) \approx q(I_{in} | \hat{I}_0 := \mathbb{E}[I_0 | I_t]) = \mathcal{N}(I_{in}; \mu =$

$\mathcal{A}\mathbb{E}[I_0|I_t], \Sigma = \sigma_{I_{in}}^2 \boldsymbol{I})$. Essentially, they substitute the unknown clean image $I_0$ with its conditional expectation $\tilde{E}[I_0|I_t]$ and known operator $\mathcal{A}$, making the term $p(I_{in}|I_t)$ tractable. Afterwards, PSLD [48] extends pixel-level posterior sampling into latent space, showing provable sample recovery in a low-dimensional subspace. Furthermore, BlindDPS [6] introduces multiple diffusion models to construct a parallel reverse procedure to jointly estimate the operator kernel and clear image. In summary, existing posterior sampling relies on known or estimated degradation operators. The former still suffers from ill-posedness in the practical scene due to the availability of measurement operators, while the latter only fits well in kernel-based degradation modeling with extensive computational overhead and cannot be generalized to more complex degradation scenarios.

## 3 Methodology

### 3.1 Diffusion Generalist Formulations

Assume that a clear image $I_0 \in \mathbb{R}^D$ is obtained by sampling from an unknown distribution $q_0(I_0)$. We define a forward process $\{I_t\}_{t \in [0,T]}$ starting with $I_0$, such that for any $t \in [0, T]$, the distribution of $I_t$ is conditioned on $I_0, I_{in}$, and $I_{res}$, which satisfies:

$$q_{0t}(I_t|I_0, I_{res}, I_{in}) = \mathcal{N}(I_t; I_0 + \overline{\alpha}(t)I_{res} - \overline{\delta}(t)I_{in}, \overline{\beta}^2(t)\boldsymbol{I}), \tag{4}$$

where $\overline{\alpha}(t), \overline{\delta}(t), \overline{\beta}^2(t) \in \mathbb{R}^+$ are differentiable functions of $t$ with bounded derivatives, and $I_{res}$ is the residual items equal to $I_{in} - I_0$. When $t \to T, \overline{\alpha}(T) \to 1, \overline{\delta}(T) \to 1$, the endpoint $I_t = \overline{\beta}(t)\boldsymbol{I}$ is merely represented by a pure Gaussian distribution with a $\overline{\beta}(t)$-modulated variance. Therefore, we refer to the *linear noise schedule* [14] to generate parameter series and modify the value of $\overline{\alpha}(T), \overline{\delta}(T)$ to 1. Consequently, these variables are monotonic, and we denote them as $\overline{\alpha}_t, \overline{\delta}_t, \overline{\beta}_t$ for simplicity. We prove that the following stochastic differential equation (SDE) possesses an identical transition distribution $q_{0t}(I_t|I_0, I_{res}, I_{in})$ as in Eq. (4) for any $t \in [0, T]$ (proof in Appendix A.15):

$$dI_t = f(t)I_{res}dt + h(t)I_{in}dt + g(t)d\omega_t, \quad I_0 \sim q_0(I_0), \tag{5}$$

$$f(t) = \frac{d\overline{\alpha}_t}{dt}, \quad h(t) = -\frac{d\overline{\delta}_t}{dt}, \quad g^2(t) = \frac{d\overline{\beta}_t^2}{dt}. \tag{6}$$

To accelerate the sampling process, we consider the associated *probability flow ODE* with the same marginal distribution at each time $t$ as that of the SDE, which can be formulated as:

$$\frac{dI_t}{dt} = f(t)I_{res} + h(t)I_{in} - \frac{1}{2}g^2(t)\nabla_{I_t} \log q_t(I_t), \quad I_t \sim q_T(I_T). \tag{7}$$

Apparently, the *r.h.s* of Eq. (7) comprises two indeterminate variables, *i.e.* residual components $I_{res}$ and score functions $\nabla_{I_t} \log q_t(I_t)$, necessitating estimation via neural networks conditioned on the temporal variable $t$. Pursuant to the *Tweedie formula*, the score functions $\nabla_{I_t} \log q_t(I_t)$ can be equivalent to noise prediction, implying that $\epsilon = -\overline{\beta}_t \nabla_{I_t} \log q_t(I_t)$. Hence, the optimal sample can be obtained by solving the following ODE from $T$ to 0:

$$\frac{dI_t}{dt} = f(t)I_{res}^\theta(I_t, I_{in}, t) + h(t)I_{in} + \frac{g^2(t)}{2\overline{\beta}_t}\epsilon_\theta(I_t, I_{in}, t), \quad I_t \sim q_T(I_T), \tag{8}$$

where $I_{res}^\theta(I_t, I_{in}, t)$ and $\epsilon_\theta(I_t, I_{in}, t)$ are the residual estimator and noise predictor, respectively. In Eq. (8), $h(t)I_{in}$ is a linear function of $I_{in}$, while the other two parts $I_{res}^\theta(I_t, I_{in}, t)$ and $\frac{g^2(t)}{2\overline{\beta}_t}$ typically exhibit nonlinear dependencies on $I_t$ because of neural networks. *Previous works are ignorant of this semi-linear structure by decorating the entire Eq. (8) into a black-box ODE solver, which not only increases the complexity of the solution but also exacerbates the discretization errors of both linear and nonlinear terms.* Given an initial value $I_s$ at time $s > 0$, the solution $I_t$ at time $s < t < T$ can be precisely formulated as follows:

$$I_t = I_s + \int_s^t \frac{d\overline{\alpha}_\tau}{d\tau} I_{res}^\theta(I_\tau, I_{in}, \tau)d\tau - I_{in}\int_s^t \frac{d\overline{\delta}_\tau}{d\tau}d\tau + \int_s^t \frac{d\overline{\beta}_\tau}{d\tau}\epsilon_\theta(I_\tau, I_{in}, \tau)d\tau. \tag{9}$$

Such a taxonomy decouples the origin of discrete error, where the linear part can be accurately computed, but the integrals of the nonlinear parts are still complicated. As $\overline{\alpha}_t, \overline{\delta}_t, \overline{\beta}_t$ are

strictly monotonic function of $t$, there exist inverse functions $t_{\overline{\alpha}}(\cdot)$, $t_{\overline{\delta}}(\cdot)$, and $t_{\overline{\beta}}(\cdot)$ satisfying $t = t_\lambda(\lambda(t))$, $\lambda \in \{\overline{\alpha}, \overline{\delta}, \overline{\beta}\}$, respectively. We change the subscripts of $I_{res}^\theta$ and $\epsilon_\theta$ from $t$ to $\lambda$ and denote $I_{res}^\theta(I_\tau, I_{in}, \tau) = \widehat{I}_{res}^\theta(\widehat{I}_{\overline{\alpha}}, I_{in}, \overline{\alpha})$, $\epsilon_\theta(I_\tau, I_{in}, \tau) = \widehat{\epsilon}_\theta(\widehat{I}_{\overline{\beta}}, I_{in}, \overline{\beta})$. For each $\lambda$, Eq. (9) can be reformulated as:

**Proposition 1** *(Exact solution of ODEs for diffusion generalist models, proof in Appendix B). Given an initial value $I_s$ at time $s > 0$, the solution $I_t$ at time $s < t < T$ of that ODEs in Eq. (8) is*

$$I_t = I_s - (\overline{\delta}_t - \overline{\delta}_s)I_{in} + \int_{\overline{\alpha}_s}^{\overline{\alpha}_t} \widehat{I}_{res}^\theta(\widehat{I}_{\overline{\alpha}}, I_{in}, \overline{\alpha})d\overline{\alpha} + \int_{\overline{\beta}_s}^{\overline{\beta}_t} \widehat{\epsilon}_\theta(\widehat{I}_{\overline{\beta}}, I_{in}, \overline{\beta})d\overline{\beta}. \qquad (10)$$

In consequence, Eq. (10) provides a new perspective for approximating the optimal solution, where estimating the solution at time $t$ is equivalent to directly approximating the integral of $\widehat{I}_{res}^\theta(\widehat{I}_{\overline{\alpha}}, I_{in}, \overline{\alpha})$ from $\overline{\alpha}_s$ to $\overline{\alpha}_t$, and $\widehat{\epsilon}_\theta(\widehat{I}_{\overline{\beta}}, I_{in}, \overline{\beta})$ from $\overline{\beta}_s$ to $\overline{\beta}_t$. In light of such formulation, we are capable of customizing diffusion generalist solvers to improve the convergence and accuracy of solutions.

### 3.2 Diffusion Generalist Solvers

To enhance the precision in solving Eq. (8), we propose high-order solvers for the ODEs with a convergence order guarantee. Specifically, given an initial point $I_T$ at time $T$ and total $M + 1$ time steps $\{t_i\}_{i=0}^M$ decreasing from $t_0 = T$ to $t_M = 0$. Let $\{I_{t_i}\}_{i=0}^M$ be the sequence iteratively computed at time steps $\{t_i\}_{i=0}^M$ using the presented solvers. To mitigate the cumulative effect of significant errors, we endeavor to minimize the inaccuracy for the solution $I_{t_i}$ at each step. Starting with the previous solution $I_{t_{i-1}}$ at time $t_{i-1}$, the exact solution $I_{t_{i-1} \to t_i}$ at target time $t_i$ is:

$$I_{t_{i-1} \to t_i} = I_{t_{i-1}} - (\overline{\delta}_{t_i} - \overline{\delta}_{t_{i-1}})I_{in} + \int_{\overline{\alpha}_{t_{i-1}}}^{\overline{\alpha}_{t_i}} \widehat{I}_{res}^\theta(\widehat{I}_{\overline{\alpha}}, I_{in}, \overline{\alpha})d\overline{\alpha} + \int_{\overline{\beta}_{t_{i-1}}}^{\overline{\beta}_{t_i}} \widehat{\epsilon}_\theta(\widehat{I}_{\overline{\beta}}, I_{in}, \overline{\beta})d\overline{\beta}. \quad (11)$$

We utilize the Taylor expansion to make $k$ orders approximation to calculate the integral of both $\widehat{I}_{res}^\theta(\widehat{I}_{\overline{\alpha}}, I_{in}, \overline{\alpha})$ from $\overline{\alpha}_{t_{i-1}}$ to $\overline{\alpha}_{t_i}$ and $\widehat{\epsilon}_\theta(\widehat{I}_{\overline{\beta}}, I_{in}, \overline{\beta})$ from $\overline{\beta}_{t_{i-1}}$ to $\overline{\beta}_{t_i}$:

$$\widehat{I}_{res}^\theta(\widehat{I}_{\overline{\alpha}}, I_{in}, \overline{\alpha}) = \sum_{n=0}^{k-1} \frac{(\overline{\alpha} - \overline{\alpha}_{t_{i-1}})^n}{n!} \widehat{I}_{res}^\theta(\widehat{I}_{\overline{\alpha}_{t_{i-1}}}, I_{in}, \overline{\alpha}_{t_{i-1}}) + \mathcal{O}((\overline{\alpha} - \overline{\alpha}_{t_{i-1}})^k), \qquad (12)$$

$$\widehat{\epsilon}_\theta(\widehat{I}_{\overline{\beta}}, I_{in}, \overline{\beta}) = \sum_{n=0}^{k-1} \frac{(\overline{\beta} - \overline{\beta}_{t_{i-1}})^n}{n!} \widehat{\epsilon}_\theta(\widehat{I}_{\overline{\beta}_{t_{i-1}}}, I_{in}, \overline{\beta}_{t_{i-1}}) + \mathcal{O}((\overline{\beta} - \overline{\beta}_{t_{i-1}})^k). \qquad (13)$$

Substituting the above Taylor expansions into Eq. (11), yields:

$$I_{t_{i-1} \to t_i} = I_{t_{i-1}} - (\overline{\delta}_{t_i} - \overline{\delta}_{t_{i-1}})I_{in} + \sum_{n=0}^{k-1} \frac{(\overline{\alpha}_{t_i} - \overline{\alpha}_{t_{i-1}})^{n+1}}{(n+1)!} \widehat{I}_{res}^{\theta(n)}(\widehat{I}_{t_{i-1}}, I_{in}, t_{i-1})$$
$$+ \sum_{n=0}^{k-1} \frac{(\overline{\beta}_{t_i} - \overline{\beta}_{t_{i-1}})^{n+1}}{(n+1)!} \widehat{\epsilon}_\theta^{(n)}(\widehat{I}_{t_{i-1}}, I_{in}, t_{i-1}) + \mathcal{O}((\overline{\alpha} - \overline{\alpha}_{t_{i-1}})^{k+1}) + \mathcal{O}((\overline{\beta} - \overline{\beta}_{t_{i-1}})^{k+1}). \qquad (14)$$

By discarding the approximation errors terms $\mathcal{O}(\cdot)$, we can approximate the $n$-th order total derivatives $\widehat{I}_{res}^{\theta(n)}(\cdot)$ and $\widehat{\epsilon}_\theta^{(n)}(\cdot)$ for $n \leq k - 1$ with finite difference [38]. In the case of $k = 1$, Eq. (14) becomes:

$$I_{t_i} = I_{t_{i-1}} - (\overline{\delta}_{t_i} - \overline{\delta}_{t_{i-1}})I_{in} + (\overline{\alpha}_{t_i} - \overline{\alpha}_{t_{i-1}})I_{res}^\theta(I_{t_{i-1}}, I_{in}, t_{i-1}) + (\overline{\beta}_{t_i} - \overline{\beta}_{t_{i-1}})\epsilon_\theta(I_{t_{i-1}}, I_{in}, t_{i-1}). \quad (15)$$

Notably, prior works [75, 23] are a special case of the first-order solver within our methods by omitting the discrete error. For $k = 2$, the calculation of the first derivative requires additional intermediate time points $t_u = rt_i + (1 - r)t_{i-1}$ in the range of $[t_i, t_{i-1}]$ with controllable parameters $r$, and we have:

$$I_{t_i} = I_{t_{i-1}} - (\overline{\delta}_{t_i} - \overline{\delta}_{t_{i-1}})I_{in} + (\overline{\alpha}_{t_i} - \overline{\alpha}_{t_{i-1}})I_{res}^\theta(I_{t_{i-1}}, I_{in}, t_{i-1}) + (\overline{\beta}_{t_i} - \overline{\beta}_{t_{i-1}})\epsilon_\theta(I_{t_{i-1}}, I_{in}, t_{i-1}) + \frac{(\overline{\alpha}_{t_i} - \overline{\alpha}_{t_{i-1}})}{2r} \cdot$$
$$\left(I_{res}^\theta(I_{t_u}, I_{in}, t_u) - I_{res}^\theta(I_{t_{i-1}}, I_{in}, t_{i-1})\right) + \frac{(\overline{\beta}_{t_i} - \overline{\beta}_{t_{i-1}})}{2r} \cdot \left(\epsilon_\theta(I_{t_u}, I_{in}, t_u) - \epsilon_\theta(I_{t_{i-1}}, I_{in}, t_{i-1})\right). \quad (16)$$

For $k \geq 2$, the technical derivation details and algorithms are deferred to Appendix B.

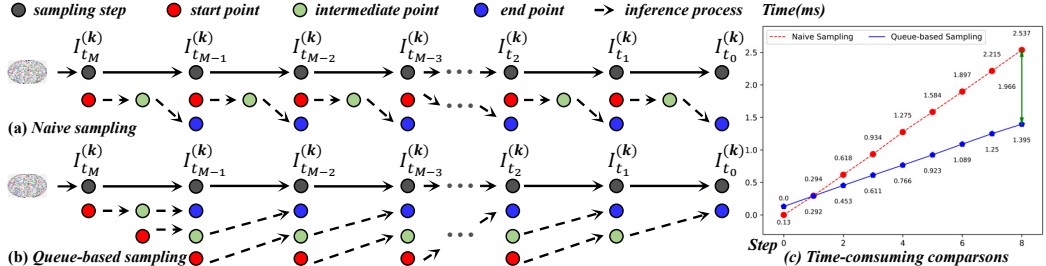

Figure 3: Different sampling strategies of diffusion generalist solvers ($k = 2$).

## 3.3 Universal Posterior Sampling

DPS [7] circumvents the intractability of posterior sampling in diffusion models via a novel approximation, which is generally applicable to inverse problems with accessible kernels. Inspired by this, we propose universal posterior sampling from the perspective of residual modeling. Under well-defined conditions, Eq. (4) adequately unifies diverse degradations into a degradation-agnostic representation, facilitating the perception of commonalities while exacerbating the restoration burden for discriminability reduction. Eq. (3) offers insights that $\nabla_{I_t} \log q(I_{in}|I_t)$ can be adapted to prevent samples from deviating from the generative manifolds when the measurements are deteriorated, as illustrated in Fig. 2. In contrast to linear inverse modeling, we endeavor to leverage the underlying mechanism of generalist modeling by universally formulating the degradation as $I_{in} = I_0 + I_{res}^\theta + n$. Such a model exhibits versatility across various degradation scenarios. In order to seek a tractable approximation for $q(I_{in}|I_t)$, we utilize surrogate functions to maximize the likelihood, yielding approximations of universal posterior sampling (UPS). Specifically, given the formulation $q(I_{in}|I_t) = \mathbb{E}_{I_0, I_{res} \sim q(I_0, I_{res}^\theta|I_t)}[q(I_{in}|I_0, I_{res}^\theta)]$, we can make approximations by replacing the outer expectation of $q(I_{in}|I_0, I_{res}^\theta)$ over the posterior distribution with inner expectations:

$$q(I_{in}|I_t) \simeq q(I_{in}|\widehat{I}_0 := \mathbb{E}[I_0|I_t, \widehat{I}_{res}^\theta], \widehat{I}_{res}^\theta := \mathbb{E}[I_{res}^\theta|I_t]). \tag{17}$$

As the clean images are continuously disrupted by Gaussian noise in the diffusion process, we can derive the closed-form upper bound of the Jensen gap for the general degradation modeling.

**Theorem 1** *For the measurement model $I_{in} = I_0 + I_{res} + n$ in linear verse problem with $n \sim \mathcal{N}(0, \sigma^2 \boldsymbol{I})$, we have (proof in Appendix C):*

$$q(I_{in}|I_t) \simeq q(I_{in}|\widehat{I}_0, \widehat{I}_{res}), \tag{18}$$

*the approximation error can be quantified with the Jensen gap, which is upper bounded by:*

$$\mathcal{J}(\sigma, M) \leq \frac{d}{\sqrt{2\pi\sigma^2}} \exp\left(-\frac{1}{2\sigma^2}\right) M, \tag{19}$$

*where $M := \int \|(I_0 + I_{res}) - (\widehat{I}_0 + \widehat{I}_{res})\| q(I_0, I_{res}|I_t) dI_0 dI_{res}$.*

Note that $M$ is finite for most of the distribution in practice, which can be considered as the generalized absolute distance between the observed reference and estimated data. $\mathcal{J}(\sigma, M)$ can approach 0 as $\sigma \to 0$ or $\infty$, suggesting that the approximation errors reduce with extreme measurement noise. Specifically, if the predictions of $\widehat{I}_0$ and $\widehat{I}_{res}$ are accurate, the upper bound $\mathcal{J}(\sigma, M)|_{\sigma \to 0, M \to 0}$ shrinks due to the low variance and distortion. Oppositely, if the predictions lack accuracy, the upper bound of the $\mathcal{J}(\sigma, M)|_{\sigma \to \infty, M \to M_{max}}$ shrinks as well for large variance and limited distortion. According to Theorem 1, universal posterior sampling can stabilize the multi-step inverse sampling process and provide the tractable gradient approximations of log-likelihood, as measurement distribution is given:

$$\nabla_{I_t} \log q(I_{in}|I_t) \simeq \nabla_{I_t} \log q(I_{in}|\widehat{I}_0, \widehat{I}_{res}^\theta) = -1/\sigma^2 \nabla_{I_t} \|I_{in} - (\widehat{I}_0 + \widehat{I}_{res})\|^2. \tag{20}$$

It serves as a compensatory mechanism to furnish reliable gradient guidance for optimization in the latent manifold, thereby mitigating the restoration burden and efficiency deterioration. Notably, our method is formed through the collaboration of diffusion generalist solvers and universal posterior sampling. Both components exclusively depend on $I_{res}^\theta$ and $\epsilon_\theta$ predictors, which are intended for

Table 1: Quantitative comparisons of five image restoration tasks. The FLOPS is calculated in the inference stage with 256×256 resolution. The best results of task-specific models and universal models are shown in **blue** and **red**, respectively.

| Method | Year | Deraining PSNR↑ | Deraining SSIM↑ | Enhancement PSNR↑ | Enhancement SSIM↑ | Desnowing PSNR↑ | Desnowing SSIM↑ | Dehazing PSNR↑ | Dehazing SSIM↑ | Deblurring PSNR↑ | Deblurring SSIM↑ | Average PSNR↑ | Average SSIM↑ | Params(M) | FLOPs(G) |
|---|---|---|---|---|---|---|---|---|---|---|---|---|---|---|---|
| **Task-specific Method** | | | | | | | | | | | | | | | |
| SwinIR [27] | 2021 | 27.81 | 0.845 | 17.49 | 0.715 | 25.90 | 0.913 | 24.22 | 0.886 | 28.01 | 0.839 | - | - | 0.87 | 58.71 |
| Restormer [67] | 2022 | **30.02** | **0.878** | 23.07 | 0.774 | **33.24** | **0.958** | **32.04** | **0.969** | **32.87** | **0.939** | - | - | 26.12 | 140.99 |
| MAXIM [53] | 2022 | 29.36 | 0.875 | **23.73** | **0.893** | 29.93 | 0.954 | 28.50 | 0.931 | 32.86 | 0.939 | - | - | 22.14 | 158.16 |
| IR-SDE [32] | 2023 | 24.37 | 0.782 | 18.76 | 0.625 | 19.78 | 0.866 | 16.36 | 0.826 | 27.91 | 0.865 | - | - | 137.13 | 379.33 |
| RDDM [29] | 2024 | 25.61 | 0.919 | 19.94 | 0.727 | 26.79 | 0.891 | 27.87 | 0.941 | 27.87 | 0.848 | - | - | 15.47 | 65.87 |
| **Universal Method** | | | | | | | | | | | | | | | |
| Restormer [67] | 2022 | 28.54 | 0.847 | 21.75 | 0.742 | 28.53 | 0.919 | 26.54 | 0.924 | 26.44 | 0.799 | 27.61 | 0.869 | 26.09 | 140.99 |
| AirNet [22] | 2022 | 24.78 | 0.774 | 13.05 | 0.485 | 25.80 | 0.885 | 18.53 | 0.827 | 25.76 | 0.782 | 24.01 | 0.809 | 5.76 | 301.27 |
| Prompt-IR [42] | 2023 | 28.97 | 0.856 | 20.97 | 0.733 | 29.52 | 0.938 | 25.80 | 0.929 | 26.25 | 0.797 | 27.89 | 0.878 | 32.96 | 158.14 |
| ProRes [35] | 2023 | 22.42 | 0.752 | 20.31 | 0.741 | 24.53 | 0.859 | 24.81 | 0.888 | 26.08 | 0.792 | 24.08 | 0.814 | 370.26 | 97.17 |
| IDR [72] | 2023 | 28.40 | 0.844 | 20.95 | 0.706 | 27.77 | 0.911 | 24.48 | 0.914 | 26.33 | 0.799 | 26.96 | 0.863 | 6.19 | 32.16 |
| AutoDIR [18] | 2024 | 29.32 | 0.863 | 15.65 | 0.707 | 15.31 | 0.706 | 19.01 | 0.829 | 28.47 | 0.864 | 22.43 | 0.799 | 115.86 | 63.38 |
| DA-CLIP [33] | 2024 | 28.63 | 0.854 | 19.50 | 0.730 | 28.23 | 0.934 | 27.26 | 0.941 | 26.47 | 0.818 | 27.54 | 0.881 | 32.96 | 158.14 |
| DiffUIR [75] | 2024 | 30.67 | 0.887 | 21.21 | 0.769 | 30.70 | 0.943 | 30.29 | 0.944 | 29.00 | 0.877 | 29.93 | 0.907 | 7.73 | 32.93 |
| Ours-T | - | 28.10 | 0.841 | 19.48 | 0.719 | 26.47 | 0.896 | 22.03 | 0.875 | 25.82 | 0.784 | 25.95 | 0.849 | 0.45 | 5.74 |
| Ours-S | - | 28.99 | 0.852 | 20.36 | 0.749 | 26.98 | 0.905 | 25.73 | 0.906 | 25.89 | 0.785 | 26.93 | 0.861 | 1.07 | 8.01 |
| Ours-B | - | 29.50 | 0.874 | 20.85 | 0.747 | 31.04 | 0.948 | 28.12 | 0.937 | 27.74 | 0.850 | 29.16 | 0.898 | 3.65 | 23.97 |
| Ours-L | - | **31.46** | **0.896** | **23.84** | **0.801** | **32.69** | **0.955** | **31.68** | **0.946** | **30.15** | **0.899** | **31.36** | **0.920** | 7.73 | 32.93 |

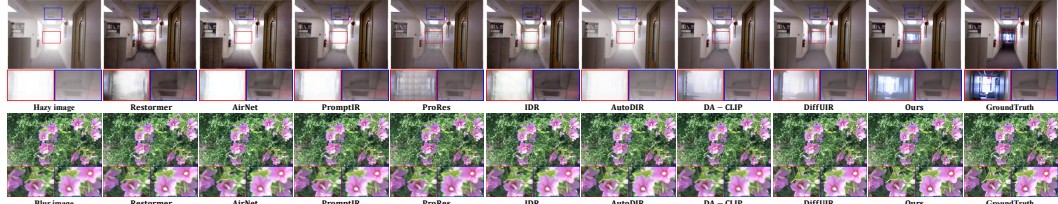

Figure 4: Visualization comparison with state-of-the-art methods on different restoration tasks.

inverse inference. Once the neural networks are adequately optimized, restoration improvements can be achieved without re-training. For the different solvers in Eq. (14), we employ universal posterior sampling to modify the noise $\epsilon_\theta(\cdot)$ predicted by neural networks as follows (See Appendix D):

$$\widehat{\epsilon}_\theta(\widehat{I}_{t_i}, I_{in}, t_i) := \epsilon_\theta(\widehat{I}_{t_i}, I_{in}, t_i) + \overline{\beta}_{t_i}/\sigma^2 \nabla_{I_{t_i}} \|I_{in} - (\widehat{I}_0 + I^\theta_{res}(I_{t_i}, I_{in}, t_i))\|. \tag{21}$$

Our computational overhead primarily originates from the total derivatives of $\widehat{I}^{\theta(n)}_{res}(\cdot), \widehat{\epsilon}^{(n)}_\theta(\cdot)$, and the approximation of $\nabla_{I_t} \log q(I_{in}|I_t)$. To accelerate the inverse inference, we introduce a queue-based sampling strategy to improve efficiency, as illustrated in Fig. 3. Specifically, we precompute the items of intermediate time points during the initial sampling, and subsequent sampling operations will use these items of intermediate time points as their starting points. Furthermore, each sampling iteration solely necessitates the calculation of updated intermediate time points to obtain the final results. In this context, for a $k$-order solver, naive sampling from time $s$ to $t$ requires interpolating $k-1$ intermediate points within the interval, resulting in $k$ number of function evaluations (NFEs) per step and $\mathcal{O}(nk)$ computational complexity for $n$ steps. In contrast, the queue-based sampling precomputes $k-1$ time points and caches them for reuse in subsequent steps, reducing total complexity to $\mathcal{O}(n+k-1)$. In conclusion, this approach improves efficiency by effectively reducing the consumption of computational resources. For more details, please refer to the Appendix D.

## 4 Experiment

### 4.1 Benchmarks and Implementations

Extensive experiments are conducted on five image restoration tasks, encompassing deraining [17, 43, 66, 69, 3, 24], low-light enhancement [61, 30, 36], desnowing [4, 31], dehazing [21, 8, 1, 2] and deblurring [39, 46]. Datasets details are summarized in Appendix E.1, and we use PSNR [16] and SSIM [60] to evaluate the performance on RGB space. For fairness, five task-specific methods [27, 67, 53, 32, 29] are trained separately on task-related datasets and all universal methods [67, 42, 35, 72, 33, 75] except for [18] are re-implemented on the mixed datasets.

Table 2: Effects of universal posterior sampling and solver orders of our method.

| Configurations | | | Deraining | | Enhancement | | Desnowing | | Dehazing | | Deblurring | | Average | |
|---|---|---|---|---|---|---|---|---|---|---|---|---|---|---|
| | Order | UPS | PSNR↑ | SSIM↑ | PSNR↑ | SSIM↑ | PSNR↑ | SSIM↑ | PSNR↑ | SSIM↑ | PSNR↑ | SSIM↑ | PSNR↑ | SSIM↑ |
| (i) | $k=1$ | ✗ | 30.69 | 0.886 | 23.04 | 0.795 | 32.06 | 0.952 | 31.47 | 0.947 | 29.33 | 0.887 | 30.70 | 0.913 |
| (ii) | $k=1$ | ✓ | 31.10 | 0.892 | 23.07 | 0.798 | 32.32 | 0.954 | 31.54 | 0.947 | 29.80 | 0.895 | 31.01 | 0.918 |
| (iii) | $k=2$ | ✗ | 31.31 | 0.894 | 23.08 | 0.798 | 32.51 | 0.955 | 31.62 | 0.948 | 30.07 | 0.897 | 31.20 | 0.919 |
| (iv) | $k=2$ | ✓ | 31.46 | 0.896 | 23.84 | 0.801 | 32.69 | 0.955 | 31.68 | 0.946 | 30.15 | 0.899 | 31.36 | 0.920 |
| (v) | $k=3$ | ✗ | 31.33 | 0.894 | 23.13 | 0.799 | 32.52 | 0.955 | 31.71 | 0.948 | 30.09 | 0.898 | 31.23 | 0.919 |
| (vi) | $k=3$ | ✓ | 31.45 | 0.896 | 23.84 | 0.801 | 32.69 | 0.955 | 31.67 | 0.946 | 30.15 | 0.899 | 31.36 | 0.920 |

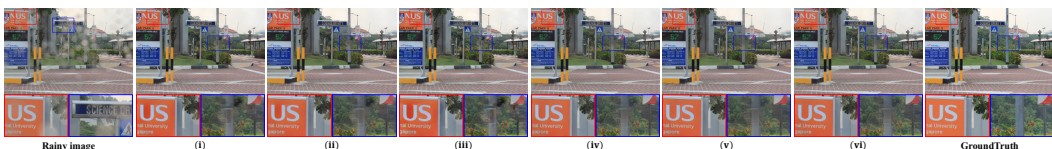

| Rainy image | (i) | (ii) | (iii) | (iv) | (v) | (vi) | GroundTruth |

Figure 5: Visualization effects of varied solver and universal posterior sampling.

Our method is trained using 8 Nvidia A800 40GB GPUs with PyTorch [41] framework for 210h. Adam optimizer and L1 loss are employed for 500k iterations with a learning rate of 1e-4. We set the batch size as 20 and distribute it evenly to each task. We randomly crop patches with a size of $256 \times 256$ from the original image as network input for training and use 8 timesteps for full-resolution testing. Additionally, we offer an adaptation for the Ascend 910B 64GB NPUs, aligned with our GPU settings. We use U-Net [47] architecture and change the channel number of the hidden layers $C$ to obtain different versions with varied parameter quantities: Ours-T(C = 32, 32, 32, 32), Ours-S(C = 32, 64, 64, 128), Ours-B(C = 64, 128, 128, 256), Ours-L(C = 64, 128, 256, 512).

## 4.2 Comparative Experiments

Quantitative results are presented in Tab. 1. In contrast to universal methods, we achieve substantial performance improvement across all tasks, indicating our highest accuracy. Compared with task-specific methods, we still achieve competitive results with the lowest number of parameters and computational overhead. Evidently, the performance of our method gets better as the network capacity increases, fully validating its scalability. Visual comparisons are shown in Fig. 4, and for more results please refer to Appendix E.2. Our method outperforms others and is the most similar to ground-truth.

## 4.3 Ablation Study

To thoroughly harness the effectiveness of each component, we conduct ablation studies that include three categories: (i) Effects of varied solver orders and universal posterior sampling, (ii) influence of the value of $\bar{\delta}_T$, and (iii) impact of the sampling steps.

**Effects of varied solver orders and universal posterior sampling.** We present the qualitative comparisons and quantitative analysis in Fig. 5 and Tab. 2, respectively. Obviously, metric assessments improve as order increases and the performance of our method reaches saturation in configuration (iv). Besides, the activation of UPS can steadily enhance the solution accuracy and its effect weakens as solver order increases. Therefore, we select $k=2$ with UPS enabled as our default settings to strike a balance between performance and efficiency.

Table 3: Quantitative results of our method with different values of $\bar{\delta}_T$.

| $\bar{\delta}_T$ | Deraining | | Enhancement | | Desnowing | | Dehazing | | Deblurring | | Average | |
|---|---|---|---|---|---|---|---|---|---|---|---|---|
| | PSNR↑ | SSIM↑ | PSNR↑ | SSIM↑ | PSNR↑ | SSIM↑ | PSNR↑ | SSIM↑ | PSNR↑ | SSIM↑ | PSNR↑ | SSIM↑ |
| 0.1 | 29.44 | 0.875 | 21.03 | 0.751 | 32.24 | 0.955 | 29.11 | 0.942 | 29.31 | 0.883 | 29.99 | 0.908 |
| 0.25 | 29.44 | 0.877 | 21.75 | 0.765 | 32.25 | 0.956 | 29.57 | 0.945 | 29.36 | 0.884 | 30.10 | 0.910 |
| 0.5 | 30.83 | 0.887 | 22.56 | 0.786 | 32.12 | 0.953 | 31.58 | 0.944 | 30.13 | 0.897 | 30.91 | 0.915 |
| 0.75 | 31.37 | 0.896 | 22.46 | 0.786 | 32.32 | 0.955 | 31.52 | 0.948 | 30.26 | 0.899 | 31.15 | 0.919 |
| 0.9 | 31.40 | 0.896 | 23.75 | 0.807 | 32.34 | 0.954 | 32.05 | 0.946 | 30.13 | 0.897 | 31.26 | 0.919 |
| 1 | 31.46 | 0.896 | 23.84 | 0.801 | 32.69 | 0.955 | 31.68 | 0.946 | 30.15 | 0.899 | 31.36 | 0.920 |

Table 4: Restoration performance of different sampling steps.

| Steps | Deraining | | Enhancement | | Desnowing | | Dehazing | | Deblurring | | Average | |
|---|---|---|---|---|---|---|---|---|---|---|---|---|
| | PSNR↑ | SSIM↑ | PSNR↑ | SSIM↑ | PSNR↑ | SSIM↑ | PSNR↑ | SSIM↑ | PSNR↑ | SSIM↑ | PSNR↑ | SSIM↑ |
| 4 | 31.35 | 0.895 | 23.30 | 0.801 | 32.56 | 0.956 | 31.70 | 0.948 | 30.09 | 0.898 | 31.25 | 0.920 |
| 7 | 31.33 | 0.894 | 23.22 | 0.800 | 32.52 | 0.956 | 31.66 | 0.948 | 30.08 | 0.898 | 31.22 | 0.919 |
| 8 | 31.46 | 0.896 | 23.84 | 0.801 | 32.69 | 0.955 | 31.68 | 0.946 | 30.15 | 0.899 | 31.36 | 0.920 |
| 9 | 31.31 | 0.894 | 22.99 | 0.798 | 32.46 | 0.955 | 31.67 | 0.948 | 30.05 | 0.897 | 31.18 | 0.919 |
| 10 | 31.28 | 0.893 | 23.09 | 0.799 | 32.46 | 0.955 | 31.57 | 0.948 | 30.04 | 0.897 | 31.16 | 0.919 |

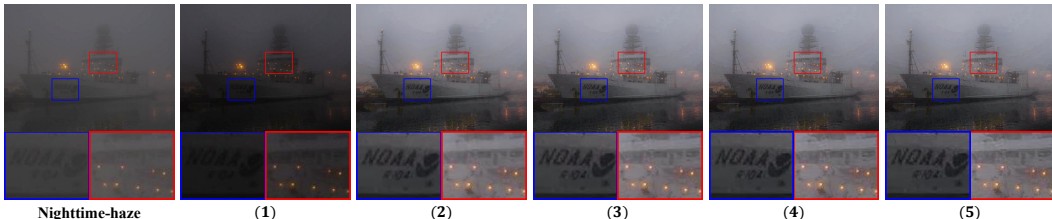

Figure 6: Results of low-light enhancement and dehazing for five iterations on NHRW [70].

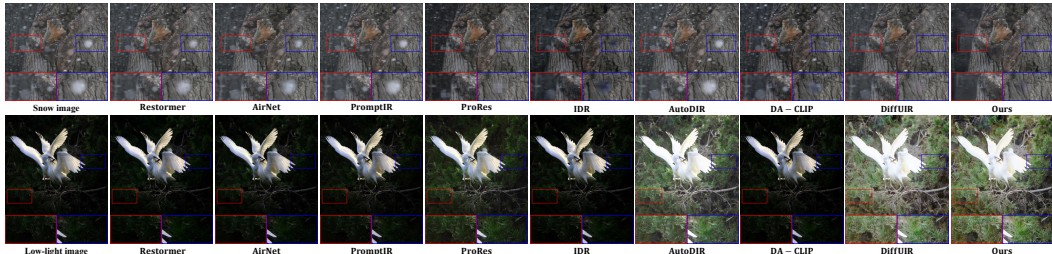

Figure 7: Visualization of real-world scenes generalization.

**Influence of the value of $\bar{\delta}_T$.** $\bar{\delta}_T$ determines the commonality of degradation representation in a positively correlated manner. We show the restoration performance with various $\bar{\delta}_T$ settings in Tab. 3. Clearly, the optimal commonality ratio varies across different tasks, and the performance under low commonality is markedly inferior to that under high commonality. To fully verify that our method can handle complicated distribution conditions, we perform image restoration with $\bar{\delta}_T = 1$.

**Impact of the sampling steps.** Restoration performance and efficiency of the inverse inference depend on sampling steps, as shown in Tab. 4 and Tab. A4, respectively. As a result, increasing the number of sampling steps beyond a certain point leads to a decline in PSNR/SSIM. This counter-intuitive phenomenon may stem from two main factors. (1) Our approach for image restoration is motivated by the principle of commonality, aiming to handle diverse degradation types within a unified framework. The model tends to first remove the primary degradation before addressing secondary ones. Consequently, in cases where samples from the deraining or deblurring datasets also suffer from additional degradations (e.g., low-light conditions), the restored output may diverge from the available reference, leading to reduced evaluation metrics despite better perceptual quality, as presented in Appendix E.3. (2) The stochasticity in both the forward and reverse processes, combined with network estimated errors, can cause slight influence on performance. Considering the balance between efficiency and performance, we select 8 as the optimal number of sampling steps.

### 4.4 Compound Image Restoration

To demonstrate that our method can handle more complex restoration tasks where various degradations occur in one image, we repeatedly perform our method on the degraded images, as displayed in Fig. 6. Evidently, our method eliminates the former degradation before the latter one within its capability range. It incurs no additional information interference for well-restored data, ensuring high fidelity. For more visualization, please refer to Appendix E.3.

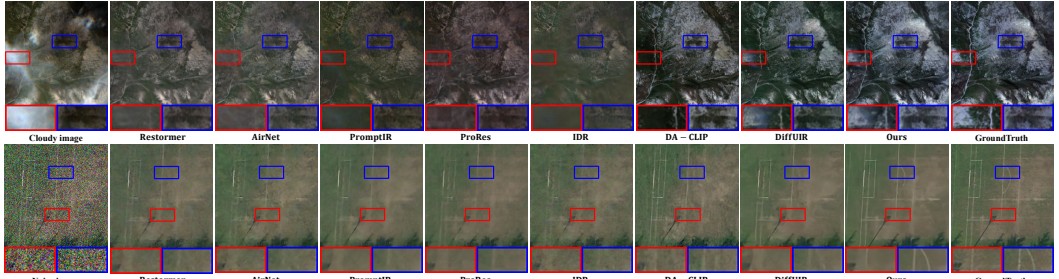

Figure 8: Visualization of typical remote sensing restoration tasks.

Table 5: Performance comparison of our method with other universal image restoration methods in remote sensing image restoration tasks. The best results are highlighted in **bold**.

| Method | Year | Denoising | | Enhancement | | Deblur | | Decloud | | Super-resolve | | Average | |
|---|---|---|---|---|---|---|---|---|---|---|---|---|---|
| | | PSNR↑ | SSIM↑ | PSNR↑ | SSIM↑ | PSNR↑ | SSIM↑ | PSNR↑ | SSIM↑ | PSNR↑ | SSIM↑ | PSNR↑ | SSIM↑ |
| Restomer [67] | 2022 | 29.21 | 0.773 | 32.61 | 0.976 | 28.14 | 0.781 | 20.13 | 0.520 | 28.58 | 0.809 | 28.97 | 0.810 |
| AirNet [22] | 2022 | 28.37 | 0.770 | 25.91 | 0.881 | 26.00 | 0.684 | 20.04 | 0.516 | 28.06 | 0.798 | 26.58 | 0.761 |
| Prompt-IR [42] | 2023 | 30.06 | 0.796 | 33.03 | 0.982 | 25.72 | 0.639 | 19.54 | 0.498 | 27.97 | 0.788 | 28.56 | 0.778 |
| ProRes [35] | 2023 | 28.30 | 0.769 | 30.73 | 0.932 | 25.63 | 0.654 | 20.28 | 0.508 | 27.95 | 0.793 | 27.57 | 0.764 |
| IDR [72] | 2023 | 27.56 | 0.695 | 30.32 | 0.967 | 25.32 | 0.625 | 18.98 | 0.476 | 27.81 | 0.784 | 27.07 | 0.740 |
| DA-CLIP [33] | 2024 | 29.98 | 0.780 | 33.69 | 0.986 | 27.80 | 0.747 | 19.06 | 0.463 | 27.11 | 0.762 | 29.01 | 0.793 |
| DiffUIR [75] | 2024 | 29.59 | 0.740 | 35.28 | 0.988 | 30.01 | 0.837 | 20.34 | 0.554 | 28.63 | 0.818 | 30.17 | 0.821 |
| Ours-T | - | 25.57 | 0.671 | 28.34 | 0.963 | 24.78 | 0.610 | 17.40 | 0.425 | 27.63 | 0.786 | 25.79 | 0.725 |
| Ours-S | - | 27.38 | 0.713 | 31.59 | 0.976 | 26.97 | 0.711 | 18.12 | 0.462 | 28.49 | 0.809 | 27.78 | 0.772 |
| Ours-B | - | 29.22 | 0.748 | 32.00 | 0.980 | 29.48 | 0.829 | 18.96 | 0.510 | 28.75 | 0.823 | 29.10 | 0.817 |
| Ours-L | - | **31.83** | **0.802** | **35.52** | **0.988** | **31.84** | **0.875** | **21.35** | **0.597** | **29.47** | **0.839** | **31.52** | **0.854** |

## 4.5 Zero-shot Generalization in Real-world Scenes

To evaluate the generalization ability, we conduct the zero-shot restoration on real-world scenes. We generalize all methods to the degradation scene that matches our task specification. Due to the absence of ground-truth, we merely provide the qualitative results, as shown in Fig. 7. Obviously, we outperform others and achieve the best visual effects. More comparisons are placed in Appendix E.4.

## 4.6 Remote Sensing Image Restoration

Remote sensing imaging possesses different depth attributes and degradation types. To further validate the superiority of our method, we adapt all universal methods to remote sensing image restoration tasks, including denoising, illumination adjustment, deblurring, cloud removal, and super-resolution. Metric evaluations and visual comparisons are presented in Tab. 5 and Fig. 8, respectively. We achieve the best quantitative and qualitative results (See more details in Appendix E.5).

## 5 Conclusion

In this paper, we introduced DGSolver, a diffusion generalist solver with universal posterior sampling for image restoration. We derived exact ODE formulations for diffusion generalist models and developed customized high-order solvers with a queue-based sampling strategy to reduce cumulative discretization errors and enhance efficiency. Additionally, our universal posterior sampling improves the approximation of manifold-constrained gradients, leading to more accurate noise and residual estimation. Extensive experiments across natural and remote sensing datasets demonstrate that DGSolver consistently outperforms state-of-the-art methods in accuracy, stability, and scalability.

## Acknowledgments and Disclosure of Funding

This work was supported in part by the National Natural Science Foundation of China (62276192, 62225113, 624B2109, 623B2079), in part by the Zhongguancun Academy Project (20240308), and in part by the Fund of National Key Laboratory of Multispectral Information Intelligent Processing Technology (61421132302).

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

# A Mathematical Theory of Diffusion Models

Diffusion models that are most pertinent to our methods are theoretically discussed as follows, including diffusion probabilistic models (DPMs), diffusion residual models (DRMs), and diffusion generalist models (DGMs). For each category, we analyze their discrete formulations of perturbed generation process [14] and deterministic implicit sampling [50], and continuous formulations of score-based stochastic differential equations (SDEs).

## A.1 Forward Process of Diffusion Probabilistic Models

Diffusion probabilistic models consist of a non-parametric noise addition forward process and a denoising reverse process. In the forward process, it gradually disrupts the original image content by adding the Gaussian noise. The sampling distribution in a markovian manner between the adjacent time step, *i.e.* $I_t$ and $I_{t-1}$, can be defined as $q(I_t|I_{t-1}) = \mathcal{N}(\alpha_t I_{t-1}, \beta_t \boldsymbol{I})$, which is equivalent to:

$$I_t = \alpha_t I_{t-1} + \beta_t \epsilon, \tag{22}$$

where $\alpha_t$ and $\beta_t = \sqrt{1 - \alpha_t^2}$ are predefined coefficients over time $t$ to modulate the image content and noise intensity, respectively. $\epsilon$ is a random variable following the normal Gaussian distribution $\mathcal{N}(0, \boldsymbol{I})$.

## A.2 Perturbed Generation Process of Diffusion Probabilistic Models

By the property of the markovian chain, Eq. (22) can be converted to the below formula that DPMs can diffuse from the clean image $I_0$ to any intermediate variable $I_t$ at time step $t$ in just one step:

$$I_t = \overline{\alpha}_t I_0 + \overline{\beta}_t \epsilon, \tag{23}$$

where $\overline{\alpha}_t = \prod_{i=0}^t \alpha_i$ and $\overline{\beta}_t = \sqrt{1 - \overline{\alpha}_t^2}$ are denoted for simplicity. Based on Bayesian formula, we have:

$$q(I_t|I_{t-1}) = \mathcal{N}(\alpha_t I_{t-1}, \beta_t \boldsymbol{I}),\ q(I_{t-1}|I_0) = \mathcal{N}(\overline{\alpha}_{t-1} I_0, \overline{\beta}_{t-1} \boldsymbol{I}),\ q(I_t|I_0) = \mathcal{N}(\overline{\alpha}_t I_0, \overline{\beta}_t \boldsymbol{I}). \tag{24}$$

$$q(I_{t-1}|I_t, I_0) = \frac{q(I_t|I_{t-1})q(I_{t-1}|I_0)}{q(I_t|I_0)} = \exp\{\log q(I_t|I_{t-1}) + \log q(I_{t-1}|I_0) - \log q(I_t|I_0)\}$$

$$\propto \exp\left\{ -\frac{(I_t - \alpha_t I_{t-1})^2}{2\beta_t^2} - \frac{(I_{t-1} - \overline{\alpha}_{t-1} I_0)^2}{2\overline{\beta}_{t-1}^2} + \frac{(I_t - \overline{\alpha}_t I_0)^2}{2\overline{\beta}_t^2} \right\} \tag{25}$$

$$\propto \exp\left\{ -\frac{1}{2}(\frac{\alpha_t^2}{\beta_t^2} + \frac{1}{\overline{\beta}_{t-1}^2})I_{t-1}^2 - 2I_{t-1}(\frac{\alpha_t I_t}{\beta_t^2} + \frac{\overline{\alpha}_{t-1} I_0}{\overline{\beta}_{t-1}^2}) + C(I_t, I_0) \right\}.$$

Apparently, Eq. (25) can be supposed as a general Gaussian distribution $\mathcal{N}(\mu_{t-1}, \Sigma_{t-1} \boldsymbol{I})$:

$$\Sigma_{t-1}^2(I_{t-1}|I_t, I_0) = (\frac{\alpha_t^2}{\beta_t^2} + \frac{1}{\overline{\beta}_{t-1}^2})^{-1} = \frac{\beta_t^2 \overline{\beta}_{t-1}^2}{\overline{\beta}_t^2}, \tag{26}$$

$$\mu_{t-1}(I_{t-1}|I_t, I_0) = (\frac{\alpha_t I_t}{\beta_t^2} + \frac{\overline{\alpha}_{t-1} I_0}{\overline{\beta}_{t-1}^2})\frac{\beta_t^2 \overline{\beta}_{t-1}^2}{\overline{\beta}_t^2} = \frac{\alpha_t \overline{\beta}_{t-1}^2}{\overline{\beta}_t^2} I_t + \frac{\overline{\alpha}_{t-1} \beta_t^2}{\overline{\beta}_t^2} I_0. \tag{27}$$

Consequently, the inverse process is regarded as sampling from the following Gaussian distribution step by step:

$$q(I_{t-1}|I_t, I_0) = \mathcal{N}(\frac{\alpha_t \overline{\beta}_{t-1}^2}{\overline{\beta}_t^2} I_t + \frac{\overline{\alpha}_{t-1} \beta_t^2}{\overline{\beta}_t^2} I_0, \frac{\beta_t^2 \overline{\beta}_{t-1}^2}{\overline{\beta}_t^2} \boldsymbol{I}), \tag{28}$$

$$I_{t-1} = \frac{\alpha_t \overline{\beta}_{t-1}^2}{\overline{\beta}_t^2} I_t + \frac{\overline{\alpha}_{t-1} \beta_t^2}{\overline{\beta}_t^2} I_0 + \frac{\beta_t^2 \overline{\beta}_{t-1}^2}{\overline{\beta}_t^2} \epsilon. \tag{29}$$

## A.3 Deterministic Implicit Sampling of Diffusion Probabilistic Models

Discarding the markovian modeling $q(I_t|I_{t-1})$ in the forward process and assuming that the sampling distribution still follows the Gaussian distribution in the reverse process, we can obtain:

$$q(I_t|I_0) = \mathcal{N}(I_t; \overline{\alpha}_t I_0, \overline{\beta}_t^2 \boldsymbol{I}) \rightarrow I_t = \overline{\alpha}_t I_0 + \overline{\beta}_t \epsilon_1, \tag{30}$$

$$q(I_{t-1}|I_0) = \mathcal{N}(I_{t-1}; \overline{\alpha}_{t-1} I_0, \overline{\beta}_{t-1}^2 \boldsymbol{I}) \rightarrow I_{t-1} = \overline{\alpha}_{t-1} I_0 + \overline{\beta}_{t-1} \epsilon_2, \tag{31}$$

$$q(I_{t-1}|I_0, I_t) = \mathcal{N}(I_{t-1}; \kappa_t I_t + \lambda_t I_0, \sigma_t^2 \boldsymbol{I}) \rightarrow I_{t-1} = \kappa_t I_t + \lambda_t I_0 + \sigma_t \epsilon_3. \tag{32}$$

Substituting Eq. (30) into Eq. (32) and comparing the coefficients with Eq. (31), we have:

$$I_{t-1} = (\kappa_t \overline{\alpha}_t + \lambda_t)I_0 + \kappa_t \overline{\beta}_t \epsilon_1 + \sigma_t \epsilon_3 = (\kappa_t \overline{\alpha}_t + \lambda_t)I_0 + \sqrt{\kappa_t^2 \overline{\beta}_t^2 + \sigma_t^2}\epsilon, \tag{33}$$

$$\overline{\alpha}_{t-1} = \kappa_t \overline{\alpha}_t + \lambda_t, \quad \overline{\beta}_{t-1}^2 = \kappa_t^2 \overline{\beta}_t^2 + \sigma_t^2, \tag{34}$$

these parameters can be calculated as:

$$\kappa_t = \frac{\sqrt{\overline{\beta}_{t-1}^2 - \sigma_t^2}}{\overline{\beta}_t}, \quad \lambda_t = \overline{\alpha}_{t-1} - \frac{\overline{\alpha}_t \sqrt{\overline{\beta}_{t-1}^2 - \sigma_t^2}}{\overline{\beta}_t}. \tag{35}$$

Consequently, a common reverse process in the non-markovian manner is:

$$q(I_{t-1}|I_0, I_t) = \mathcal{N}(\frac{\sqrt{\overline{\beta}_{t-1}^2 - \sigma_t^2}}{\overline{\beta}_t}I_t + (\overline{\alpha}_{t-1} - \frac{\overline{\alpha}_t \sqrt{\overline{\beta}_{t-1}^2 - \sigma_t^2}}{\overline{\beta}_t})I_0, \sigma_t^2 \boldsymbol{I}). \tag{36}$$

If we set the variance $\sigma_t$ equal to $\frac{\beta_t^2 \overline{\beta}_{t-1}^2}{\overline{\beta}_t^2}$, Eq. (36) holds the identical distributions with Eq. (28). Typically, we can derive another representative sampling distribution by setting variance $\sigma_t$ equal to 0:

$$q(I_{t-1}|I_0, I_t)|_{\sigma_t=0} = \mathcal{N}(\frac{\overline{\beta}_{t-1}}{\overline{\beta}_t}I_t + (\overline{\alpha}_{t-1} - \frac{\overline{\alpha}_t \overline{\beta}_{t-1}}{\overline{\beta}_t})I_0, \sigma_t^2 \boldsymbol{I}), \tag{37}$$

in this way, the inverse process is deterministic for ignoring the random noise:

$$I_{t-1} = \frac{\overline{\beta}_{t-1}}{\overline{\beta}_t}I_t + (\overline{\alpha}_{t-1} - \frac{\overline{\alpha}_t \overline{\beta}_{t-1}}{\overline{\beta}_t})I_0. \tag{38}$$

## A.4 Training Objective of Diffusion Probabilistic Models

The training objective for the noise estimators $\epsilon_\theta$ in DPMs is calculated as follows:

$$\mathcal{L} = \mathbb{E}_{I_0, \epsilon}\left[\|\epsilon - \epsilon_\theta(\overline{\alpha}_t I_0 + \overline{\beta}_t \epsilon)\|^2\right]. \tag{39}$$

## A.5 Stochastic Differential Equations of Diffusion Probabilistic Models

The formulation of SDEs of DPMs' forward process can be reformulated from a continuous perspective with diffusion term $f_t(I_t)$ and drift term $g_t$ as follows:

$$dI_t = f_t(I_t)dt + g_t d\omega_t, \tag{40}$$

where $\omega_t$ is the standard Wiener process. Its formulation of inverse SDEs can be proved as follows [20]:

$$dI_t = [f_t(I_t) - g_t^2 \nabla_{I_t} \log p_{I_t}(I_t)]dt + g_t d\overline{\omega}_t. \tag{41}$$

Similarly, we modify the formulas in Eq. (30)-(32) and consider the linear solution $f_t(I_t) = f_t I_t$:

$$q(I_{t+\Delta t}|I_0) = \mathcal{N}(I_{t+\Delta t}; \overline{\alpha}_{t+\Delta t}I_0, \overline{\beta}_{t+\Delta t}^2 \boldsymbol{I}) \rightarrow I_{t+\Delta t} = \overline{\alpha}_{t+\Delta t}I_0 + \overline{\beta}_{t+\Delta t}\epsilon_1, \tag{42}$$

$$q(I_t|I_0) = \mathcal{N}(I_t; \overline{\alpha}_t I_0, \overline{\beta}_t^2 \boldsymbol{I}) \rightarrow I_t = \overline{\alpha}_t I_0 + \overline{\beta}_t \epsilon_2, \tag{43}$$

$$q(I_{t+\Delta t}|I_t) = \mathcal{N}(I_{t+\Delta t}; (1 + f_t \Delta t)I_t, g_t^2 \Delta t \boldsymbol{I}) \rightarrow I_{t+\Delta t} = (1 + f_t \Delta t)I_t + g_t \sqrt{\Delta t}\epsilon_3. \tag{44}$$

We can obtain:

$$\overline{\alpha}_{t+\Delta t} = (1 + f_t \Delta t)\overline{\alpha}_t, \tag{45}$$

$$\overline{\beta}_{t+\Delta t}^2 = (1 + f_t \Delta t)^2 \overline{\beta}_t^2 + g_t^2 \Delta t, \tag{46}$$

By setting $\Delta t \rightarrow 0$, we can derive:

$$f_t = \frac{1}{\overline{\alpha}_t}\frac{d\overline{\alpha}_t}{dt} = \frac{d}{dt}(\ln \overline{\alpha}_t), \quad g_t^2 = \overline{\alpha}_t^2 \frac{d}{dt}\left(\frac{\overline{\beta}_t^2}{\overline{\alpha}_t^2}\right) = 2\overline{\alpha}_t \overline{\beta}_t \frac{d}{dt}\left(\frac{\overline{\beta}_t}{\overline{\alpha}_t}\right). \tag{47}$$

## A.6 Forward Process of Diffusion Residual models

Diffusion residual models additionally incorporate a residual term $I_{res}$ to assist the forward process. The forward process between adjacent steps is defined as:

$$q(I_t|I_{t-1}, I_{res}) = \mathcal{N}(I_t; I_{t-1} + \alpha_t I_{res}, \beta_t^2 \boldsymbol{I}), \tag{48}$$

$$I_t = I_{t-1} + \alpha_t I_{res} + \beta_t \epsilon, \tag{49}$$

where $I_{res} = I_{in} - I_0$. By the properties of Markov Chain and reparameterization, we have:

$$q(I_t|I_0, I_{res}) = \mathcal{N}(I_t; I_0 + \overline{\alpha}_t I_{res}, \overline{\beta}_t^2 \boldsymbol{I}), \tag{50}$$

$$
\begin{aligned}
I_t &= I_{t-1} + \alpha_t I_{res} + \beta_t \epsilon_t = \left[I_{t-2} + \alpha_{t-1} I_{res} + \beta_{t-1}\epsilon_{t-1}\right] + \alpha_t I_{res} + \beta_t \epsilon_t \\
&= I_{t-2} + \left[\alpha_{t-1} + \alpha_t\right] I_{res} + \sqrt{\left[\beta_{t-1}^2 + \beta_t^2\right]}\epsilon \\
&= \cdots \\
&= I_0 + \overline{\alpha}_t I_{res} + \overline{\beta}_t \epsilon,
\end{aligned}
\tag{51}
$$

where $\overline{\alpha}_t = \sum_{i=1}^{t} \alpha_i$ and $\overline{\beta}_t = \sqrt{\sum_{i=1}^{t} \beta_i^2}$.

## A.7 Perturbed Generation Process of Diffusion Residual Models

Evidently, if $\overline{\alpha}_t \to 1$, the forward process makes the representations of $I_0$ be the degraded version with additive noise $I_t = I_{in} + \overline{\beta}_t \epsilon$. Similarly, we can obtain:

$$q(I_t|I_{t-1}, I_{res}) = \mathcal{N}(I_t; I_{t-1} + \alpha_t I_{res}, \beta_t^2 \boldsymbol{I}), \tag{52}$$

$$q(I_{t-1}|I_0, I_{res}) = \mathcal{N}(I_{t-1}; I_0 + \overline{\alpha}_{t-1} I_{res}, \overline{\beta}_{t-1}^2 \boldsymbol{I}), \tag{53}$$

$$q(I_t|I_0, I_{res}) = \mathcal{N}(I_t; I_0 + \overline{\alpha}_t I_{res}, \overline{\beta}_t^2 \boldsymbol{I}). \tag{54}$$

Suppose the form of $q(I_{t-1}|I_t, I_0, I_{res})$ belongs to Gaussian distribution family as all the distributions are Gaussian, we can denote:

$$q(I_{t-1}|I_t, I_0, I_{res}) = \mathcal{N}(I_{t-1}|\mu_{t-1}(I_t, I_0, I_{res}), \Sigma_{t-1}(I_t, I_0, I_{res})). \tag{55}$$

To derive the expressions of variables $\mu_{t-1}(I_t, I_0, I_{res})$ and $\Sigma_{t-1}(I_t, I_0, I_{res})$, we take the logarithm directly and ignore the constant term:

$$q(I_{t-1}|I_t, I_0, I_{res}) = \frac{q(I_t|I_{t-1}, I_0, I_{res})q(I_{t-1}|I_0, I_{res})}{q(I_t|I_0, I_{res})}, \tag{56}$$

$$
\begin{aligned}
q(I_{t-1}|I_t, I_0, I_{res}) &= \exp\left\{\log q(I_t|I_{t-1}, I_0, I_{res}) + \log q(I_{t-1}|I_0, I_{res}) - \log q(I_t|I_0, I_{res})\right\} \\
&\propto \exp\left\{-\frac{\left(I_t - (I_{t-1} + \alpha_t I_{res})\right)^2}{2\beta_t^2} - \frac{\left(I_{t-1} - (I_0 + \overline{\alpha}_{t-1} I_{res})\right)^2}{2\overline{\beta}_{t-1}^2} + \frac{\left(I_t - (I_0 + \overline{\alpha}_t I_{res})\right)^2}{2\overline{\beta}_t^2}\right\} \\
&\propto \exp\left\{-\frac{1}{2}\left((\frac{1}{\beta_t^2} + \frac{1}{\overline{\beta}_{t-1}^2})I_{t-1}^2 - 2I_{t-1}\left(\frac{I_t - \alpha_t I_{res}}{\beta_t^2} + \frac{I_0 + \overline{\alpha}_{t-1} I_{res}}{\overline{\beta}_{t-1}^2}\right) + C(I_t, I_0, I_{res})\right)\right\}.
\end{aligned}
\tag{57}
$$

Comparing the coefficients, we can derive the variables as below:

$$\Sigma_{t-1}^2(I_t, I_0, I_{res}) = \frac{\beta_t^2 \overline{\beta}_{t-1}^2}{\beta_t^2 + \overline{\beta}_{t-1}^2} = \frac{\beta_t^2 \overline{\beta}_{t-1}^2}{\overline{\beta}_t^2}, \tag{58}$$

$$\mu_{t-1}(I_t, I_0, I_{res}) = \left(\frac{I_t - \alpha_t I_{res}}{\beta_t^2} + \frac{I_0 + \overline{\alpha}_{t-1} I_{res}}{\overline{\beta}_{t-1}^2}\right)\frac{\beta_t^2 \overline{\beta}_{t-1}^2}{\overline{\beta}_t^2} = I_t - \alpha_t I_{res} - \frac{\beta_t^2}{\overline{\beta}_t}\epsilon. \tag{59}$$

## A.8 Deterministic Implicit Sampling of Diffusion Residual Models

For generality, we discard the assumptions that the forward process follows a Markov Chain and denote:

$$q(I_{t-1}|I_0, I_{res}) = \mathcal{N}(I_{t-1}; I_0 + \overline{\alpha}_{t-1} I_{res}, \overline{\beta}_{t-1}^2 \boldsymbol{I}), \tag{60}$$

$$q(I_t|I_0, I_{res}) = \mathcal{N}(I_t; I_0 + \overline{\alpha}_t I_{res}, \overline{\beta}_t^2 \boldsymbol{I}), \tag{61}$$

then the conditional distribution $q(I_{t-1}|I_0, I_t)$ can be defined as:

$$q(I_{t-1}|I_0, I_t, I_{res}) = \mathcal{N}(I_{t-1}; \lambda_t I_0 + \kappa_t I_t + \eta_t I_{res}, \sigma_t^2 \boldsymbol{I}). \tag{62}$$

we can reformulate the probability formulas into algebraic expressions:

$$I_{t-1} = I_0 + \overline{\alpha}_{t-1} I_{res} + \overline{\beta}_{t-1}\epsilon, \tag{63}$$

$$I_t = I_0 + \overline{\alpha}_t I_{res} + \overline{\beta}_t \epsilon, \tag{64}$$

$$I_{t-1} = \lambda_t I_0 + \kappa_t I_t + \eta_t I_{res} + \sigma_t \dot{\epsilon}$$
$$= \lambda_t I_0 + \kappa_t (I_0 + \overline{\alpha}_t I_{res} + \overline{\beta}_t \epsilon) + \eta_t I_{res} + \sigma_t \dot{\epsilon} \tag{65}$$
$$= (\lambda_t + \kappa_t) I_0 + (\kappa_t \overline{\alpha}_t + \eta_t) I_{res} + (\kappa_t^2 \overline{\beta}_t^2 + \sigma_t^2)^{\frac{1}{2}} \epsilon.$$

By comparing the coefficients of Eq.(65) and Eq.(63), we can get the following quadratic equation:

$$\begin{cases} \lambda_t + \kappa_t = 1, \\ \kappa_t \overline{\alpha}_t + \eta_t = \overline{\alpha}_{t-1}, \\ \kappa_t^2 \overline{\beta}_t^2 + \sigma_t^2 = \overline{\beta}_{t-1}^2. \end{cases} \tag{66}$$

To complete the derivation, we treat $\sigma_t^2$ as a predefined parameters:

$$\kappa_t = \sqrt{\frac{\overline{\beta}_{t-1}^2 - \sigma_t^2}{\overline{\beta}_t^2}}, \quad \eta_t = \overline{\alpha}_{t-1} - \overline{\alpha}_t \sqrt{\frac{\overline{\beta}_{t-1}^2 - \sigma_t^2}{\overline{\beta}_t^2}}, \quad \lambda_t = 1 - \sqrt{\frac{\overline{\beta}_{t-1}^2 - \sigma_t^2}{\overline{\beta}_t^2}}. \tag{67}$$

Consequently, the inverse process can be formally simplified as:

$$\mu_{t-1} = I_t - \alpha_t I_{res} - (\overline{\beta}_t - \sqrt{\overline{\beta}_{t-1}^2 - \sigma_t^2})\epsilon. \tag{68}$$

Obviously, the mean of $q(I_{t-1}|I_0, I_t, I_{res})$ is manipulated by the variance intensity $\sigma_t$. If we make $\sigma_t^2$ equals that of Eq. (58), then the value of $\mu_{t-1}$ in Eq.(68) equals that of Eq. (59). Besides, if we make $\sigma_t^2$ equals to 0, then the deterministic inverse process can be denoted as:

$$I_{t-1} = I_t - \alpha_t I_{res} - (\overline{\beta}_t - \overline{\beta}_{t-1})\epsilon. \tag{69}$$

## A.9 Training Objective of Diffusion Residual Models

The training objective of DRMs is calculated as follows:

$$\mathcal{L} = \mathbb{E}\left[\left\|I_t - \alpha_t I_{res} - \frac{\beta_t^2}{\overline{\beta}}\epsilon - (I_t - \alpha_t I_{res}^\theta - \frac{\beta_t^2}{\overline{\beta}}\epsilon_\theta)\right\|\right]. \tag{70}$$

Leveraging Eq. 51, then Eq. 70 can be simplified into two equivalent formulas:

$$\mathcal{L} = \mathbb{E}\left[\lambda_{res}\left\|I_{res} - I_{res}^\theta(I_t, I_{in}, t)\right\|\right], \tag{71}$$

$$\mathcal{L} = \mathbb{E}\left[\lambda_\epsilon\left\|\epsilon - \epsilon_\theta(I_t, I_{in}, t)\right\|\right], \tag{72}$$

where the weights, *i.e.*, $\lambda_{res}, \lambda_\epsilon \in \{0, 1\}$, are the balance coefficients.

## A.10 Stochastic Differential Equations of Diffusion Residual Models

Like the derivation from Eq. (42) to Eq. (46), we have:

$$q(I_t|I_0, I_{res}, I_{in}) = \mathcal{N}(I_t; I_0 + \overline{\alpha}_t I_{res}, \overline{\beta}_t^2 \mathbf{I}), \tag{73}$$

$$q(I_{t+\Delta t}|I_0, I_{res}, I_{in}) = \mathcal{N}(I_{t+\Delta t}; I_0 + \overline{\alpha}_{t+\Delta t} I_{res}, \overline{\beta}_{t+\Delta t}^2 \mathbf{I}), \tag{74}$$

$$q(I_{t+\Delta t}|I_t, I_{res}, I_{in}) = \mathcal{N}(I_{t+\Delta t}; (1 + f(t)\Delta t)I_t + h(t)\Delta t I_{res}, g^2(t)\Delta t \mathbf{I}), \tag{75}$$

the marginal distribution $\int q(I_{t+\Delta t}|I_t)q(I_t|I_0)dI_t$ can be computed as:

$$I_{t+\Delta t} = (1 + f(t)\Delta t)(I_0 + \overline{\alpha}_t I_{res} + \overline{\beta}_t \epsilon_1) + h(t)\Delta t I_{res} + g(t)\sqrt{\Delta t}\epsilon_2. \tag{76}$$

Comparing the undetermined coefficients in Eq. 74 with Eq. 76:

$$1 + f(t)\Delta t = 1, \tag{77}$$

$$h(t)\Delta t + (1 + f(t)\Delta t)\overline{\alpha}_t = \overline{\alpha}_{t+\Delta t}, \tag{78}$$

$$(1 + f(t)\Delta t)^2 \overline{\beta}_t^2 + g^2(t)\Delta t = \overline{\beta}_{t+\Delta t}^2, \tag{79}$$

then the above four functions can be solved as follows:

$$f(t) = 0, \quad h(t) = \frac{\overline{\alpha}_{t+\Delta t} - \overline{\alpha}_t}{\Delta t}, \quad g^2(t) = \frac{\overline{\beta}_{t+\Delta t}^2 - \overline{\beta}_t^2}{\Delta t}. \tag{80}$$

Let $\Delta t \to 0$, then:

$$f(t) = 0, \quad h(t) = \frac{d\overline{\alpha}_t}{dt}, \quad g^2(t) = \frac{d\overline{\beta}_t^2}{dt} = 2\overline{\beta}_t \frac{d\overline{\beta}_t}{dt}. \tag{81}$$

## A.11 Forward Process of Diffusion Generalist Model

Based on the formulation of DGMs, diffusion generalist models additionally incorporate the degraded image manipulated by $\delta_t$ in the forward process:

$$q(I_t|I_{t-1}, I_{res}, I_{in}) = \mathcal{N}(I_t; I_{t-1} + \alpha_t I_{res} - \delta_t I_{in}, \beta_t^2 \boldsymbol{I}), \tag{82}$$

$$I_t = I_{t-1} + \alpha_t I_{res} + \beta_t \epsilon_t - \delta_t I_{in}, \tag{83}$$

Leveraging the property of Markov Chain and reparameterization, we have:

$$
\begin{aligned}
I_t &= I_{t-1} + \alpha_t I_{res} + \beta_t \epsilon_t - \delta_t I_{in} \\
&= \left[ I_{t-2} + \alpha_{t-1} I_{res} + \beta_{t-1}\epsilon_{t-1} - \delta_{t-1} I_{in} \right] + \alpha_t I_{res} + \beta_t \epsilon_t - \delta_t I_{in} \\
&= I_{t-2} + \left[ \alpha_{t-1} + \alpha_t \right] I_{res} + \sqrt{\left[ \beta_{t-1}^2 + \beta_t^2 \right]}\epsilon + \left[ \delta_{t-1} + \delta_t \right] I_{in} \\
&= \cdots \\
&= I_0 + \overline{\alpha}_t I_{res} + \overline{\beta}_t \epsilon - \overline{\delta}_t I_{in} \\
&= (\overline{\alpha}_t - 1)I_{res} + \overline{\beta}_t \epsilon + (1 - \overline{\delta}_t)I_{in},
\end{aligned}
\tag{84}
$$

where $\overline{\alpha}_t = \sum_{i=1}^t \alpha_i, \overline{\beta}_t = \sqrt{\sum_{i=1}^t \beta_i^2}$ and $\overline{\delta}_t = \sum_{i=1}^t \delta_i$. If $\overline{\alpha}_t \to 1$, clean images are diffused as:

$$I_t = (1 - \overline{\delta}_t)I_{in} + \overline{\beta}_t \epsilon. \tag{85}$$

Thus, we obtain an expression for degraded image representation, in which $\overline{\delta}_t$ plays a role in controlling the commonality. If $\overline{\delta}_t \to 0$, Eq. (85) degrades to the pattern of diffusion residual models; whereas if $\overline{\delta}_t \to 1$, all representations are unified into a pure Gaussian distribution.

## A.12 Perturbed Generation Process of Diffusion Generalist Models

By repeating the aforementioned derivation process, akin to Eq. (52)-Eq. (57), we have:

$$q(I_t|I_{t-1}, I_{res}, I_{in}) = \mathcal{N}(I_t; I_{t-1} + \alpha_t I_{res} - \delta_t I_{in}, \beta_t^2 \boldsymbol{I}), \tag{86}$$

$$q(I_{t-1}|I_0, I_{res}, I_{in}) = \mathcal{N}(I_{t-1}; I_0 + \overline{\alpha}_{t-1} I_{res} - \overline{\delta}_{t-1} I_{in}, \overline{\beta}_{t-1}^2 \boldsymbol{I}), \tag{87}$$

$$q(I_t|I_0, I_{res}, I_{in}) = \mathcal{N}(I_t; I_0 + \overline{\alpha}_t I_{res} - \overline{\delta}_t I_{in}, \overline{\beta}_t^2 \boldsymbol{I}). \tag{88}$$

$$
q(I_{t-1}|I_t, I_{res}, I_{in}) = \exp\left\{ \log q(I_t|I_{t-1}, I_{res}, I_{in}) + \log q(I_{t-1}|I_{res}, I_{in}) - \log q(I_t|I_{res}, I_{in}) \right\}
$$

$$
\propto \exp\left\{ -\frac{\left(I_t - (I_{t-1} + \alpha_t I_{res} - \delta_t I_{in})\right)^2}{2\beta_t^2} - \frac{\left(I_{t-1} - (\overline{\alpha}_{t-1} - 1)I_{res} - (1 - \overline{\delta}_{t-1})I_{in}\right)^2}{2\overline{\beta}_{t-1}^2} \right.
$$

$$
\left. + \frac{\left(I_t - (\overline{\alpha}_t - 1)I_{res} - (1 - \overline{\delta}_t)I_{in}\right)^2}{2\overline{\beta}_t^2} \right\}
$$

$$
\propto \exp\left\{ -\frac{1}{2}\left( (\frac{1}{\beta_t^2} + \frac{1}{\overline{\beta}_{t-1}^2})I_{t-1}^2 - 2I_{t-1}\big(\frac{I_t - \alpha_t I_{res} + \delta_t I_{in}}{\beta_t^2} + \frac{(\overline{\alpha}_{t-1} - 1)I_{res} + (1 - \overline{\delta}_{t-1})I_{in}}{\overline{\beta}_{t-1}^2}\big) + C(I_t, I_0, I_{res}) \right) \right\}.
\tag{89}
$$

Comparing the coefficients of each term, we can obtain:

$$\Sigma_{t-1}^2(I_t, I_{in}, I_{res}) = \frac{\beta_t^2 \overline{\beta}_{t-1}^2}{\beta_t^2 + \overline{\beta}_{t-1}^2} = \frac{\beta_t^2 \overline{\beta}_{t-1}^2}{\overline{\beta}_t^2}, \tag{90}$$

$$
\begin{aligned}
\mu_{t-1}(I_t, I_{in}, I_{res}) &= \big(\frac{I_t - \alpha_t I_{res} + \delta I_{in}}{\beta_t^2} + \frac{I_0 + \overline{\alpha}_{t-1} I_{res} - \overline{\delta}_{t-1} I_{in}}{\overline{\beta}_{t-1}^2}\big)\frac{\beta_t^2 \overline{\beta}_{t-1}^2}{\overline{\beta}_t^2} \\
&= \frac{1}{\overline{\beta}_t^2}\big(\overline{\beta}_{t-1}^2 I_t - \alpha_t \overline{\beta}_{t-1}^2 I_{res} + \delta_t \overline{\beta}_{t-1}^2 I_{in} + \beta_t^2 I_0 + \overline{\alpha}_{t-1}\beta_t^2 I_{res} - \overline{\delta}_{t-1}\beta_t^2 I_{in}\big) \\
&= \frac{\overline{\beta}_{t-1}^2}{\overline{\beta}_t^2} I_t + \frac{\overline{\alpha}_{t-1}\beta_t^2 - \alpha_t \overline{\beta}_{t-1}^2}{\overline{\beta}_t^2} I_{res} + \frac{\delta_t \overline{\beta}_{t-1}^2 - \overline{\delta}_{t-1}\beta_t^2}{\overline{\beta}_t^2} I_{in} + \frac{\beta_t^2}{\overline{\beta}_t^2}(I_t - \overline{\alpha}_t I_{res} - \overline{\beta}_t \epsilon + \overline{\delta}_t I_{in}) \\
&= I_t - \alpha_t I_{res} + \delta_t I_{in} - \frac{\beta_t^2}{\overline{\beta}_t}\epsilon,
\end{aligned}
\tag{91}
$$

the inverse process can be considered as sampling from the following distribution:

$$q(I_{t-1}|I_t, I_0, I_{res}, I_{in}) \sim \mathcal{N}(I_{t-1}|I_t - \alpha_t I_{res} + \delta_t I_{in} - \frac{\beta_t^2}{\overline{\beta}_t}\epsilon; \frac{\beta_t \overline{\beta}_{t-1}}{\overline{\beta}_t}\boldsymbol{I}). \tag{92}$$

## A.13 Deterministic Implicit Sampling of Diffusion Generalist Models

A common forward process can be determined as follows:

$$q(I_{t-1}|I_0, I_{res}, I_{in}) = \mathcal{N}(I_{t-1}; I_0 + \overline{\alpha}_{t-1}I_{res} - \overline{\delta}_{t-1}I_{in}, \overline{\beta}_{t-1}^2 \boldsymbol{I}), \tag{93}$$

$$q(I_t|I_0, I_{res}, I_{in}) = \mathcal{N}(I_t; I_0 + \overline{\alpha}_t I_{res} - \overline{\delta}_t I_{in}, \overline{\beta}_t^2 \boldsymbol{I}). \tag{94}$$

For simplicity, we still define the formula of the conditional distribution $q(I_{t-1}|I_0, I_t)$ as:

$$q(I_{t-1}|I_t, I_0, I_{res}, I_{in}) = \mathcal{N}(I_{t-1}; \kappa_t I_t + \eta_t I_{res} + \gamma_t I_0 + \zeta_t I_{in}, \sigma_t^2 \boldsymbol{I}), \tag{95}$$

then we can reformulate the probability formulas into algebraic expressions:

$$I_{t-1} = I_0 + \overline{\alpha}_{t-1}I_{res} - \overline{\delta}_{t-1}I_{in} + \overline{\beta}_{t-1}\epsilon_1, \tag{96}$$

$$I_t = I_0 + \overline{\alpha}_t I_{res} - \overline{\delta}_t I_{in} + \overline{\beta}_t \epsilon_2, \tag{97}$$

$$
\begin{aligned}
I_{t-1} &= \kappa_t I_t + \eta_t I_{res} + \gamma_t I_0 + \zeta_t I_{in} + \sigma_t \epsilon_3 \\
&= \kappa_t \left( I_0 + \overline{\alpha}_t I_{res} - \overline{\delta}_t I_{in} + \overline{\beta}_t \epsilon_2 \right) + \eta_t I_{res} + \gamma_t I_0 + \zeta_t I_{in} + \sigma_t \epsilon_3 \\
&= (\kappa_t + \gamma_t)I_0 + (\kappa_t \overline{\alpha}_t + \eta_t)I_{res} - (\kappa_t \overline{\delta}_t - \zeta_t)I_{in} + (\kappa_t^2 \overline{\beta}_t^2 + \sigma_t^2)^{\frac{1}{2}}\epsilon.
\end{aligned}
\tag{98}
$$

By comparing the coefficients of Eq. (96) and Eq.(98), we have:

$$\kappa_t + \gamma_t = 1, \tag{99}$$

$$\kappa_t \overline{\alpha}_t + \eta_t = \overline{\alpha}_{t-1}, \tag{100}$$

$$\kappa_t \overline{\delta}_t - \zeta_t = \overline{\delta}_{t-1}, \tag{101}$$

$$\kappa_t^2 \overline{\beta}_t^2 + \sigma_t^2 = \overline{\beta}_{t-1}^2. \tag{102}$$

To complete the derivation, we treat $\sigma_t^2$ as a known variable:

$$\kappa_t = \sqrt{\frac{\overline{\beta}_{t-1}^2 - \sigma_t^2}{\overline{\beta}_t^2}}, \eta_t = \overline{\alpha}_{t-1} - \overline{\alpha}_t \sqrt{\frac{\overline{\beta}_{t-1}^2 - \sigma_t^2}{\overline{\beta}_t^2}}, \tag{103}$$

$$\gamma_t = 1 - \sqrt{\frac{\overline{\beta}_{t-1}^2 - \sigma_t^2}{\overline{\beta}_t^2}}, \zeta_t = \overline{\delta}_t \sqrt{\frac{\overline{\beta}_{t-1}^2 - \sigma_t^2}{\overline{\beta}_t^2}} - \overline{\delta}_{t-1}, \tag{104}$$

The inverse process in Eq.(98) can be reformulated as:

$$
\begin{aligned}
I_{t-1} = & \left( \sqrt{\frac{\overline{\beta}_{t-1}^2 - \sigma_t^2}{\overline{\beta}_t^2}} \right) I_t + \left( \overline{\alpha}_{t-1} - \overline{\alpha}_t \sqrt{\frac{\overline{\beta}_{t-1}^2 - \sigma_t^2}{\overline{\beta}_t^2}} \right) I_{res} \\
& + \left( 1 - \sqrt{\frac{\overline{\beta}_{t-1}^2 - \sigma_t^2}{\overline{\beta}_t^2}} \right) I_0 + \left( \overline{\delta}_t \sqrt{\frac{\overline{\beta}_{t-1}^2 - \sigma_t^2}{\overline{\beta}_t^2}} - \overline{\delta}_{t-1} \right) I_{in} + \sigma_t \epsilon.
\end{aligned}
\tag{105}
$$

Furthermore, Eq. (105) can be simplified and decomposed into mean $\mu_{t-1}(I_t, I_0, I_{res})$ and variance $\Sigma_{t-1}^2(I_t, I_0, I_{res})$, which can be denoted as:

$$\mu_{t-1}(I_t, I_0, I_{res}) = I_t - (\overline{\alpha}_t - \overline{\alpha}_{t-1})I_{res} + (\overline{\delta}_t - \overline{\delta}_{t-1})I_{in} - \left( \overline{\beta}_t - \sqrt{\overline{\beta}_{t-1}^2 - \sigma_t^2} \right)\epsilon. \tag{106}$$

Similarly, we set the variance to zero for deterministic inverse inference:

$$I_{t-1} = I_t - (\overline{\alpha}_t - \overline{\alpha}_{t-1})I_{res} + (\overline{\delta}_t - \overline{\delta}_{t-1})I_{in} - (\overline{\beta}_t - \overline{\beta}_{t-1})\epsilon. \tag{107}$$

## A.14 Training Objective of Diffusion Generalist Models

The training objective of DGMs is calculated as follows:

$$
\begin{aligned}
\mathcal{L} &= \mathbb{E} \left[ \left\| I_t - \alpha_t I_{res} + \delta_t I_{in} - \frac{\beta_t^2}{\overline{\beta}} \epsilon - (I_t - \alpha_t I_{res}^\theta + \delta_t I_{in} - \frac{\beta_t^2}{\overline{\beta}} \epsilon_\theta) \right\|_1 \right] \\
&= \mathbb{E} \left[ \left\| \alpha_t(I_{res}^\theta - I_{res}) + \frac{\beta_t^2}{\overline{\beta}} (\epsilon_\theta - \epsilon) \right\|_1 \right].
\end{aligned}
\tag{108}
$$

Leveraging the Eq. 84, then Eq. 108 can be simplified into the following formula:

$$\mathcal{L} = \mathbb{E}\big[\big\|\alpha_t(I_{res} - I_{res}^\theta(I_t, I_{in}, t)) + \frac{(1-\overline{\alpha}_t)\beta_t^2}{\overline{\beta}_t^2}(I_{res} - I_{res}^\theta(I_t, I_{in}, t))\big\|_1\big] \tag{109}$$

$$= \mathbb{E}\big[\big\|\lambda_{res}(I_{res} - I_{res}^\theta(I_t, I_{in}, t))\big\|_1\big],$$

where the coefficient $\lambda_{res}$ is a constant. The detailed training algorithm is summarized in Alg. 1.

---

**Algorithm 1:** Training of diffusion generalist model.

---

**Input:** $I_{in}, I_0, I_{res}^\theta = I_{in} - I_0, \{\overline{\alpha}_{t_i}\}_{i=0}^M, \{\overline{\beta}_{t_i}\}_{i=0}^M, \{\overline{\delta}_{t_i}\}_{i=0}^M, \{t_i\}_{i=0}^M.$

1 **repeat**
2    $I_0 \sim q(I_0)$
3    $i \sim Uniform(1, ..., T)$
4    $\epsilon \sim \mathcal{N}(0, \boldsymbol{I})$
5    $I_t = I_0 + \overline{\alpha}_{t_i}I_{res} + \overline{\beta}_{t_i}\epsilon - \overline{\delta}_{t_i}I_{in}$
6    Take the gradient descent step on
7    $\nabla_\theta\|I_{res} - I_{res}^\theta(I_{t_i}, I_{in}, t_i)\|_1$
8 **until**;
   **Output:** $I_{t_0}$.

---

## A.15 Stochastic Differential Equations of Diffusion Generalist Models

The forward process is not only constrained by time-related variables $I_t$ but also limited by $I_{res}$ and $I_{in}$. Therefore, the stochastic differential equations and the inverse equations can be denoted as:

$$dI_t = f(I_t, t)dt + h(I_{res}, t)dt + l(I_{in}, t)dt + g(t)d\omega_t, \quad I_0 \sim q_0(I_0), \tag{110}$$

$$\frac{dI_t}{dt} = [f(I_t, t) + h(I_{res}, t) + l(I_{in}, t) - g^2(t)\nabla \log q_t(I_t)] + g(t)d\overline{\omega}_t, \quad I_t \sim q_T(I_t), \tag{111}$$

where $f(I_t, t) = f(t)I_t, h(I_{res}, t) = h(t)I_{res}$, and $l(I_{in}, t) = l(t)I_{in}$. According to the definitions of Eq. (84), we can derive:

$$q(I_t|I_0, I_{res}, I_{in}) = \mathcal{N}(I_t; I_0 + \overline{\alpha}_t I_{res} - \overline{\delta}_t I_{in}, \overline{\beta}_t^2 \boldsymbol{I}), \tag{112}$$

$$q(I_{t+\Delta t}|I_0, I_{res}, I_{in}) = \mathcal{N}(I_{t+\Delta t}; I_0 + \overline{\alpha}_{t+\Delta t}I_{res} - \overline{\delta}_{t+\Delta t}I_{in}, \overline{\beta}_{t+\Delta t}^2\boldsymbol{I}), \tag{113}$$

we assume that the general form of the one-step forward process is:

$$q(I_{t+\Delta t}|I_t, I_{res}, I_{in}) = \mathcal{N}((1 + f(t)\Delta t)I_t + h(t)\Delta t I_{res} + l(t)\Delta t I_{in}, g^2(t)\Delta t \boldsymbol{I}), \tag{114}$$

similarly, we compute its marginal distribution $\int q(I_{t+\Delta t}|I_t)q(I_t|I_0)dI_t$:

$$I_{t+\Delta t} = (1 + f(t)\Delta t)(I_0 + \overline{\alpha}_t I_{res} - \overline{\delta}_t I_{in} + \overline{\beta}_t\epsilon_1) + h(t)\Delta t I_{res} + l(t)\Delta t I_{in} + g(t)\sqrt{\Delta t}\epsilon_2, \tag{115}$$

comparing the undetermined coefficients in Eq. 113 with Eq. 115, we have:

$$1 + f(t)\Delta t = 1, \tag{116}$$

$$h(t)\Delta t + (1 + f(t)\Delta t)\overline{\alpha}_t = \overline{\alpha}_{t+\Delta t}, \tag{117}$$

$$l(t)\Delta t - (1 + f(t)\Delta t)\overline{\delta}_t = -\overline{\delta}_{t+\Delta t}, \tag{118}$$

$$(1 + f(t)\Delta t)^2\overline{\beta}_t^2 + g^2(t)\Delta t = \overline{\beta}_{t+\Delta t}^2, \tag{119}$$

then the above four functions can be solved as follows:

$$f(t) = 0, h(t) = \frac{\overline{\alpha}_{t+\Delta t} - \overline{\alpha}_t}{\Delta t}, l(t) = \frac{\overline{\delta}_t - \overline{\delta}_{t+\Delta t}}{\Delta t}, g^2(t) = \frac{\overline{\beta}_{t+\Delta t}^2 - \overline{\beta}_t^2}{\Delta t}, \tag{120}$$

let $\Delta t \to 0$, then we get:

$$f(t) = 0, h(t) = \frac{d\overline{\alpha}_t}{dt}, l(t) = -\frac{d\overline{\delta}_t}{dt}, g^2(t) = \frac{d\overline{\beta}_t^2}{dt}. \tag{121}$$

# B Customized Solver for Diffusion Generalist Model

## B.1 Reverse ODEs Formulation of Diffusion Generalist Model

For generality, we follow the notations in Section 3. The inherent parameters are time continuous satisfying $\overline{\alpha}(t) = \overline{\alpha}_t, \overline{\delta}(t) = \overline{\delta}_t, \overline{\beta}(t) = \overline{\beta}_t$. The inverse ordinary differential equations can be concretely formulated as [51]:

$$\frac{dI_t}{dt} = \frac{d\overline{\alpha}_t}{dt}I_{res} - \frac{d\overline{\delta}_t}{dt}I_{in} - \frac{1}{2}\frac{d\overline{\beta}_t^2}{dt}\nabla \log q_t(I_t). \tag{122}$$

Given an initial value $I_s$ at time $s > 0$, the solution $I_t$ at each time $s < t < T$ can be computed as:

$$I_t = I_s + \int_s^t \frac{d\overline{\alpha}_\tau}{d\tau} I_{res} d\tau - \int_s^t \frac{d\overline{\delta}_\tau}{d\tau} I_{in} d\tau - \int_s^t \frac{1}{2} \frac{d\overline{\beta}_\tau^2}{d\tau} \nabla \log q_\tau(I_\tau) d\tau. \tag{123}$$

According to ***Tweedie's Formula*** in Lemma 1: Given a Gaussian variable $z \sim \mathcal{N}(z; \mu_z, \Sigma_z^2)$, the estimated parameters $\mu_z$ can be computed as follows:

$$\mu_z = z + \Sigma_z^2 \nabla \log P(z). \tag{124}$$

For forward process in Eq.(94), there exists:

$$I_0 + \overline{\alpha}_t I_{res} - \overline{\delta}_t I_{in} = I_t + \overline{\beta}_t^2 \nabla \log P(I_t) = I_t - \overline{\beta}_t \epsilon, \tag{125}$$

$$\nabla \log q_t(I_t) = -\frac{\epsilon}{\overline{\beta}_t}. \tag{126}$$

Hence, substituting the relationships between score functions and noise into Eq.(123), we have:

$$I_t = I_s + \int_s^t \frac{d\overline{\alpha}_\tau}{d\tau} I_{res} d\tau - \int_s^t \frac{d\overline{\delta}_\tau}{d\tau} I_{in} d\tau + \int_s^t \frac{d\overline{\beta}_\tau}{d\tau} \epsilon d\tau, \tag{127}$$

Furthermore, we have:

$$I_t = I_s + \int_s^t \frac{d\overline{\alpha}_\tau}{d\tau} I_{res}^\theta(I_t, I_{in}, \tau) d\tau - I_{in} \int_s^t \frac{d\overline{\delta}_\tau}{d\tau} d\tau + \int_s^t \frac{d\overline{\beta}_\tau}{d\tau} \epsilon_\theta(I_t, I_{in}, \tau) d\tau, \tag{128}$$

where $I_{res} = I_{res}^\theta(I_t, I_{in}, t)$ and $\epsilon_\theta(I_t, I_{in}, t)$ are time-relevant, $I_{in}$ is input degraded image and can be considered as constant. Since the above inherent parameters are time continuous and monotonic for linear noise schedule, there must exist inverse functions satisfying $t = t_{\overline{\alpha}}(\overline{\alpha}(t)) = t_{\overline{\beta}}(\overline{\beta}(t)) = t_{\overline{\delta}}(\overline{\delta}(t))$. We denote $\widehat{I}_{\overline{\alpha}} := I_{t_{\overline{\alpha}}(\overline{\alpha})}, I_{res}^\theta(I_\tau, I_{in}, \tau) := \widehat{I}_{res}^\theta(\widehat{I}_{\overline{\alpha}}, I_{in}, \overline{\alpha})$ and $\widehat{I}_{\overline{\beta}} := I_{t_{\overline{\beta}}(\overline{\beta})}, \epsilon_\theta(I_\tau, I_{in}, \tau) := \widehat{\epsilon}_\theta(\widehat{I}_{\overline{\beta}}, I_{in}, \overline{\beta})$, and then we have:

$$I_t = I_s - (\overline{\delta}_t - \overline{\delta}_s) I_{in} + \int_{\overline{\alpha}_s}^{\overline{\alpha}_t} I_{res}^\theta(\widehat{I}_{\overline{\alpha}}, I_{in}, \overline{\alpha}) d\overline{\alpha} + \int_{\overline{\beta}_s}^{\overline{\beta}_t} \epsilon_\theta(\widehat{I}_{\overline{\beta}}, I_{in}, \overline{\beta}) d\overline{\beta}. \tag{129}$$

With the simplicity of the continuous formula of Eq. (129), we can propose high-order solvers that correspond to the ODEs. Specifically, given an initial point $I_T$ at time $T$ and total $M + 1$ time steps $\{t_i\}_{i=0}^M$ decreasing from $t_0 = T$ to $t_M = 0$. Let $\{I_{t_i}\}_{i=0}^M$ be the sequence iteratively computed at time steps $\{t_i\}_{i=0}^M$ using the presented solvers. Starting with the previous solution $I_{t_{i-1}}$ at time $t_{i-1}$, the exact solution $I_{t_{i-1} \to t_i}$ at target time $t_i$ is:

$$I_{t_{i-1} \to t_i} = I_{t_{i-1}} - (\overline{\delta}_{t_i} - \overline{\delta}_{t_{i-1}}) I_{in} + \int_{\overline{\alpha}_{t_{i-1}}}^{\overline{\alpha}_{t_i}} \widehat{I}_{res}^\theta(\widehat{I}_{\overline{\alpha}}, I_{in}, \overline{\alpha}) d\overline{\alpha} + \int_{\overline{\beta}_{t_{i-1}}}^{\overline{\beta}_{t_i}} \widehat{\epsilon}_\theta(\widehat{I}_{\overline{\beta}}, I_{in}, \overline{\beta}) d\overline{\beta}. \tag{130}$$

We need to approximate the integral of both $\widehat{I}_{res}^\theta(\widehat{I}_{\overline{\alpha}}, I_{in}, \overline{\alpha})$ from $\overline{\alpha}_{t_{i-1}}$ to $\overline{\alpha}_{t_i}$ and $\widehat{\epsilon}_\theta(\widehat{I}_{\overline{\beta}}, I_{in}, \overline{\beta})$ from $\overline{\beta}_{t_{i-1}}$ to $\overline{\beta}_{t_i}$. We utilize the Taylor expansion to make $k$ orders approximation:

$$\widehat{I}_{res}^\theta(\widehat{I}_{\overline{\alpha}}, I_{in}, \overline{\alpha}) = \sum_{n=0}^{k-1} \frac{(\overline{\alpha} - \overline{\alpha}_{t_{i-1}})^n}{n!} \widehat{I}_{res}^\theta(\widehat{I}_{\overline{\alpha}_{t_{i-1}}}, I_{in}, \overline{\alpha}_{t_{i-1}}) + \mathcal{O}((\overline{\alpha} - \overline{\alpha}_{t_{i-1}})^k), \tag{131}$$

$$\widehat{\epsilon}_\theta(\widehat{I}_{\overline{\beta}}, I_{in}, \overline{\beta}) = \sum_{n=0}^{k-1} \frac{(\overline{\beta} - \overline{\beta}_{t_{i-1}})^n}{n!} \widehat{\epsilon}_\theta(\widehat{I}_{\overline{\beta}_{t_{i-1}}}, I_{in}, \overline{\beta}_{t_{i-1}}) + \mathcal{O}((\overline{\beta} - \overline{\beta}_{t_{i-1}})^k), \tag{132}$$

substituting the above Taylor expansion into Eq. (130), we can obtain:

$$\begin{aligned} I_{t_{i-1} \to t_i} = I_{t_{i-1}} - (\overline{\delta}_{t_i} - \overline{\delta}_{t_{i-1}}) I_{in} + \sum_{n=0}^{k-1} \widehat{I}_{res}^\theta(\widehat{I}_{\overline{\alpha}_{t_{i-1}}}, I_{in}, \overline{\alpha}_{t_{i-1}}) \int_{\overline{\alpha}_{t_{i-1}}}^{\overline{\alpha}_{t_i}} \frac{(\overline{\alpha} - \overline{\alpha}_{t_{i-1}})^n}{n!} d\overline{\alpha} \\ + \sum_{n=0}^{k-1} \widehat{\epsilon}_\theta(\widehat{I}_{\overline{\beta}_{t_{i-1}}}, I_{in}, \overline{\beta}_{t_{i-1}}) \int_{\overline{\beta}_{t_{i-1}}}^{\overline{\beta}_{t_i}} \frac{(\overline{\beta} - \overline{\beta}_{t_{i-1}})^n}{n!} d\overline{\beta} + \mathcal{O}((\overline{\alpha} - \overline{\alpha}_{t_{i-1}})^{k+1}) + \mathcal{O}((\overline{\beta} - \overline{\beta}_{t_{i-1}})^{k+1}). \end{aligned} \tag{133}$$

since the integral is linear, its analytical expression can be easily written as:

$$\begin{aligned} I_{t_{i-1} \to t_i} = I_{t_{i-1}} - (\overline{\delta}_{t_i} - \overline{\delta}_{t_{i-1}}) I_{in} + \sum_{n=0}^{k-1} \frac{(\overline{\alpha}_{t_i} - \overline{\alpha}_{t_{i-1}})^{n+1}}{(n+1)!} \widehat{I}_{res}^{\theta(n)}(\widehat{I}_{t_{i-1}}, I_{in}, t_{i-1}) \\ + \sum_{n=0}^{k-1} \frac{(\overline{\beta}_{t_i} - \overline{\beta}_{t_{i-1}})^{n+1}}{(n+1)!} \widehat{\epsilon}_\theta^{(n)}(\widehat{I}_{t_{i-1}}, I_{in}, t_{i-1}) + \mathcal{O}((\overline{\alpha} - \overline{\alpha}_{t_{i-1}})^{k+1}) + \mathcal{O}((\overline{\beta} - \overline{\beta}_{t_{i-1}})^{k+1}). \end{aligned} \tag{134}$$

Therefore, to approximate $I_{t_{i-1}} \to I_{t_i}$, we only need to approximate the $n$-th order total derivatives $I_{res}^{\theta(n)}(\widehat{I}_{t_{i-1}}, I_{in}, t_{i-1})$ and $\epsilon_\theta^{(n)}(\widehat{I}_{t_{i-1}}, I_{in}, t_{i-1})$ for $n \leq k-1$. In the case of $k = 1$, Eq. (134) becomes:

$$I_{t_i} = I_{t_{i-1}} - (\overline{\delta}_{t_i} - \overline{\delta}_{t_{i-1}})I_{in} + (\overline{\alpha}_{t_i} - \overline{\alpha}_{t_{i-1}})I_{res}^\theta(I_{t_{i-1}}, I_{in}, t_{i-1}) + (\overline{\beta}_{t_i} - \overline{\beta}_{t_{i-1}})\epsilon_\theta(I_{t_{i-1}}, I_{in}, t_{i-1}). \quad (135)$$

By dropping the high-order error term $\mathcal{O}(\cdots)$, we can obtain the first-order approximation for $I_{t_{i-1}} \to I_{t_i}$. As $k = 1$, we call this Diffusion Generalist Solver-1, and the detailed algorithm is as follows.

**First-order Diffusion Generalist Solver** Given an initial value $I_T$ and $M + 1$ time steps $\{t_i\}_{i=0}^M$ decreasing from $t_0 = T$ to $t_M = 0$. Starting with $\widehat{I}_{t_0} = I_T$, the sequence $\{\widehat{I}_{t_i}\}_{i=0}^M$ is computed iteratively as follows:

$$I_{t_i} = I_{t_{i-1}} - (\overline{\delta}_{t_i} - \overline{\delta}_{t_{i-1}})I_{in} + (\overline{\alpha}_{t_i} - \overline{\alpha}_{t_{i-1}})I_{res}^\theta(I_{t_{i-1}}, I_{in}, t_{i-1}) + (\overline{\beta}_{t_i} - \overline{\beta}_{t_{i-1}})\epsilon_\theta(I_{t_{i-1}}, I_{in}, t_{i-1}). \quad (136)$$

Apparently, it is consistent with Eq. 107, which is a special case of first-order solver within our methods by omitting the discrete error. The overall algorithm for first-order diffusion generalist solver is presented in Alg. 2.

---

**Algorithm 2:** First-order Diffusion Generalist Solver.

**Input:** $I_{in}, I_{res}^\theta, \epsilon_\theta, \{\overline{\alpha}_{t_i}\}, \{\overline{\beta}_{t_i}\}, \{\overline{\delta}_{t_i}\}, \{t_i\}_{i=0}^M$.

1   $I_{t_0} = I_T \sim \mathcal{N}(0, \overline{\beta}_{t_0}\boldsymbol{I})$
2   **for** $i = 1$ to $M$ **do**
3     $\widehat{I}_{res} = I_{res}^\theta(I_{t_{i-1}}, I_{in}, t_{i-1})$
4     $\widehat{\epsilon} = (I_{t_{i-1}} - (\overline{\alpha}_{t_{i-1}} - 1)\widehat{I}_{res} - (1 - \overline{\delta}_{t_{i-1}})I_{in})/\overline{\beta}_{t_{i-1}}$
5     $\widehat{I}_0 = I_{t_{i-1}} - \overline{\alpha}_{t_{i-1}}\widehat{I}_{res} - \overline{\beta}_{t_{i-1}}\widehat{\epsilon} + \overline{\delta}_{t_{i-1}}I_{in}$
6     $I_{t_i} = I_{t_{i-1}} - (\overline{\delta}_{t_i} - \overline{\delta}_{t_{i-1}})I_{in} + (\overline{\alpha}_{t_i} - \overline{\alpha}_{t_{i-1}})\widehat{I}_{res} + (\overline{\beta}_{t_i} - \overline{\beta}_{t_{i-1}})\widehat{\epsilon}$

**Output:** $I_{t_M}$.

---

For $k = 2$, we need an auxiliary time variable $t_u = rt_i + (1 - r)t_{i-1}$, where $r$ is the controllable variable measuring the intermediate points in the range of $[t_i, t_{i-1}]$. Then Eq. (133) becomes:

$$\begin{aligned} I_{t_{i-1} \to t_i} = {} & I_{t_{i-1}} - (\overline{\delta}_{t_i} - \overline{\delta}_{t_{i-1}})I_{in} + (\overline{\alpha}_{t_i} - \overline{\alpha}_{t_{i-1}})I_{res}^\theta(\widehat{I}_{t_{i-1}}, I_{in}, t_{i-1}) + (\overline{\beta}_{t_i} - \overline{\beta}_{t_{i-1}})\epsilon_\theta(\widehat{I}_{t_{i-1}}, I_{in}, t_{i-1}) \\ & + \frac{1}{2}(\overline{\alpha}_{t_i} - \overline{\alpha}_{t_{i-1}})^2 I_{res}^{\theta(1)}(\widehat{I}_{t_{i-1}}, I_{in}, t_{i-1}) + \frac{1}{2}(\overline{\beta}_{t_i} - \overline{\beta}_{t_{i-1}})^2 \epsilon_\theta^{(1)}(\widehat{I}_{t_{i-1}}, I_{in}, t_{i-1}), \end{aligned} \quad (137)$$

leveraging the finite-difference methods [52], we can approximate the derivative part:

$$I_{res}^{\theta(1)}(\widehat{I}_{t_{i-1}}, I_{in}, t_{i-1}) \approx \frac{I_{res}^\theta(\widehat{I}_{\overline{\alpha}_u}, I_{in}, \overline{\alpha}_u) - I_{res}^\theta(\widehat{I}_{\overline{\alpha}_{i-1}}, I_{in}, \overline{\alpha}_{i-1})}{\overline{\alpha}_{t_u} - \overline{\alpha}_{t_{i-1}}} = \frac{1}{r} \frac{I_{res}^\theta(\widehat{I}_{t_u}, I_{in}, t_u) - I_{res}^\theta(\widehat{I}_{t_{i-1}}, I_{in}, t_{i-1})}{\overline{\alpha}_{t_i} - \overline{\alpha}_{t_{i-1}}}, \quad (138)$$

$$\epsilon_\theta^{(1)}(\widehat{I}_{t_{i-1}}, I_{in}, t_{i-1}) \approx \frac{\epsilon_\theta(\widehat{I}_{\overline{\beta}_u}, I_{in}, \overline{\beta}_u) - \epsilon_\theta(\widehat{I}_{\overline{\beta}_{i-1}}, I_{in}, \overline{\beta}_{i-1})}{\overline{\beta}_{t_u} - \overline{\beta}_{t_{i-1}}} = \frac{1}{r} \frac{\epsilon_\theta(\widehat{I}_{t_u}, I_{in}, t_u) - \epsilon_\theta(\widehat{I}_{t_{i-1}}, I_{in}, t_{i-1})}{\overline{\beta}_{t_i} - \overline{\beta}_{t_{i-1}}}. \quad (139)$$

Therefore, the simplified result of Eq. (137) is:

$$\begin{aligned} I_{t_i} = {} & I_{t_{i-1}} - (\overline{\delta}_{t_i} - \overline{\delta}_{t_{i-1}})I_{in} + (\overline{\alpha}_{t_i} - \overline{\alpha}_{t_{i-1}})I_{res}^\theta(\widehat{I}_{t_{i-1}}, I_{in}, t_{i-1}) + (\overline{\beta}_{t_i} - \overline{\beta}_{t_{i-1}})\epsilon_\theta(\widehat{I}_{t_{i-1}}, I_{in}, t_{i-1}) \\ & + \frac{1}{2r}(\overline{\alpha}_{t_i} - \overline{\alpha}_{t_{i-1}})(I_{res}^\theta(\widehat{I}_{t_u}, I_{in}, t_u) - I_{res}^\theta(\widehat{I}_{t_{i-1}}, I_{in}, t_{i-1})) \\ & + \frac{1}{2r}(\overline{\beta}_{t_i} - \overline{\beta}_{t_{i-1}})(\epsilon_\theta(\widehat{I}_{t_u}, I_{in}, t_u) - \epsilon_\theta(\widehat{I}_{t_{i-1}}, I_{in}, t_{i-1})). \end{aligned} \quad (140)$$

**Second-order Diffusion Generalist Solver** Given an initial value $I_T$ and $M + 1$ time steps $\{t_i\}_{i=0}^M$ decreasing from $t_0 = T$ to $t_M = 0$, starting with $\widehat{I}_{t=0} = I_T$, the sequence $\{\widehat{I}_{t_i}\}_{i=0}^M$ is computed iteratively as follows:

$$\begin{aligned} I_{t_i} = {} & I_{t_{i-1}} - (\overline{\delta}_{t_i} - \overline{\delta}_{t_{i-1}})I_{in} + (\overline{\alpha}_{t_i} - \overline{\alpha}_{t_{i-1}})I_{res}^\theta(\widehat{I}_{t_{i-1}}, I_{in}, t_{i-1}) + (\overline{\beta}_{t_i} - \overline{\beta}_{t_{i-1}})\epsilon_\theta(\widehat{I}_{t_{i-1}}, I_{in}, t_{i-1}) \\ & + \frac{1}{2r}(\overline{\alpha}_{t_i} - \overline{\alpha}_{t_{i-1}})(I_{res}^\theta(\widehat{I}_{t_u}, I_{in}, t_u) - I_{res}^\theta(\widehat{I}_{t_{i-1}}, I_{in}, t_{i-1})) \\ & + \frac{1}{2r}(\overline{\beta}_{t_i} - \overline{\beta}_{t_{i-1}})(\epsilon_\theta(\widehat{I}_{t_u}, I_{in}, t_u) - \epsilon_\theta(\widehat{I}_{t_{i-1}}, I_{in}, t_{i-1})). \end{aligned} \quad (141)$$

The overall algorithm for second-order diffusion generalist solver is presented in Alg. 3.

**Algorithm 3:** Second-order Diffusion Generalist Solver.

---

**Input:** $I_{in}$, $I_{res}^{\theta}$, $\epsilon_{\theta}$, $\{\overline{\alpha}_{t_i}\}$, $\{\overline{\beta}_{t_i}\}$, $\{\overline{\delta}_{t_i}\}$, $\{t_i\}_{i=0}^{M}$, $r$.

1   $I_{t_0} \leftarrow I_T \sim \mathcal{N}(0, \overline{\beta}_{t_0} \boldsymbol{I})$

2 **for** $i = 1$ to $M$ **do**

3     $t_u = rt_i + (1-r)t_{i-1}$

4     $\widehat{I}_{res}^{t_{i-1}} = I_{res}^{\theta}(I_{t_{i-1}}, I_{in}, t_{i-1})$

5     $\widehat{\epsilon}_{t_{i-1}} = (I_{t_{i-1}} - (\overline{\alpha}_{t_{i-1}} - 1)\widehat{I}_{res}^{t_i} - (1 - \overline{\delta}_{t_{i-1}})I_{in})/\overline{\beta}_{t_{i-1}}$

6     $\widehat{I}_0^{t_{i-1}} = I_{t_{i-1}} - \overline{\alpha}_{t_{i-1}}\widehat{I}_{res}^{t_{i-1}} - \overline{\beta}_{t_{i-1}}\widehat{\epsilon}_{t_{i-1}} + \overline{\delta}_{t_{i-1}}I_{in}$

7     $I_{t_u} = I_{t_{i-1}} - (\overline{\delta}_{t_u} - \overline{\delta}_{t_{i-1}})I_{in} + (\overline{\alpha}_{t_u} - \overline{\alpha}_{t_{i-1}})\widehat{I}_{res}^{t_{i-1}} + (\overline{\beta}_{t_u} - \overline{\beta}_{t_{i-1}})\widehat{\epsilon}_{t_{i-1}}$

8     $\widehat{I}_{res}^{t_u} = I_{res}^{\theta}(I_{t_u}, I_{in}, t_u)$

9     $\widehat{\epsilon}_{t_u} = (I_{t_u} - (\overline{\alpha}_{t_u} - 1)\widehat{I}_{res}^{t_u} - (1 - \overline{\delta}_{t_u})I_{in})/\overline{\beta}_{t_u}$

10    $\widehat{I}_0^{t_u} = I_{t_u} - \overline{\alpha}_{t_u}\widehat{I}_{res}^{t_u} - \overline{\beta}_{t_u}\widehat{\epsilon}_{t_u} + \overline{\delta}_{t_u}I_{in}$

11    $I_{t_i} = I_{t_{i-1}} - (\overline{\delta}_{t_i} - \overline{\delta}_{t_{i-1}})I_{in} + (\overline{\alpha}_{t_i} - \overline{\alpha}_{t_{i-1}})\widehat{I}_{res}^{t_{i-1}} + (\overline{\beta}_{t_i} - \overline{\beta}_{t_{i-1}})\widehat{\epsilon}_{t_{i-1}} + \frac{1}{2r}(\overline{\alpha}_{t_i} - \overline{\alpha}_{t_{i-1}})(\widehat{I}_{res}^{t_u} - \widehat{I}_{res}^{t_{i-1}}) + \frac{1}{2r}(\overline{\beta}_{t_i} - \overline{\beta}_{t_{i-1}})(\widehat{\epsilon}_{t_u} - \widehat{\epsilon}_{t_{i-1}})$

**Output:** $I_{t_0}$.

---

For $k = 3$, Eq. (133) becomes:

$$
\begin{aligned}
I_{t_{i-1} \to t_i} = \ & I_{t_{i-1}} - (\overline{\delta}_{t_i} - \overline{\delta}_{t_{i-1}})I_{in} + (\overline{\alpha}_{t_i} - \overline{\alpha}_{t_{i-1}})I_{res}^{\theta}(\widehat{I}_{t_{i-1}}, I_{in}, t_{i-1}) + (\overline{\beta}_{t_i} - \overline{\beta}_{t_{i-1}})\epsilon_{\theta}(\widehat{I}_{t_{i-1}}, I_{in}, t_{i-1}) \\
& + \frac{1}{2}(\overline{\alpha}_{t_i} - \overline{\alpha}_{t_{i-1}})^2 I_{res}^{\theta(1)}(\widehat{I}_{t_{i-1}}, I_{in}, t_{i-1}) + \frac{1}{2}(\overline{\beta}_{t_i} - \overline{\beta}_{t_{i-1}})^2 \epsilon_{\theta}^{(1)}(\widehat{I}_{t_{i-1}}, I_{in}, t_{i-1}) \\
& + \frac{1}{6}(\overline{\alpha}_{t_i} - \overline{\alpha}_{t_{i-1}})^3 I_{res}^{\theta(2)}(\widehat{I}_{t_{i-1}}, I_{in}, t_{i-1}) + \frac{1}{6}(\overline{\beta}_{t_i} - \overline{\beta}_{t_{i-1}})^3 \epsilon_{\theta}^{(2)}(\widehat{I}_{t_{i-1}}, I_{in}, t_{i-1}),
\end{aligned}
\tag{142}
$$

we need two additional points to obtain the second-order derivative. Suppose the Taylor expansion of function $f(x)$ at $x + h_1$ and $x + h_2$ can be:

$$
f(x + h_1) = f(x) + h_1 f'(x) + \frac{h_1^2}{2} f''(x) + \mathcal{O}(h_1^3),
\tag{143}
$$

$$
f(x + h_2) = f(x) + h_2 f'(x) + \frac{h_2^2}{2} f''(x) + \mathcal{O}(h_2^3),
\tag{144}
$$

Then, we can derive the formulations by eliminating $f'(x)$:

$$
f''(x) = \frac{2}{h_1 h_2 (h_2 - h_1)} \big[ h_1 f(x + h_2) - h_2 f(x + h_1) + (h_2 - h_1)f(x) \big].
\tag{145}
$$

Similarly, two time variables $t_u = r_1 t_i + (1 - r_1)t_{i-1}$ and $t_s = r_2 t_i + (1 - r_2)t_{i-1}$, satisfying $i > s > u > i - 1$. Therefore, we can compute that:

$$
I_{res}^{\theta(2)}(\widehat{I}_{t_{i-1}}, I_{in}, t_{i-1}) = \frac{2(\overline{\alpha}_{t_i} - \overline{\alpha}_{t_{i-1}})^{-2}}{r_1 r_2 (r_2 - r_1)} \big[ r_1 I_{res}^{t_s} - r_2 I_{res}^{t_u} + (r_2 - r_1) I_{res}^{\theta}(\widehat{I}_{t_{i-1}}, I_{in}, t_{i-1}) \big],
\tag{146}
$$

$$
\epsilon_{\theta(2)}(\widehat{I}_{t_{i-1}}, I_{in}, t_{i-1}) = \frac{2(\overline{\alpha}_{t_i} - \overline{\alpha}_{t_{i-1}})^{-2}}{r_1 r_2 (r_2 - r_1)} \big[ r_1 \epsilon_{t_s} - r_2 \epsilon_{t_u} + (r_2 - r_1) \epsilon_{\theta}(\widehat{I}_{t_{i-1}}, I_{in}, t_{i-1}) \big].
\tag{147}
$$

where $I_{res}^{t_s} = I_{res}^{\theta}(\widehat{I}_{t_s}, I_{in}, t_s)$, $I_{res}^{t_u} = I_{res}^{\theta}(\widehat{I}_{t_u}, I_{in}, t_u)$ and $\epsilon_{t_s} = \epsilon_{\theta}(\widehat{I}_{t_s}, I_{in}, t_s)$, $\epsilon_{t_u} = \epsilon_{\theta}(\widehat{I}_{t_u}, I_{in}, t_u)$. Consequently, we can derive the formula as follows:

$$
\begin{aligned}
I_{t_i} = \ & I_{t_{i-1}} - (\overline{\delta}_{t_i} - \overline{\delta}_{t_{i-1}})I_{in} + (\overline{\alpha}_{t_i} - \overline{\alpha}_{t_{i-1}})I_{res}^{\theta}(\widehat{I}_{t_{i-1}}, I_{in}, t_{i-1}) + (\overline{\beta}_{t_i} - \overline{\beta}_{t_{i-1}})\epsilon_{\theta}(\widehat{I}_{t_{i-1}}, I_{in}, t_{i-1}) \\
& + \frac{1}{2r_1}(\overline{\alpha}_{t_i} - \overline{\alpha}_{t_{i-1}})(I_{res}^{\theta}(\widehat{I}_{t_u}, I_{in}, t_u) - I_{res}^{\theta}(\widehat{I}_{t_{i-1}}, I_{in}, t_{i-1})) \\
& + \frac{1}{2r_1}(\overline{\beta}_{t_i} - \overline{\beta}_{t_{i-1}})(\epsilon_{\theta}(\widehat{I}_{t_u}, I_{in}, t_u) - \epsilon_{\theta}(\widehat{I}_{t_{i-1}}, I_{in}, t_{i-1})) \\
& + \frac{(\overline{\alpha}_{t_i} - \overline{\alpha}_{t_{i-1}})}{3r_1 r_2 (r_2 - r_1)} \big[ r_1 I_{res}^{\theta}(\widehat{I}_{t_s}, I_{in}, t_s) - r_2 I_{res}^{\theta}(\widehat{I}_{t_u}, I_{in}, t_u) + (r_2 - r_1)I_{res}^{\theta}(\widehat{I}_{t_{i-1}}, I_{in}, t_{i-1}) \big] \\
& + \frac{(\overline{\alpha}_{t_i} - \overline{\alpha}_{t_{i-1}})}{3r_1 r_2 (r_2 - r_1)} \big[ r_1 \epsilon_{\theta}(\widehat{I}_{t_s}, I_{in}, t_s) - r_2 \epsilon_{\theta}(\widehat{I}_{t_u}, I_{in}, t_u) + (r_2 - r_1)\epsilon_{\theta}(\widehat{I}_{t_{i-1}}, I_{in}, t_{i-1}) \big].
\end{aligned}
\tag{148}
$$

The overall algorithm for second-order diffusion generalist solver is presented in Alg. 4.

---
**Algorithm 4:** Third-order Diffusion Generalist Solver.

---
**Input:** $I_{in}, I_{res}^{\theta}, \epsilon_{\theta}, \{\overline{\alpha}_{t_i}\}, \{\overline{\beta}_{t_i}\}, \{\overline{\delta}_{t_i}\}, \{t_i\}_{i=0}^{M}, r_1, r_2$.

**1** $I_{t_0} \leftarrow I_T \sim \mathcal{N}(0, \overline{\beta}_{t_0}\boldsymbol{I})$

**2 for** $i = 1$ to $M$ **do**

**3** $\quad t_u = r_1 t_i + (1 - r_1)t_{i-1}, t_s = r_2 t_i + (1 - r_2)t_{i-1}$

**4** $\quad \widehat{I}_{res}^{t_{i-1}} = I_{res}^{\theta}(I_{t_{i-1}}, I_{in}, t_{i-1}), \widehat{\epsilon}_{t_{i-1}} = (I_{t_{i-1}} - (\overline{\alpha}_{t_{i-1}} - 1)\widehat{I}_{res}^{t_{i-1}} - (1 - \overline{\delta}_{t_{i-1}})I_{in})/\overline{\beta}_{t_{i-1}},$
$\quad \widehat{I}_0^{t_{i-1}} = I_{t_{i-1}} - \overline{\alpha}_{t_{i-1}}\widehat{I}_{res}^{t_{i-1}} - \overline{\beta}_{t_{i-1}}\widehat{\epsilon}_{t_{i-1}} + \overline{\delta}_{t_{i-1}}I_{in}$

**5** $\quad I_{t_u} = I_{t_{i-1}} - (\overline{\delta}_{t_u} - \overline{\delta}_{t_{i-1}})I_{in} + (\overline{\alpha}_{t_u} - \overline{\alpha}_{t_{i-1}})\widehat{I}_{res}^{t_{i-1}} + (\overline{\beta}_{t_u} - \overline{\beta}_{t_{i-1}})\widehat{\epsilon}_{t_{i-1}}$

**6** $\quad \widehat{I}_{res}^{t_u} = I_{res}^{\theta}(I_{t_u}, I_{in}, t_u), \widehat{\epsilon}_{t_u} = (I_{t_u} - (\overline{\alpha}_{t_u} - 1)\widehat{I}_{res}^{t_u} - (1 - \overline{\delta}_{t_u})I_{in})/\overline{\beta}_{t_u},$
$\quad \widehat{I}_0^{t_u} = I_{t_u} - \overline{\alpha}_{t_u}\widehat{I}_{res}^{t_u} - \overline{\beta}_{t_u}\widehat{\epsilon}_{t_u} + \overline{\delta}_{t_u}I_{in}$

**7** $\quad I_{t_s} = I_{t_{i-1}} - (\overline{\delta}_{t_s} - \overline{\delta}_{t_{i-1}})I_{in} + (\overline{\alpha}_{t_s} - \overline{\alpha}_{t_{i-1}})\widehat{I}_{res}^{t_{i-1}} + (\overline{\beta}_{t_s} - \overline{\beta}_{t_{i-1}})\widehat{\epsilon}_{t_{i-1}} + \frac{r_2}{2r_1}(\overline{\alpha}_{t_s} -$
$\quad \overline{\alpha}_{t_{i-1}})(\widehat{I}_{res}^{t_u} - \widehat{I}_{res}^{t_{i-1}}) + \frac{r_2}{2r_1}(\overline{\beta}_{t_s} - \overline{\beta}_{t_{i-1}})(\widehat{\epsilon}_{t_u} - \widehat{\epsilon}_{t_{i-1}})$

**8** $\quad \widehat{I}_{res}^{t_s} = I_{res}^{\theta}(I_{t_s}, I_{in}, t_s), \widehat{\epsilon}_{t_s} = (I_{t_s} - (\overline{\alpha}_{t_s} - 1)\widehat{I}_{res}^{t_s} - (1 - \overline{\delta}_{t_s})I_{in})/\overline{\beta}_{t_s},$
$\quad \widehat{I}_0^{t_s} = I_{t_s} - \overline{\alpha}_{t_s}\widehat{I}_{res}^{t_s} - \overline{\beta}_{t_s}\widehat{\epsilon}_{t_s} + \overline{\delta}_{t_s}I_{in}$

**9** $\quad D_1^{res} = \widehat{I}_{res}^{t_u} - \widehat{I}_{res}^{t_{i-1}}, D_1^{\epsilon} = \widehat{\epsilon}_{t_u} - \widehat{\epsilon}_{t_{i-1}}$

**10** $\quad D_2^{res} = \frac{2}{r_1 r_2(r_2 - r_1)}(r_1\widehat{I}_{res}^{t_s} - r_2\widehat{I}_{res}^{t_u} + (r_2 - r_1)\widehat{I}_{res}^{t_{i-1}})$

**11** $\quad D_2^{\epsilon} = \frac{2}{r_1 r_2(r_2 - r_1)}(r_1\widehat{\epsilon}_{t_s} - r_2\widehat{\epsilon}_{t_u} + (r_2 - r_1)\widehat{\epsilon}_{t_{i-1}})$

**12** $\quad I_{t_i} = I_{t_{i-1}} - (\overline{\delta}_{t_i} - \overline{\delta}_{t_{i-1}})I_{in} + (\overline{\alpha}_{t_i} - \overline{\alpha}_{t_{i-1}})\widehat{I}_{res}^{t_{i-1}} + (\overline{\beta}_{t_i} - \overline{\beta}_{t_{i-1}})\widehat{\epsilon}_{t_{i-1}} + \frac{1}{2r_1}(\overline{\alpha}_{t_i} -$
$\quad \overline{\alpha}_{t_{i-1}})D_1^{res} + \frac{1}{2r_1}(\overline{\beta}_{t_i} - \overline{\beta}_{t_{i-1}})D_1^{\epsilon} + \frac{1}{6}(\overline{\alpha}_{t_i} - \overline{\alpha}_{t_{i-1}})D_2^{res} + \frac{1}{6}(\overline{\beta}_{t_i} - \overline{\beta}_{t_{i-1}})D_2^{\epsilon}$

**Output:** $I_{t_M}$.

---

## B.2 Cumulative Error Bound of Diffusion Generalist Solvers

For $k$-order diffusion generalist solvers, we make the following assumptions:

**Assumption 1.** The total derivatives $I_{res}^{\theta(n)}(I_{\overline{\alpha}_t}, I_{in}, \overline{\alpha}) = \frac{d^{(n)}\widehat{I}_{res}^{\theta}(\widehat{I}_{\overline{\alpha}_t}, I_{in}, \overline{\alpha})}{d\overline{\alpha}^{(n)}}$ and $\epsilon_{\theta}^{(n)}(I_{\overline{\beta}_t}, I_{in}, \overline{\beta}) = \frac{d^{(n)}\widehat{\epsilon}(\widehat{I}_{\overline{\beta}_t}, I_{in}, \overline{\beta})}{d\overline{\beta}^{(n)}}$ exist and are continuous for $0 \leq n \leq k + 1$.

**Assumption 2.** The function $I_{res}^{\theta}(I_{\overline{\alpha}_t}, I_{in}, \overline{\alpha})$ and $\epsilon_{\theta}(I_{\overline{\beta}_t}, I_{in}, \overline{\beta})$ are Lipschitz with respect to $I_t$.

**Assumption 3.** The step sizes for $\overline{\alpha}$ and $\overline{\beta}$ are limited by $\max(\overline{\alpha}_{t_i} - \overline{\alpha}_{t_{i-1}}) = \mathcal{O}(1/M_{\overline{\alpha}})$ and $\max(\overline{\beta}_{t_i} - \overline{\beta}_{t_{i-1}}) = \mathcal{O}(1/M_{\overline{\beta}})$, respectively. Besides, $1/M_{\overline{\alpha}} < 1/M$ and $1/M_{\overline{\beta}} < 1/M$.

*Proof of $k = 1$.* Taking $n = 0, t = t_i, s = t_{i-1}$ in Eq. (134), we obtain:

$$I_{t_i} = I_{t_{i-1}} - (\overline{\delta}_{t_i} - \overline{\delta}_{t_{i-1}})I_{in} + (\overline{\alpha}_{t_i} - \overline{\alpha}_{t_{i-1}})I_{res}^{\theta}(I_{t_{i-1}}, I_{in}, t_{i-1}) + \mathcal{O}((\overline{\alpha}_t - \overline{\alpha}_{t_s})^2)$$
$$+ (\overline{\beta}_{t_i} - \overline{\beta}_{t_{i-1}})\epsilon_{\theta}(I_{t_{i-1}}, I_{in}, t_{i-1}) + \mathcal{O}((\overline{\beta}_t - \overline{\beta}_{t_s})^2), \tag{149}$$

Based on the Assumptions and Eq. (136), we can derive:

$$\widehat{I}_{t_i} = \widehat{I}_{t_{i-1}} - (\overline{\delta}_{t_i} - \overline{\delta}_{t_{i-1}})I_{in} + (\overline{\alpha}_{t_i} - \overline{\alpha}_{t_{i-1}})I_{res}^{\theta}(\widehat{I}_{t_{i-1}}, I_{in}, t_{i-1}) + (\overline{\beta}_{t_i} - \overline{\beta}_{t_{i-1}})\epsilon_{\theta}(\widehat{I}_{t_{i-1}}, I_{in}, t_{i-1})$$
$$= I_{t_{i-1}} - (\overline{\delta}_{t_i} - \overline{\delta}_{t_{i-1}})I_{in} + (\overline{\alpha}_{t_i} - \overline{\alpha}_{t_{i-1}})(I_{res}^{\theta}(I_{t_{i-1}}, I_{in}, t_{i-1}) + \mathcal{O}(\widehat{I}_{t_{i-1}} - I_{t_{i-1}}))$$
$$+ (\overline{\beta}_{t_i} - \overline{\beta}_{t_{i-1}})(\epsilon_{\theta}(I_{t_{i-1}}, I_{in}, t_{i-1}) + \mathcal{O}(\widehat{I}_{t_{i-1}} - I_{t_{i-1}}))$$
$$= I_{t_i} + \mathcal{O}(\max(\overline{\alpha}_{t_i} - \overline{\alpha}_{t_{i-1}})^2) + \mathcal{O}(\max(\overline{\beta}_{t_i} - \overline{\beta}_{t_{i-1}})^2) + \mathcal{O}(\widehat{I}_{t_{i-1}} - I_{t_{i-1}}).$$
$$\tag{150}$$

By repeating the process recursively $M$ times, we get:

$$\widehat{I}_{t_M} = I_{t_0} + \mathcal{O}(M\max(\overline{\alpha}_{t_i} - \overline{\alpha}_{t_{i-1}})^2) + \mathcal{O}(M\max(\overline{\beta}_{t_i} - \overline{\beta}_{t_{i-1}})^2)$$
$$= I_{t_0} + \mathcal{O}(\max(\overline{\alpha}_{t_i} - \overline{\alpha}_{t_{i-1}})) + \mathcal{O}(\max(\overline{\beta}_{t_i} - \overline{\beta}_{t_{i-1}})). \tag{151}$$

*Proof of $k = 2$.* We consider the following update for $0 < s < u < r < M$ in Alg. 3:

$$\widehat{I}_{t_u} = I_{t_s} - (\overline{\delta}_{t_u} - \overline{\delta}_{t_s})I_{in} + (\overline{\alpha}_{t_u} - \overline{\alpha}_{t_s})I_{res}^{\theta}(I_{t_s}, I_{in}, t_s) + (\overline{\beta}_{t_u} - \overline{\beta}_{t_s})\epsilon_{\theta}(I_{t_s}, I_{in}, t_s), \tag{152}$$

$$\widehat{I}_{t_r} = I_{t_s} - (\overline{\delta}_{t_r} - \overline{\delta}_{t_s})I_{in} + (\overline{\alpha}_{t_r} - \overline{\alpha}_{t_s})I_{res}^{\theta}(I_{t_s}, I_{in}, t_s) + (\overline{\beta}_{t_r} - \overline{\beta}_{t_s})\epsilon_{\theta}(I_{t_s}, I_{in}, t_s)$$
$$+ \frac{1}{2r}(\overline{\alpha}_{t_r} - \overline{\alpha}_{t_s})(I_{res}^{\theta}(\widehat{I}_{t_u}, I_{in}, t_u) - I_{res}^{\theta}(I_{t_s}, I_{in}, t_s)) \tag{153}$$
$$+ \frac{1}{2r}(\overline{\beta}_{t_r} - \overline{\beta}_{t_s})(\epsilon_{\theta}(\widehat{I}_{t_u}, I_{in}, t_u) - \epsilon_{\theta}(I_{t_s}, I_{in}, t_s)).$$

Taking $n = 1$, we can obtain:

$$I_{t_r} = I_{t_s} - (\overline{\delta}_{t_r} - \overline{\delta}_{t_s})I_{in} + (\overline{\alpha}_{t_r} - \overline{\alpha}_{t_s})I_{res}^{\theta}(I_{t_s}, I_{in}, t_s) + (\overline{\beta}_{t_r} - \overline{\beta}_{t_s})\epsilon_{\theta}(I_{t_s}, I_{in}, t_s)$$
$$+ \frac{1}{2}(\overline{\alpha}_{t_r} - \overline{\alpha}_{t_s})^2 I_{res}^{\theta(1)}(I_{\overline{\alpha}_{t_s}}, I_{in}, \overline{\alpha}_{t_s}) + \mathcal{O}((\overline{\alpha}_t - \overline{\alpha}_{t_s})^3) \tag{154}$$
$$+ \frac{1}{2}(\overline{\beta}_{t_r} - \overline{\beta}_{t_s})^2 \epsilon_{\theta}^{(1)}(I_{\overline{\beta}_{t_s}}, I_{in}, \overline{\beta}_{t_s}) + \mathcal{O}((\overline{\beta}_t - \overline{\beta}_{t_s})^3).$$

From Eq. (153), we have:

$$I_{res}^{\theta}(I_{t_u}, I_{in}, t_u) = I_{res}^{\theta}(I_{t_s}, I_{in}, t_s) + (\overline{\alpha}_{t_u} - \overline{\alpha}_{t_s})\widehat{I}_{res}^{\theta(1)}(I_{\overline{\alpha}_{t_s}}, I_{in}, \overline{\alpha}_{t_s}) + \mathcal{O}((\overline{\alpha}_{t_u} - \overline{\alpha}_{t_s})^2), \tag{155}$$

$$\epsilon_{\theta}(I_{t_u}, I_{in}, t_u) = \epsilon_{\theta}(I_{t_s}, I_{in}, t_s) + (\overline{\beta}_{t_u} - \overline{\beta}_{t_s})\widehat{\epsilon}_{\theta}^{(1)}(I_{\overline{\beta}_{t_s}}, I_{in}, \overline{\beta}_{t_s}) + \mathcal{O}((\overline{\beta}_{t_u} - \overline{\beta}_{t_s})^2), \tag{156}$$

$$\widehat{I}_{t_r} = I_{t_s} - (\overline{\delta}_{t_r} - \overline{\delta}_{t_s})I_{in} + (\overline{\alpha}_{t_r} - \overline{\alpha}_{t_s})I_{res}^{\theta}(I_{t_s}, I_{in}, t_s) + (\overline{\beta}_{t_r} - \overline{\beta}_{t_s})\epsilon_{\theta}(I_{t_s}, I_{in}, t_s)$$
$$+ \frac{1}{2r}(\overline{\alpha}_{t_r} - \overline{\alpha}_{t_s})(I_{res}^{\theta}(\widehat{I}_{t_u}, I_{in}, t_u) - I_{res}^{\theta}(I_{t_u}, I_{in}, t_u) + (\overline{\alpha}_{t_u} - \overline{\alpha}_{t_s})\widehat{I}_{res}^{\theta(1)}(I_{\overline{\alpha}_{t_s}}, I_{in}, \overline{\alpha}_{t_s}) + \mathcal{O}((\overline{\alpha}_{t_u} - \overline{\alpha}_{t_s})^2))$$
$$+ \frac{1}{2r}(\overline{\beta}_{t_r} - \overline{\beta}_{t_s})(\epsilon_{\theta}(\widehat{I}_{t_u}, I_{in}, t_u) - \epsilon_{\theta}(I_{t_u}, I_{in}, t_u) + (\overline{\beta}_{t_u} - \overline{\beta}_{t_s})\widehat{\epsilon}_{\theta}^{(1)}(I_{\overline{\beta}_{t_s}}, I_{in}, \overline{\beta}_{t_s}) + \mathcal{O}((\overline{\beta}_{t_u} - \overline{\beta}_{t_s})^2)), \tag{157}$$

According to the Lipschitzness of $I_{res}^{\theta}$ and $\epsilon_{\theta}$ and the conclusion from Eq. (150), we can compute:

$$\|I_{res}^{\theta}(\widehat{I}_{t_u}, I_{in}, t_u) - I_{res}^{\theta}(I_{t_u}, I_{in}, t_u)\| = \mathcal{O}(\|\widehat{I}_{t_u} - I_{t_u}\|) = \mathcal{O}((\overline{\alpha}_{t_u} - \overline{\alpha}_{t_s})^2) + \mathcal{O}((\overline{\beta}_{t_u} - \overline{\beta}_{t_s})^2), \tag{158}$$

$$\|\epsilon_{\theta}(\widehat{I}_{t_u}, I_{in}, t_u) - \epsilon_{\theta}(I_{t_u}, I_{in}, t_u)\| = \mathcal{O}(\|\widehat{I}_{t_u} - I_{t_u}\|) = \mathcal{O}((\overline{\alpha}_{t_u} - \overline{\alpha}_{t_s})^2) + \mathcal{O}((\overline{\beta}_{t_u} - \overline{\beta}_{t_s})^2). \tag{159}$$

As $(\overline{\alpha}_{t_u} - \overline{\alpha}_{t_s}) = r(\overline{\alpha}_{t_r} - \overline{\alpha}_{t_s})$ and $(\overline{\beta}_{t_u} - \overline{\beta}_{t_s}) = r(\overline{\beta}_{t_r} - \overline{\beta}_{t_s})$ for linear generation schedule, we subtract $\widehat{I}_{t_r}$ in Eq. (157) from $I_{t_r}$ in Eq. (154):

$$I_{t_r} - \widehat{I}_{t_r} = (\overline{\alpha}_{t_r} - \overline{\alpha}_{t_s})\left(\frac{(\overline{\alpha}_{t_r} - \overline{\alpha}_{t_s})}{2} - \frac{(\overline{\alpha}_{t_u} - \overline{\alpha}_{t_s})}{2r}\right)\widehat{I}_{res}^{\theta(1)}(I_{\overline{\alpha}_{t_s}}, I_{in}, \overline{\alpha}_{t_s}) + \mathcal{O}((\overline{\alpha}_{t_u} - \overline{\alpha}_{t_s})^3)$$
$$+ (\overline{\beta}_{t_r} - \overline{\beta}_{t_s})\left(\frac{(\overline{\beta}_{t_r} - \overline{\beta}_{t_s})}{2} - \frac{(\overline{\beta}_{t_u} - \overline{\beta}_{t_s})}{2r}\right)\widehat{\epsilon}_{\theta}^{(1)}(I_{\overline{\beta}_{t_s}}, I_{in}, \overline{\beta}_{t_s})) + \mathcal{O}((\overline{\beta}_{t_u} - \overline{\beta}_{t_s})^3) \tag{160}$$
$$= \mathcal{O}((\overline{\alpha}_{t_r} - \overline{\alpha}_{t_s})^3) + \mathcal{O}((\overline{\beta}_{t_r} - \overline{\beta}_{t_s})^3).$$

*Proof of $k = 3$.* We consider the following update for $i - 1 < s < u < v < r < i$ in Alg. 4, and we can obtain:

$$\widehat{I}_{t_r} = I_{t_s} - (\overline{\delta}_{t_r} - \overline{\delta}_{t_s})I_{in} + (\overline{\alpha}_{t_r} - \overline{\alpha}_{t_s})I_{res}^{\theta}(I_{t_s}, I_{in}, t_s) + (\overline{\beta}_{t_r} - \overline{\beta}_{t_s})\epsilon_{\theta}(I_{t_s}, I_{in}, t_s)$$
$$+ \frac{1}{2r_1}(\overline{\alpha}_{t_r} - \overline{\alpha}_{t_s})(I_{res}^{\theta}(I_{\overline{\alpha}_{t_u}}, I_{in}, \overline{\alpha}_{t_u}) - I_{res}^{\theta}(I_{\overline{\alpha}_{t_s}}, I_{in}, \overline{\alpha}_{t_s}))$$
$$+ \frac{1}{2r_1}(\overline{\beta}_{t_r} - \overline{\beta}_{t_s})(\epsilon_{\theta}(I_{\overline{\beta}_{t_u}}, I_{in}, \overline{\beta}_{t_u}) - \epsilon_{\theta}(I_{\overline{\beta}_{t_s}}, I_{in}, \overline{\beta}_{t_s}))$$
$$+ \frac{(\overline{\alpha}_{t_r} - \overline{\alpha}_{t_s})}{3r_1r_2(r_2 - r_1)}(r_1 I_{res}^{\theta}(I_{\overline{\alpha}_{t_v}}, I_{in}, \overline{\alpha}_{t_v}) - r_2 I_{res}^{\theta}(I_{\overline{\alpha}_{t_u}}, I_{in}, \overline{\alpha}_{t_u}) + (r_2 - r_1)I_{res}^{\theta}(I_{\overline{\alpha}_{t_s}}, I_{in}, \overline{\alpha}_{t_s}))$$
$$+ \frac{(\overline{\beta}_{t_r} - \overline{\beta}_{t_s})}{3r_1r_2(r_2 - r_1)}(r_1 \epsilon_{\theta}(I_{\overline{\beta}_{t_v}}, I_{in}, \overline{\beta}_{t_v}) - r_2 \epsilon_{\theta}(I_{\overline{\beta}_{t_u}}, I_{in}, \overline{\beta}_{t_u}) + (r_2 - r_1)\epsilon_{\theta}(I_{\overline{\beta}_{t_s}}, I_{in}, \overline{\beta}_{t_s}))$$
$$+ \mathcal{O}((\overline{\beta}_{t_r} - \overline{\beta}_{t_s})^4) + \mathcal{O}((\overline{\alpha}_{t_r} - \overline{\alpha}_{t_s})^4). \tag{161}$$

Similar to aforementioned proof, we have:

$$I_{t_u} = \widehat{I}_{t_u} + \mathcal{O}((\overline{\beta}_{t_r} - \overline{\beta}_{t_s})^2) + \mathcal{O}((\overline{\alpha}_{t_r} - \overline{\alpha}_{t_s})^2), \tag{162}$$

$$I_{t_v} = \widehat{I}_{t_v} + \mathcal{O}((\overline{\beta}_{t_r} - \overline{\beta}_{t_s})^3) + \mathcal{O}((\overline{\alpha}_{t_r} - \overline{\alpha}_{t_s})^3), \tag{163}$$

$$I_{t_r} = I_{t_s} - (\overline{\delta}_{t_r} - \overline{\delta}_{t_s})I_{in} + (\overline{\alpha}_{t_r} - \overline{\alpha}_{t_s})I_{res}^{\theta}(I_{t_s}, I_{in}, t_s) + (\overline{\beta}_{t_r} - \overline{\beta}_{t_s})\epsilon_{\theta}(I_{t_s}, I_{in}, t_s)$$
$$+ \frac{1}{2}(\overline{\alpha}_{t_r} - \overline{\alpha}_{t_s})^2 I_{res}^{\theta(1)}(I_{\overline{\alpha}_{t_s}}, I_{in}, \overline{\alpha}_{t_s}) + \frac{1}{6}(\overline{\alpha}_{t_r} - \overline{\alpha}_{t_s})^3 I_{res}^{\theta(2)}(I_{\overline{\alpha}_{t_s}}, I_{in}, \overline{\alpha}_{t_s}) + \mathcal{O}((\overline{\alpha}_t - \overline{\alpha}_{t_s})^4) \tag{164}$$
$$+ \frac{1}{2}(\overline{\beta}_{t_r} - \overline{\beta}_{t_s})^2 \epsilon_{\theta(1)}(I_{\overline{\beta}_{t_s}}, I_{in}, \overline{\beta}_{t_s}) + \frac{1}{6}(\overline{\beta}_{t_r} - \overline{\beta}_{t_s})^3 \epsilon_{\theta(2)}(I_{\overline{\beta}_{t_s}}, I_{in}, \overline{\beta}_{t_s}) + \mathcal{O}((\overline{\beta}_t - \overline{\beta}_{t_s})^4).$$

By applying Taylor expansion, we can get the conclusion as follows:

$$I_{t_r} - \widehat{I}_{t_r} = \mathcal{O}((\overline{\beta}_{t_r} - \overline{\beta}_{t_s})^4) + \mathcal{O}((\overline{\alpha}_{t_r} - \overline{\alpha}_{t_s})^4). \tag{165}$$

## C    Universal Diffusion Posterior Sampling

**Lemma 1** *Tweedie's formula. Let $q(y|\eta)$ be the exponential family distribution:*

$$q(y|\eta) = q_0(y) \exp(\eta^T(y) - \varphi(\eta)), \tag{166}$$

*among them, $\eta$ is called the natural parameter, $\varphi(\eta)$ is called the cumulant generating function for normalizing the density, and $q_0(y)$ is the density up to the scale factor when $\eta = 0$. Then, the posterior mean $\widehat{\eta} := \mathbb{E}[\eta|y]$ should satisfy:*

$$(\nabla_y T(y))^T \widehat{\eta} = \nabla_y \log q(y) - \nabla_y \log q_0(y). \tag{167}$$

*Proof.* Marginal distribution $q(y)$ could be expressed as:

$$q(y) = \int q(y|\eta)q(\eta)d\eta = \int q_0(y) \exp\left(\eta^T T(y) - \varphi(\eta)\right) q(\eta)d\eta, \tag{168}$$

Then, the derivative of the marginal distribution $q(y)$ with respect to $y$ becomes:

$$\nabla_y q(y) = \nabla_y q_0(y) \int \exp(\eta^T T(y) - \varphi(\eta))q(\eta)d\eta + \int (\nabla_y T(y))^T \eta q_0(y) \exp(\eta^T T(y) - \varphi(\eta))q(\eta)d\eta$$

$$= \frac{\nabla_y q_0(y)}{q_0(y)} \int q(y|\eta)q(\eta)d\eta + (\nabla_y T(y))^T \int \eta q(y|\eta)q(\eta)d\eta$$

$$= \frac{\nabla_y q_0(y)}{q_0(y)} q(y) + (\nabla_y T(y))^T \int \eta q(y, \eta)d\eta. \tag{169}$$

Therefore:

$$\frac{\nabla_y q(y)}{q(y)} = \frac{\nabla_y q_0(y)}{q_0(y)} + (\nabla_y T(y))^T \int \eta q(\eta|y)d\eta, \tag{170}$$

which is equivalent to:

$$\nabla_y \log q(y) = \nabla_y \log q_0(y) + (\nabla_y T(y))^T \mathbb{E}(\eta|y). \tag{171}$$

This concludes the proof.

**Proposition 2** *Jensen gap upper bound [12]. Define the absolute centered moment as $m_p := \sqrt[p]{\mathbb{E}[\|X - \mu\|^p]}$, and the mean as $\mu = \mathbb{E}[X]$. Assume that for $\alpha > 0$, there exists a positive number $K$ such that for any $x \in \mathbb{R}$, $|f(c) - f(\mu)| \leq K|x - \mu|^\alpha$. Then:*

$$\mathbb{E}[|f(X) - f(\mathbb{E}[X])|] \leq \int |f(x) - f(\mu)|dq(X)$$

$$\leq K \int |x - \mu|^\alpha dq(X) \leq Mm_\alpha^\alpha, \tag{172}$$

*where $M$ is an upper bound estimator constant that can be taken as $K$ or other constants related to the function $f$ and the distribution $q(X)$.*

**Lemma 2** *Let $\phi(\cdot)$ be a univariate Gaussian density function with mean $\mu$ and variance $\sigma^2$, there exists a Lipschitz constant $L$ such that:*

$$|\phi(x) - \phi(y)| \leq L|x - y|, \tag{173}$$

*where $L = \frac{1}{\sqrt{2\pi\sigma^2}} \exp\left(-\frac{1}{2\sigma^2}\right)$.*

*Proof.* As $\phi'$ is continuous and bounded, we use the mean value theorem to get:

$$\forall (x, y) \in \mathbb{R}^2, |\phi(x) - \phi(y)| \leq \|\phi'\|_\infty |x - y|, \tag{174}$$

Since $L$ is the Lipschitz constant, we have that $L \leq \|\phi'\|_\infty$. Taking the limit $y \to x$ gives $\|\phi'\| \leq L$, and thus $\|\phi'\|_\infty \leq L$. Hence:

$$L = \|\phi'\|_\infty = \| - \frac{x - \mu}{\sigma^2}\phi(x)\|_\infty. \tag{175}$$

Since the derivative of $\phi'$ is given as:

$$\phi'' = \sigma^{-2}(1 - \sigma^{-2}(x - \mu)^2)\phi(x), \tag{176}$$

Setting $\phi'' = 0$, we get $\sigma^{-2}(1 - \sigma^{-2}(x - \mu)^2) = 0$, which gives $x = \mu \pm \sigma$, and we have:

$$L = \|\phi'\|_\infty = \frac{e^{-1/2\sigma^2}}{\sqrt{2\pi\sigma^2}}. \tag{177}$$

**Lemma 3** *Let $\phi(\cdot)$ be an isotropic multivariate Gaussian density function with mean $\mu$ and variance $\sigma^2 \boldsymbol{I}$. There exists a constant $L$ such that $\forall x, y \in \mathbb{R}^d$:*

$$|\phi(x) - \phi(y)| \leq L|x - y|, \tag{178}$$

*where $L = \frac{d}{\sqrt{2\pi\sigma^2}} \exp\left(-\frac{1}{2\sigma^2}\right)$*

*Proof.*

$$\|\phi(x) - \phi(y)\| \leq \max_z \|\nabla_z \phi(z)\| \cdot \|x - y\| = \frac{d}{\sqrt{2\pi\sigma^2}} \exp(-\frac{1}{2\sigma^2}) \cdot \|x - y\|, \tag{179}$$

each element of $d$ dimensions $\nabla_z \phi(z)$ are bounded by $\frac{1}{\sqrt{2\pi\sigma^2}} \exp(-\frac{1}{2\sigma^2})$.

**Proposition 3** *For the measurement model $I_{in} = \mathcal{A}I_0 + n$ in linear verse problem with $n \sim \mathcal{N}(0, \sigma^2 \boldsymbol{I})$, we have:*

$$q(I_{in}|I_t) \simeq q(I_{in}|\widehat{I}_0), \tag{180}$$

*where the approximation error can be quantified with the Jensen gap, which is upper bounded by:*

$$\mathcal{J} \leq \frac{d}{\sqrt{2\pi\sigma^2}} \exp\left(-\frac{1}{2\sigma^2}\right) \|\nabla_{I_0} \mathcal{A}(I_0)\| m_1, \tag{181}$$

*where $\|\nabla_I \mathcal{A}(I)\| := \max_I \|\nabla_{I_0} \mathcal{A}(I_0)\|$ and $m_1 := \int \|I_0 - \widehat{I}_0\| q(I_0|I_t) dI_0$.*

*Proof.* The measurement model $I_{in} = \mathcal{A}I_0 + n$ can be formulated as:

$$
\begin{aligned}
q(I_{in}|I_0) \sim \mathcal{N}(\mathcal{A}I_0, \sigma^2 \boldsymbol{I}) &= \frac{1}{\sqrt{2\pi\sigma^2}} \exp\left(-\frac{(I_{in} - \mathcal{A}I_0)^2}{2\sigma^2}\right) \\
&= \frac{1}{\sqrt{2\pi\sigma^2}} \exp\left(\frac{\mathcal{A}I_0}{\sigma^2} I_{in} - \frac{(\mathcal{A}I_0)^2}{2\sigma^2}\right) \exp\left(-\frac{I_{in}^2}{2\sigma^2}\right) \\
&= \left[\frac{1}{\sqrt{2\pi\sigma^2}} \exp\left(-\frac{I_{in}^2}{2\sigma^2}\right)\right] \exp\left(\frac{\mathcal{A}I_0}{\sigma^2} I_{in} - \frac{(\mathcal{A}I_0)^2}{2\sigma^2}\right).
\end{aligned}
\tag{182}
$$

Hence, as a member of the exponential family distribution $q(y|\eta) = q_0(y) \exp(\eta^T(y) - \varphi(\eta))$, the undetermined functions are:

$$q_0(I_{in}) = \left[\frac{1}{\sqrt{2\pi\sigma^2}} \exp\left(-\frac{I_{in}^2}{2\sigma^2}\right)\right], \ \eta^T(I_{in}) = \frac{\mathcal{A}I_0}{\sigma^2} I_{in}, \ \varphi(I_0) = \exp\left(\frac{\mathcal{A}I_0}{\sigma^2} I_{in} - \frac{(\mathcal{A}I_0)^2}{2\sigma^2}\right). \tag{183}$$

In order to exploit the measurement model $q(I_{in}|I_0)$, we factorize $q(I_{in}|I_t)$ as follows:

$$q(I_{in}|I_t) = \int q(I_{in}|I_0) q(I_0|I_t) dI_0 = \mathbb{E}_{I_0 \sim q(I_0|I_t)}[f(I_0)], \tag{184}$$

here, $f(\cdot) := h(\mathcal{A}(\cdot))$, where $\mathcal{A}$ is the measurement operator and $h(\cdot)$ is the multivariate normal distribution with mean $\mathcal{A}(I_0)$ and the covariance $\sigma^2 \boldsymbol{I}$. Therefore, we have:

$$
\begin{aligned}
\mathcal{J}(f, q(I_0|I_t)) &= |\mathbb{E}[f(I_0)] - f(\mathbb{E}[I_0])| = |\mathbb{E}[f(I_0)] - f(\widehat{I}_0)|, \tag{185} \\
&= |\mathbb{E}[h(\mathcal{A}(I_0))] - h(\mathcal{A}(\widehat{I}_0))|, \tag{186} \\
&\leq \int |h(\mathcal{A}(I_0)) - h(\mathcal{A}(\widehat{I}_0))| dQ(I_0|I_t), \tag{187} \\
&\leq \frac{d}{\sqrt{2\pi\sigma^2}} e^{-\frac{1}{2\sigma^2}} \int \|\mathcal{A}(I_0) - \mathcal{A}(\widehat{I}_0)\| dQ(I_0|I_t), \tag{188} \\
&\leq \frac{d}{\sqrt{2\pi\sigma^2}} e^{-\frac{1}{2\sigma^2}} \|\nabla_{I_0} \mathcal{A}(I_0)\| \int \|I_0 - \widehat{I}_0\| dQ(I_0|I_t), \tag{189} \\
&\leq \frac{d}{\sqrt{2\pi\sigma^2}} e^{-\frac{1}{2\sigma^2}} \|\nabla_{I_0} \mathcal{A}(I_0)\| m_1, \tag{190}
\end{aligned}
$$

where $dQ(I_0|I_t) = q(I_0|I_t) dI_0$.

**Proposition 4** *For the measurement model $I_{in} = I_0 + I_{res} + n$ in linear inverse problem with $n \sim \mathcal{N}(0, \sigma^2 \boldsymbol{I})$, we have:*

$$q(I_{in}|I_t) \simeq q(I_{in}|\widehat{I}_0, \widehat{I}_{res}), \tag{191}$$

*the approximation error can be quantified with the Jensen gap, which is upper bounded by:*

$$\mathcal{J} \leq \frac{d}{\sqrt{2\pi\sigma^2}} \exp\left(-\frac{1}{2\sigma^2}\right) m_1^{I_0 + I_{res}}, \tag{192}$$

*where $m_1^{I_0 + I_{res}} := \int \|(I_0 + I_{res}) - (\widehat{I}_0 + \widehat{I}_{res})\| q(I_0, I_{res}|I_t) dI_0 dI_{res}$.*

*Proof.* The measurement model $I_{in} = I_0 + I_{res} + n$ can be formulated as:

$$q(I_{in}|I_0) \sim \mathcal{N}((I_0 + I_{res}), \sigma^2 \boldsymbol{I}) = \frac{1}{\sqrt{2\pi\sigma^2}} \exp\big(-\frac{(I_{in} - (I_0 + I_{res}))^2}{2\sigma^2}\big)$$

$$= \frac{1}{\sqrt{2\pi\sigma^2}} \exp\big(\frac{(I_0 + I_{res})}{\sigma^2} I_{in} - \frac{((I_0 + I_{res}))^2}{2\sigma^2}\big) \exp\big(-\frac{I_{in}^2}{2\sigma^2}\big) \quad (193)$$

$$= \Big[\frac{1}{\sqrt{2\pi\sigma^2}} \exp\big(-\frac{I_{in}^2}{2\sigma^2}\big)\Big] \exp\big(\frac{(I_0 + I_{res})}{\sigma^2} I_{in} - \frac{((I_0 + I_{res}))^2}{2\sigma^2}\big).$$

In order to exploit the measurement model $q(I_{in}|I_0)$, we factorize $q(I_{in}|I_t)$ as follows:

$$q(I_{in}|I_t) = \int\int q(I_{in}|I_0, I_{res}) q(I_0, I_{res}|I_t) dI_0 dI_{res}, \quad (194)$$

here, $f(\cdot) := h(I_0 + I_{res})$, and $h(\cdot)$ is the multivariate normal distribution with mean $I_0 + I_{res}$ and the covariance $\sigma^2 \boldsymbol{I}$. Therefore, we have:

$$\mathcal{J}(f, q(I_0|I_t)) = |\mathbb{E}[f(I_0)] - f(\mathbb{E}[I_0])| = |\mathbb{E}[f(I_0)] - f(\widehat{I_0})|, \quad (195)$$

$$= |\mathbb{E}[h(I_0 + I_{res})] - h(\widehat{I_0} + \widehat{I}_{res})|, \quad (196)$$

$$\leq \int |h(I_0 + I_{res}) - h(\widehat{I_0} + \widehat{I}_{res})| dQ(I_0, I_{res}|I_t), \quad (197)$$

$$\leq \frac{d}{\sqrt{2\pi\sigma^2}} e^{-\frac{1}{2\sigma^2}} \int\int \|(I_0 + I_{res}) - (\widehat{I_0} + \widehat{I}_{res})\| q(I_0, I_{res}|I_t) dI_0 dI_{res}, \quad (198)$$

$$\leq \frac{d}{\sqrt{2\pi\sigma^2}} e^{-\frac{1}{2\sigma^2}} m_1^{I_0 + I_{res}}, \quad (199)$$

where $q(I_0, I_{res}|I_t)$ is the distribution of the clean image $I_0$ and the residual components $I_{res}$. $m_1^{I_0+I_{res}}$ is considered as a generalized absolute distance loss because it measures the mean absolute error between the predicted values and ground-truth values.

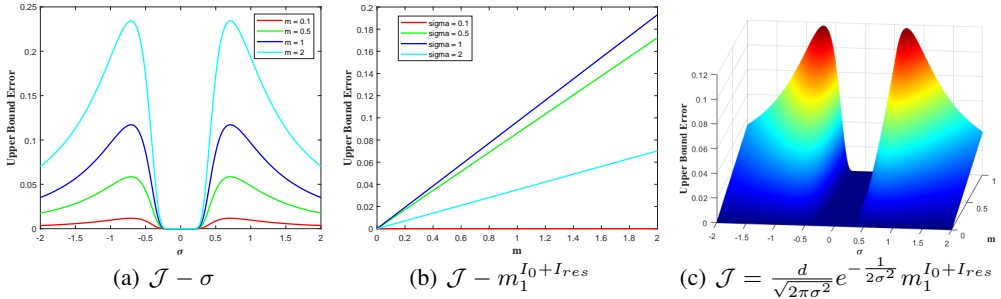

(a) $\mathcal{J} - \sigma$     (b) $\mathcal{J} - m_1^{I_0+I_{res}}$     (c) $\mathcal{J} = \frac{d}{\sqrt{2\pi\sigma^2}} e^{-\frac{1}{2\sigma^2}} m_1^{I_0+I_{res}}$

Figure A1: Relationships among the upper bound error $\mathcal{J}$ and its variables $\sigma$ and $m = m_1^{I_0+I_{res}}$.

Notably, $m_1^{I_0+I_{res}}$ is finite for most of the distribution in practice once the predicted networks are optimized well. It can be regarded as generalized absolute distance between the observed reference and estimated data. The Jensen gap $\mathcal{J}(f, m_1^{I_0+I_{res}})$ can approach to 0 as $\sigma \to 0$ or $\infty$, suggesting that the approximation error reduces with extreme measurement noise. Specifically, if the predictions of $\widehat{I_0}$ and $\widehat{I}_{res}$ are accurate, the upper bound $\mathcal{J}(\sigma, m_1^{I_0+I_{res}})|_{\sigma \to 0, m_1^{I_0+I_{res}} \to 0}$ shrinks due to the low variance and distortion. Oppositely, if the predictions lack accuracy, the upper bound of the $\mathcal{J}(\sigma, M)|_{\sigma \to \infty, m_1^{I_0+I_{res}} \to \max(m_1^{I_0+I_{res}})}$ shrinks as well for large variance and limited distortion. We further illustrate the relationships among the upper bound error $\mathcal{J}$ and its variables in Fig. A1. Evidently, the results support the aforementioned analysis. For the universal measurement model $I_{in} = I_0 + I_{res} + n$, we can derive the probability formulation as:

$$q(I_{in}|\widehat{I_0}, \widehat{I}_{res}) \sim \mathcal{N}(I_{in}|I_0 + I_{res}, \sigma^2 \boldsymbol{I}) = \frac{1}{\sqrt{2\pi\sigma^2}} exp[-\frac{(I_{in} - (I_0 + I_{res}))^2}{2\sigma^2}], \quad (200)$$

Consequently, the formulation of universal posterior sampling is:

$$\nabla_{I_t} \log q(I_{in}|I_t) \simeq \nabla_{I_t} \log q(I_{in}|\widehat{I_0}, \widehat{I}_{res}) = -1/\sigma^2 \nabla_{I_t} \|I_{in} - (I_0 + I_{res}^\theta(I_t, I_{in}, t))\|, \quad (201)$$

according to the DPS [7], the value of $\sigma^2$ varies with time $t$ and can be calculated as $\|I_{in} - (I_0 + I_{res}^\theta(I_t, I_{in}, t))\|$.

# D   Diffusion Generalist Solvers with Universal Posterior Sampling

The inverse process ordinary differential equations of diffusion generalist model are:

$$\frac{dI_t}{dt} = \frac{d\overline{\alpha}_t}{dt} I_{res} - \frac{d\overline{\delta}_t}{dt} I_{in} - \frac{1}{2}\frac{d\overline{\beta}_t^2}{dt} \nabla \log q_t(I_t). \tag{202}$$

According to the Bayes' theorem, we can derive:

$$\nabla_{I_t} \log q(I_t|I_{in}) = \nabla_{I_t} \log \frac{q(I_t)q(I_{in}|I_t)}{q(I_{in})} = \nabla_{I_t} \log q(I_t) + \nabla_{I_t} \log q(I_{in}|I_t), \tag{203}$$

where $\nabla_{I_t} \log P(I_t)$ is the score representing the gradient direction of inverse process and the discarded item $P(I_{in})$ is irrelevant to $I_t$. By incorporating the universal posterior sampling as conditional guidance, we can get:

$$\begin{aligned}
\frac{dI_t}{dt} &= \frac{d\overline{\alpha}_t}{dt} I_{res} - \frac{d\overline{\delta}_t}{dt} I_{in} - \frac{1}{2}\frac{d\overline{\beta}_t^2}{dt}(\nabla_{I_t} \log q_t(I_t) + \nabla_{I_t} \log q(I_{in}|I_t)) \\
&= \frac{d\overline{\alpha}_t}{dt} I_{res} - \frac{d\overline{\delta}_t}{dt} I_{in} - \frac{1}{2}\frac{d\overline{\beta}_t^2}{dt}(-\frac{\epsilon_\theta(I_t, I_{in}, t)}{\overline{\beta}_t} - 1/\sigma^2 \nabla_{I_t}\|I_{in} - (I_0 + I_{res}^\theta(I_t, I_{in}, t))\|) \\
&= \frac{d\overline{\alpha}_t}{dt} I_{res} - \frac{d\overline{\delta}_t}{dt} I_{in} + \frac{d\overline{\beta}_t}{dt}(\epsilon_\theta(I_t, I_{in}, t) + \overline{\beta}_t/\sigma^2 \nabla_{I_t}\|I_{in} - (I_0 + I_{res}^\theta(I_t, I_{in}, t))\|).
\end{aligned} \tag{204}$$

Eq. (204) is the detailed description of our method. Based on the proposed algorithms in Alg. 2, Alg. 3 and Alg. 4, we can further present varied orders with universal posterior sampling, as depicted in Alg. 5, Alg. 6 and Alg. 7. From the ablation experiment results in the paper, our method has the best balance between restoration performance and efficiency when $k = 2$ and universal posterior sampling is activated. Therefore, we customize an accelerated algorithm for this configuration.

---

**Algorithm 5:** First-order Diffusion Generalist Solvers with Universal Posterior Sampling.

**Input:** $I_{in}, I_{res}^\theta, \epsilon_\theta, \{\overline{\alpha}_{t_i}\}, \{\overline{\beta}_{t_i}\}, \{\overline{\delta}_{t_i}\}, \{t_i\}_{i=0}^M$.

1  $I_{t_0} = I_T \sim \mathcal{N}(0, \overline{\beta}_{t_0}\boldsymbol{I})$
2  **for** $i = 1$ to $M$ **do**
3  $\quad \widehat{I}_{res} = I_{res}^\theta(I_{t_{i-1}}, I_{in}, t_{i-1})$
4  $\quad \widehat{\epsilon} = (I_{t_{i-1}} - (\overline{\alpha}_{t_{i-1}} - 1)\widehat{I}_{res} - (1 - \overline{\delta}_{t_{i-1}})I_{in})/\overline{\beta}_{t_{i-1}}$
5  $\quad \widehat{I}_0 = I_{t_{i-1}} - \overline{\alpha}_{t_{i-1}}\widehat{I}_{res} - \overline{\beta}_{t_{i-1}}\widehat{\epsilon} + \overline{\delta}_{t_{i-1}}I_{in}$
6  $\quad \widetilde{\epsilon} = \widehat{\epsilon} + (\overline{\beta}_{t_{i-1}}/\|I_{in} - (\widehat{I}_0 + \widehat{I}_{res})\|)\nabla_{I_{t_{i-1}}}\|I_{in} - (\widehat{I}_0 + \widehat{I}_{res})\|$
7  $\quad I_{t_i} = I_{t_{i-1}} - (\overline{\delta}_{t_i} - \overline{\delta}_{t_{i-1}})I_{in} + (\overline{\alpha}_{t_i} - \overline{\alpha}_{t_{i-1}})\widehat{I}_{res} + (\overline{\beta}_{t_i} - \overline{\beta}_{t_{i-1}})\widetilde{\epsilon}$

**Output:** $I_{t_M}$.

---

**Algorithm 6:** Second-order Diffusion Generalist Solvers with Universal Posterior Sampling.

**Input:** $I_{in}, I_{res}^\theta, \epsilon_\theta, \{\overline{\alpha}_{t_i}\}, \{\overline{\beta}_{t_i}\}, \{\overline{\delta}_{t_i}\}, \{t_i\}_{i=0}^M, r$.

1  $I_{t_0} \leftarrow I_T \sim \mathcal{N}(0, \overline{\beta}_{t_0}\boldsymbol{I})$
2  **for** $i = 1$ to $M$ **do**
3  $\quad t_u = rt_i + (1 - r)t_{i-1}$
4  $\quad \widehat{I}_{res}^{t_{i-1}} = I_{res}^\theta(I_{t_{i-1}}, I_{in}, t_{i-1})$
5  $\quad \widehat{\epsilon}_{t_{i-1}} = (I_{t_{i-1}} - (\overline{\alpha}_{t_{i-1}} - 1)\widehat{I}_{res}^{t_i} - (1 - \overline{\delta}_{t_{i-1}})I_{in})/\overline{\beta}_{t_{i-1}}$
6  $\quad \widehat{I}_0^{t_{i-1}} = I_{t_{i-1}} - \overline{\alpha}_{t_{i-1}}\widehat{I}_{res}^{t_{i-1}} - \overline{\beta}_{t_{i-1}}\widehat{\epsilon}_{t_{i-1}} + \overline{\delta}_{t_{i-1}}I_{in}$
7  $\quad \widetilde{\epsilon}_{t_{i-1}} = \widehat{\epsilon}_{t_{i-1}} + (\overline{\beta}_{t_{i-1}}/\|I_{in} - (\widehat{I}_0^{t_{i-1}} + \widehat{I}_{res}^{t_{i-1}})\|)\nabla_{I_{t_{i-1}}}\|I_{in} - \widehat{I}_0^{t_{i-1}} - \widehat{I}_{res}^{t_{i-1}}\|$
8  $\quad I_{t_u} = I_{t_i} - (\overline{\delta}_{t_i} - \overline{\delta}_{t_u})I_{in} + (\overline{\alpha}_{t_i} - \overline{\alpha}_{t_u})\widehat{I}_{res}^{t_i} + (\overline{\beta}_{t_i} - \overline{\beta}_{t_u})\widetilde{\epsilon}_{t_i}$
9  $\quad \widehat{I}_{res}^{t_u} = I_{res}^\theta(I_{t_u}, t_u, I_{in})$
10 $\quad \widehat{\epsilon}_{t_u} = (I_{t_u} - (\overline{\alpha}_{t_u} - 1)\widehat{I}_{res}^{t_u} - (1 - \overline{\delta}_{t_u})I_{in})/\overline{\beta}_{t_u}$
11 $\quad \widehat{I}_0^{t_u} = I_{t_u} - \overline{\alpha}_{t_u}\widehat{I}_{res}^{t_u} - \overline{\beta}_{t_u}\widehat{\epsilon}_{t_u} + \overline{\delta}_{t_u}I_{in}$
12 $\quad \widetilde{\epsilon}_{t_u} = \widehat{\epsilon}_{t_u} + (\overline{\beta}_{t_u}/\|I_{in} - (\widehat{I}_0^{t_u} + \widehat{I}_{res}^{t_u})\|)\nabla_{I_{t_u}}\|I_{in} - \widehat{I}_0^{t_u} - \widehat{I}_{res}^{t_u}\|$
13 $\quad I_{t_i} = I_{t_{i-1}} - (\overline{\delta}_{t_i} - \overline{\delta}_{t_{i-1}})I_{in} + (\overline{\alpha}_{t_i} - \overline{\alpha}_{t_{i-1}})\widehat{I}_{res}^{t_{i-1}} + (\overline{\beta}_{t_i} - \overline{\beta}_{t_{i-1}})\widetilde{\epsilon}_{t_{i-1}} + \frac{1}{2r}(\overline{\alpha}_{t_i} -$
$\quad \overline{\alpha}_{t_{i-1}})(\widehat{I}_{res}^{t_u} - \widehat{I}_{res}^{t_{i-1}}) + \frac{1}{2r}(\overline{\beta}_{t_i} - \overline{\beta}_{t_{i-1}})(\widetilde{\epsilon}_{t_u} - \widetilde{\epsilon}_{t_{i-1}})$

**Output:** $I_{t_M}$.

**Algorithm 7:** Third-order Diffusion Generalist Solver with Universal Posterior Sampling.

**Input:** $I_{in}, I_{res}^{\theta}, \epsilon_{\theta}, \{\overline{\alpha}_{t_i}\}, \{\overline{\beta}_{t_i}\}, \{\overline{\delta}_{t_i}\}, r_1, r_2$.

1   $I_{t_0} \leftarrow I_T \sim \mathcal{N}(0, \overline{\beta}_{t_0} \boldsymbol{I})$

2   **for** $i = 1$ to $M$ **do**

3     $t_u = r_1 t_i + (1 - r_1) t_{i-1}, t_s = r_2 t_i + (1 - r_2) t_{i-1}$

4     $\widehat{I}_{res}^{t_{i-1}} = I_{res}^{\theta}(I_{t_{i-1}}, I_{in}, t_{i-1}), \widehat{\epsilon}_{i-1} = (I_{t_{i-1}} - (\overline{\alpha}_{t_{i-1}} - 1) \widehat{I}_{res}^{t_{i-1}} - (1 - \overline{\delta}_{t_{i-1}}) I_{in}) / \overline{\beta}_{t_{i-1}}$,
      $\widehat{I}_0^{t_{i-1}} = I_{t_{i-1}} - \overline{\alpha}_{t_{i-1}} \widehat{I}_{res}^{t_{i-1}} - \overline{\beta}_{t_{i-1}} \widehat{\epsilon}_{i-1} + \overline{\delta}_{t_{i-1}} I_{in}$

5     $\widetilde{\epsilon}_{t_{i-1}} = \widehat{\epsilon}_{t_{i-1}} + (\overline{\beta}_{t_{i-1}} / \|I_{in} - (\widehat{I}_0^{t_{i-1}} + \widehat{I}_{res}^{t_{i-1}})\|) \nabla_{I_{t_{i-1}}} \|I_{in} - \widehat{I}_0^{t_{i-1}} - \widehat{I}_{res}^{t_{i-1}}\|$

6     $I_{t_u} = I_{t_i} - (\overline{\delta}_{t_i} - \overline{\delta}_{t_u}) I_{in} + (\overline{\alpha}_{t_i} - \overline{\alpha}_{t_u}) \widehat{I}_{res}^{t_i} + (\overline{\beta}_{t_i} - \overline{\beta}_{t_u}) \widetilde{\epsilon}_{t_i}$

7     $\widehat{I}_{res}^{t_u} = I_{res}^{\theta}(I_{t_u}, I_{in}, t_u), \widehat{\epsilon}_{t_u} = (I_{t_u} - (\overline{\alpha}_{t_u} - 1) \widehat{I}_{res}^{t_u} - (1 - \overline{\delta}_{t_u}) I_{in}) / \overline{\beta}_{t_u}$,
      $\widehat{I}_0^{t_u} = I_{t_u} - \overline{\alpha}_{t_u} \widehat{I}_{res}^{t_u} - \overline{\beta}_{t_u} \widehat{\epsilon}_{t_u} + \overline{\delta}_{t_u} I_{in}$

8     $\widetilde{\epsilon}_{t_u} = \widehat{\epsilon}_{t_u} + (\overline{\beta}_{t_u} / \|I_{in} - (\widehat{I}_0^{t_u} + \widehat{I}_{res}^{t_u})\|) \nabla_{I_{t_u}} \|I_{in} - \widehat{I}_0^{t_u} - \widehat{I}_{res}^{t_u}\|$

9     $I_{t_s} = I_{t_{i-1}} - (\overline{\delta}_{t_s} - \overline{\delta}_{t_{i-1}}) I_{in} + (\overline{\alpha}_{t_s} - \overline{\alpha}_{t_{i-1}}) \widehat{I}_{res}^{t_{i-1}} + (\overline{\beta}_{t_s} - \overline{\beta}_{t_{i-1}}) \widetilde{\epsilon}_{t_{i-1}} + \frac{r_2}{2r_1} (\overline{\alpha}_{t_s} -$
      $\overline{\alpha}_{t_{i-1}}) (\widehat{I}_{res}^{t_u} - \widehat{I}_{res}^{t_{i-1}}) + \frac{r_2}{2r_1} (\overline{\beta}_{t_s} - \overline{\beta}_{t_{i-1}}) (\widetilde{\epsilon}_{t_u} - \widetilde{\epsilon}_{t_{i-1}})$

10    $\widehat{I}_{res}^{t_s} = I_{res}^{\theta}(I_{t_s}, I_{in}, t_s), \widehat{\epsilon}_{t_s} = (I_{t_s} - (\overline{\alpha}_{t_s} - 1) \widehat{I}_{res}^{t_s} - (1 - \overline{\delta}_{t_s}) I_{in}) / \overline{\beta}_{t_s}$,
      $\widehat{I}_0^{t_s} = I_{t_s} - \overline{\alpha}_{t_s} \widehat{I}_{res}^{t_s} - \overline{\beta}_{t_s} \widehat{\epsilon}_{t_s} + \overline{\delta}_{t_s} I_{in}$

11    $\widetilde{\epsilon}_{t_s} = \widehat{\epsilon}_{t_s} + (\overline{\beta}_{t_s} / \|I_{in} - (\widehat{I}_0^{t_s} + \widehat{I}_{res}^{t_s})\|) \nabla_{I_{t_s}} \|I_{in} - \widehat{I}_0^{t_s} - \widehat{I}_{res}^{t_s}\|$

12    $D_1^{res} = \widehat{I}_{res}^{t_u} - \widehat{I}_{res}^{t_{i-1}}, D_1^{\epsilon} = \widetilde{\epsilon}_{t_u} - \widetilde{\epsilon}_{t_{i-1}}$

13    $D_2^{res} = \frac{2}{r_1 r_2 (r_2 - r_1)} (r_1 \widehat{I}_{res}^{t_s} - r_2 \widehat{I}_{res}^{t_u} + (r_2 - r_1) \widehat{I}_{res}^{t_{i-1}})$

14    $D_2^{\epsilon} = \frac{2}{r_1 r_2 (r_2 - r_1)} (r_1 \widetilde{\epsilon}_{t_s} - r_2 \widetilde{\epsilon}_{t_u} + (r_2 - r_1) \widetilde{\epsilon}_{t_{i-1}})$

15    $I_{t_i} = I_{t_{i-1}} - (\overline{\delta}_{t_i} - \overline{\delta}_{t_{i-1}}) I_{in} + (\overline{\alpha}_{t_i} - \overline{\alpha}_{t_{i-1}}) \widehat{I}_{res}^{t_{i-1}} + (\overline{\beta}_{t_i} - \overline{\beta}_{t_{i-1}}) \widetilde{\epsilon}_{t_{i-1}} + \frac{1}{2r_1} (\overline{\alpha}_{t_i} -$
      $\overline{\alpha}_{t_{i-1}}) D_1^{res} + \frac{1}{2r_1} (\overline{\beta}_{t_i} - \overline{\beta}_{t_{i-1}}) D_1^{\epsilon} + \frac{1}{6} (\overline{\alpha}_{t_i} - \overline{\alpha}_{t_{i-1}}) D_2^{res} + \frac{1}{6} (\overline{\beta}_{t_i} - \overline{\beta}_{t_{i-1}}) D_2^{\epsilon}$

**Output:** $I_{t_M}$.

---

**Algorithm 8:** Queue-Based Second-order Solvers with Universal Posterior Sampling.

**Input:** $I_{in}, I_{res}^{\theta}, \epsilon_{\theta}, \{\overline{\alpha}_{t_i}\}, \{\overline{\beta}_{t_i}\}, \{\overline{\delta}_{t_i}\}, \{t_i\}_{i=0}^{M}, r$.

1   $I_{t_0} \leftarrow I_T \sim \mathcal{N}(0, \overline{\beta}_{t_0} \boldsymbol{I})$,

2   $R = [\,]$,

3   $(I_{t_1}, \widehat{I}_{res}^{t_1}, \widetilde{\epsilon}_{t_1}, \widehat{I}_0^{t_1}) = 1^{st}$-order Solver$(t_1)$ in Alg. 5

4   $R = \text{queue\_push}(I_{t_1}, \widehat{I}_{res}^{t_1}, \widetilde{\epsilon}_{t_1}, \widehat{I}_0^{t_1}), i = t_1$

5   **while** $i < t_M$ **do**

6     $t_p = t_{i-1}, t_u = t_i, t_q = t_{i+1}, r = (t_u - t_p) / (t_q - t_p)$

7     $(I_{t_p}, \widehat{I}_{res}^{t_p}, \widetilde{\epsilon}_{t_p}, \widehat{I}_0^{t_p}) = \text{queue\_pop}(R, t_p)$

8     $(I_{t_u}, \widehat{I}_{res}^{t_u}, \widetilde{\epsilon}_{t_u}, \widehat{I}_0^{t_u}) = 1^{st}$-order Solver$(t_u)$ in Alg. 5

9     $I_{t_q} = I_{t_p} - (\overline{\delta}_{t_q} - \overline{\delta}_{t_p}) I_{in} + (\overline{\alpha}_{t_q} - \overline{\alpha}_{t_p}) \widehat{I}_{res}^{t_p} + (\overline{\beta}_{t_q} - \overline{\beta}_{t_p}) \widehat{\epsilon}_{t_p} + \frac{1}{2r} (\overline{\alpha}_{t_q} - \overline{\alpha}_{t_p}) (\widehat{I}_{res}^{t_u} - \widehat{I}_{res}^{t_p}) +$
      $\frac{1}{2r} (\overline{\beta}_{t_q} - \overline{\beta}_{t_p}) (\widetilde{\epsilon}_{t_u} - \widetilde{\epsilon}_{t_p})$

10   $R = \text{queue\_push}(I_{t_u}, \widehat{I}_{res}^{t_u}, \widetilde{\epsilon}_{t_u}, \widehat{I}_0^{t_u}), i = t_q$

**Output:** $I_{t_M}$.

**Queue-Based Accelerated Strategy.** The introduction of variables pertaining to the intermediate time point $t_u$ incurs additional computational overhead, making the process inefficient. To alleviate this problem, we construct a queue-based accelerated strategy by collecting the previous solutions, as illustrated in Fig. 3 in the paper. Specifically, for the initial sampling from $t_{i-1}$ to $t_i$, we introduce the intermediate step $t_u$ to assist the approximation at time point $t_i$. Subsequently, $t_i$ is regarded as the intermediate time point of the sampling from $t_u$ to $t_{i+1}$ iteratively. In this context, the sampling steps are controlled by the predefined parameters $r$ without introducing extra calculation. Relevant implementation details are summarized in Alg. 8.

# E  Experiment Supplementary

## E.1  Summary about the Datasets

We evaluate the proposed method on five natural image restoration tasks, including deraining, low-light enhancement, desnowing, dehazing, and deblurring. We select the most widely used datasets for each task, as summarized in Tab. A1.

Table A1: Summary of the natural image restoration datasets utilized in this paper.

| Task | Dataset | Synthetic/Real | Train samples | Test samples |
|---|---|---|---|---|
| **Deraining** | DID [69] | Synthetic | - | 1,200 |
| | DeRaindrop [43] | Real | 861 | 307 |
| | Rain13K [17] | Synthetic | 13,711 | - |
| | Rain_100H [66] | Synthetic | - | 100 |
| | Rain_100L [66] | Synthetic | - | 100 |
| | GT-Rain [3] | Real | 26,125 | 2,100 |
| | RealRain-1k-H [24] | Real | 896 | 224 |
| | RealRain-1k-L [24] | Real | 896 | 224 |
| **Low-light Enhancement** | LOL [61] | Real | 485 | 15 |
| | MEF [36] | Real | - | 17 |
| | VE-LOL-L [30] | Synthetic/Real | 900/400 | 100/100 |
| | NPE [57] | Real | - | 8 |
| **Desnowing** | CSD [4] | Synthetic | 8,000 | 2,000 |
| | Snow100K-Real [31] | Real | - | 1,329 |
| **Dehazing** | SOTS [21] | Synthetic | - | 500 |
| | ITS_v2 [21] | Synthetic | 13,990 | - |
| | D-HAZY [8] | Synthetic | 1,178 | 294 |
| | NH-HAZE [2] | Real | - | 55 |
| | Dense-Haze [1] | Real | - | 55 |
| | NHRW [70] | Real | - | 150 |
| **Deblur** | GoPro [39] | Synthetic | 2,103 | 1,111 |
| | RealBlur [46] | Real | 3,758 | 980 |

**Image deraining.** We use the merged datasets mentioned in Rain13K [17] and DeRaindrop [43] as training parts, which cover a wide range of different rain streaks and rain densities. We test our model on both deraining and deraindrop scenes using the mixed datasets [66, 69, 43]. Furthermore, we perform zero-shot generalization on real-world datasets such as GT-Rain [3], RealRain-1k-H [24], and RealRain-1k-L [24].

**Low-light enhancement.** We utilize the LOL [61] and VE-LOL-L [30] datasets with real and synthetic paired images as the benchmark, which consist of a large number of indoor and outdoor scenes with different levels of light and noise. Furthermore, we generalize our method on the MEF [36] dataset with images of multiple exposures to conduct the compound restoration experiments.

**Image desnowing.** We use the CSD [4] dataset as the desnowing benchmark and employ the Snow100K-Real [31] to perform real-world generalization.

**Image dehazing.** We use the ITS_v2 [21] and D-HAZY [8] datasets as the dehazing benchmark, which are widely used synthetic fog datasets with different levels of fog and various scenes. Besides, the outdoor fog dataset SOTS [21] is selected for quantitative evaluations. The real-world fog datasets Dense-Haze [1], NHRW [70], and NH-HAZE [2] are utilized for generalization validation on real-world scenes.

**Image deblurring.** We use the GoPro [39] dataset as the deblurring benchmark, which contains 2,103 training pairs and 1,111 testing pairs. The dataset contains various levels of blur obtained by averaging the clear images captured in very short intervals. To further validate the generalization ability of our model, we perform zero-shot generalization on the RealBlur [46] dataset.

## E.2  More Visual Comparisons on Comparable Experiments

We show the visualization results of other degradation categories in Fig. A2 to further demonstrate our superiority. Evidently, Our method generates more stable image samples with high fidelity than other universal image restoration methods. Benefiting from the accuracy of diffusion generalist solvers and stable guidance of universal posterior sampling, we achieve the outstanding reconstruction of the missing details.

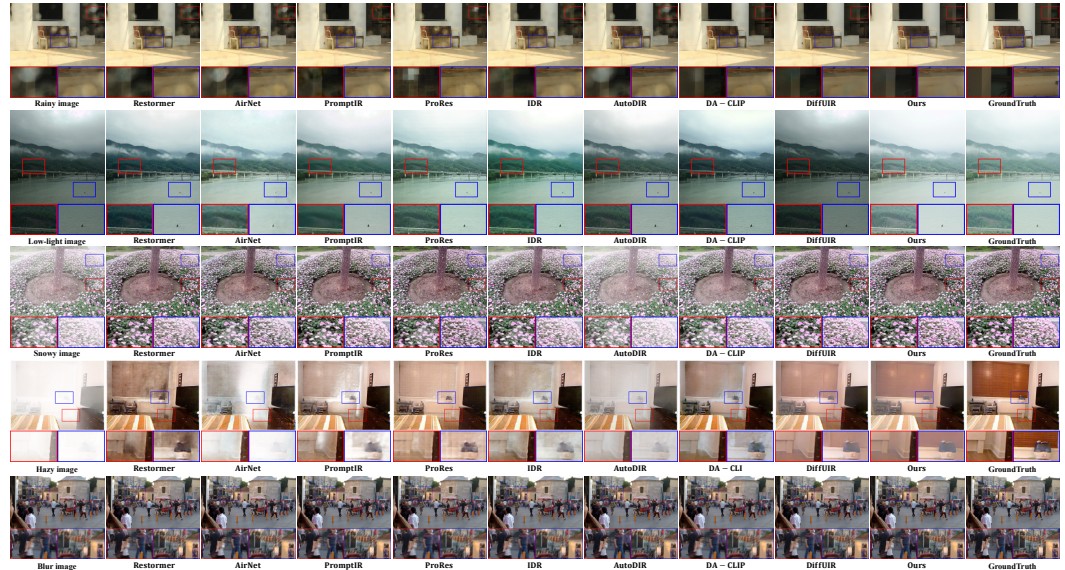

Figure A2: Visualization results of other degradation categories.

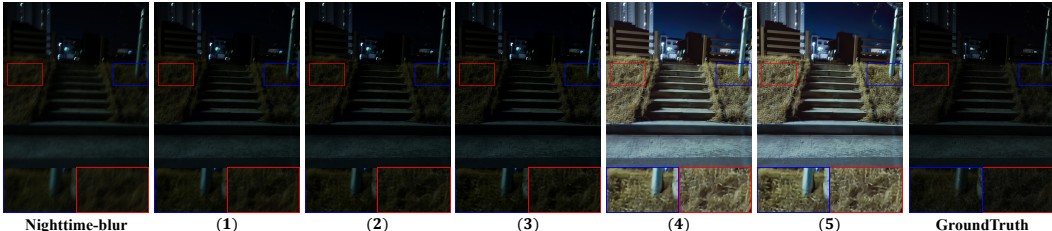

Figure A3: Results of low-light enhancement and deblurring for five iterations on RealBlur [46].

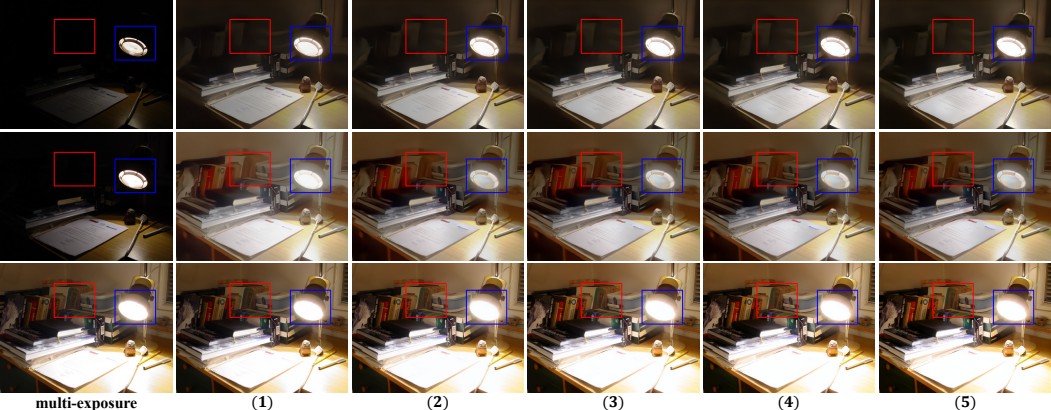

Figure A4: Results of multi-exposure restoration for five iterations on MEF [36].

### E.3 Compound Restoration on Real-world Scenes

To demonstrate that our method can handle the more complex restoration tasks that various degradation occurs in one image. We generalize our well-optimized model on RealBlur [46] datasets for low-light enhancement and deblurring, and NHRW [70] datasets for low-light enhancement and dehazing, and MEF [36] datasets for multi-exposure restoration. Specifically, we repeatedly perform our method on the degraded images for five times, and the visual comparisons are displayed in Fig. A3, Fig. 6 and Fig. A4, respectively. As a result, as our method is trained on multiple degraded datasets, it is capable of excellently accomplishing relevant composite tasks. Furthermore, it possesses the ability to eliminate the former degradation and subsequently the latter

degradation within its capability range. For well-restored samples, our method incurs no additional information interference, thus ensuring the high fidelity of the restored images.

## E.4 Real-world Scene Generalization

Except for the desnowing on the real-world scene, other restoration results of different degradation types are illustrated in Fig. A5. We utilize the RealRain-1k [24] dataset for deraining, NPE [57] dataset for low-light enhancement, Snow100k_real [31] dataset for desnowing, Dense-Haze [1] dataset for dehazing, and RealBlur [46] dataset for deblurring. The results show that our method can generalize to real-world scenes and achieve competitive visual results.

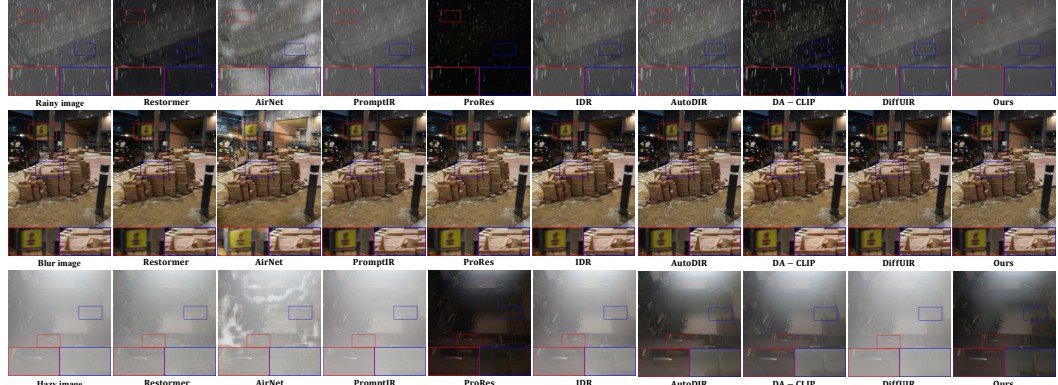

Figure A5: Visual effects of zero-shot generalization on real-world image restoration.

## E.5 Generalization on Remote Sensing

Table A2: Summary of the utilized remote sensing image restoration datasets.

| Task | Dataset | Synthetic/Real | Train samples | Test samples |
|---|---|---|---|---|
| **Cloud Removal** | Sen2_MTC [15] | Real | 9,784 | 467 |
| **Denoise** | AID [63] | Synthetic | 38,173 | 1,827 |
| **Deblur** | NWPU-RESISC45 [5] | Synthetic | 10,000 | 1,094 |
| **Super-resolve** | Second [65] | Synthetic | 22,603 | 1,141 |
| **Light Enhancement** | PatternNet [76] | Synthetic | 28,905 | 1,495 |

We further validate the scalability of our method on remote sensing image restoration tasks, including cloud removal, denoising, deblurring, super-resolution, and low-light enhancement. Except for the cloud removal task, there are no existing real-world datasets for other restoration tasks [44], so we adopt a synthetic data generation approach to simulated the datasets, as presented in Tab. A2. Specifically: **i) Cloud Removal:** We use the Sen2_MTC dataset [15] as the benchmark, which contains paired cloudy and cloud-free images. **ii) Denoising:** Due to the lack of real-world noisy remote sensing images, we synthesize a noisy dataset based on the AID dataset [63], considering three mainstream noise types (Gaussian noise, stripe noise, and speckle noise) and their random combinations. **iii) Deblurring:** We primarily simulate Gaussian blur, motion blur, and defocus blur, and conduct experiments on the NWPU-RESISC45 dataset [5]. **iv) Super-Resolution:** We generate low-resolution images using a four-fold downsampling degradation model with Gaussian blur and construct paired images on the Second dataset [65]. **v) Low-Light Enhancement:** We simulate low-light imaging conditions on the PatternNet dataset [76] through linear transformation, gamma correction, and additive Gaussian noise. The visual comparisons of each method are presented in Fig. A6 and Fig. 8, and quantitative evaluations are reported in Tab. 5 of this paper. All methods have been re-implemented except for AutoDIR, as no training code has been released for it. Evidently, our visual effect surpasses that of others and achieves the best metric evaluations.

Table A4: Time efficiency of different strategies.

| Steps | | 1 | | 2 | | 3 | | 4 | | 5 | | 6 | | 7 | | 8 | | 9 | | 10 |
|---|---|---|---|---|---|---|---|---|---|---|---|---|---|---|---|---|---|---|---|---|
| | NFE | Time(s) | NFE | Time(s) | NFE | Time(s) | NFE | Time(s) | NFE | Time(s) | NFE | Time(s) | NFE | Time(s) | NFE | Time(s) | NFE | Time(s) | NFE | Time(s) |
| Naive(k=1) | 1 | 0.063 | 2 | 0.129 | 3 | 0.194 | 4 | 0.263 | 5 | 0.322 | 6 | 0.401 | 7 | 0.466 | 8 | 0.529 | 9 | 0.596 | 10 | 0.661 |
| Naive (k=2) | 2 | 0.120 | 4 | 0.247 | 6 | 0.393 | 8 | 0.523 | 10 | 0.644 | 12 | 0.769 | 14 | 0.910 | 16 | 0.989 | 18 | 1.147 | 20 | 1.251 |
| Queue (k=2) | 2 | 0.127 | 3 | 0.191 | 4 | 0.262 | 5 | 0.320 | 6 | 0.399 | 7 | 0.476 | 8 | 0.533 | 9 | 0.598 | 10 | 0.659 | 11 | 0.728 |
| Naive (k=3) | 3 | 0.201 | 6 | 0.401 | 9 | 0.578 | 12 | 0.769 | 15 | 0.977 | 18 | 1.169 | 21 | 1.322 | 24 | 1.446 | 27 | 1.585 | 30 | 1.814 |
| Queue (k=3) | 3 | 0.197 | 4 | 0.258 | 5 | 0.318 | 6 | 0.410 | 7 | 0.479 | 8 | 0.528 | 9 | 0.594 | 10 | 0.657 | 11 | 0.719 | 12 | 0.806 |

Table A3: Efficiency comparisons among universal methods. '-' means out of memmory.

| Method | 256×256 | | | 512×512 | | | 1024×1024 | | |
|---|---|---|---|---|---|---|---|---|---|
| | Mem.(G) | Time(s) | FPS | Mem.(G) | Time(s) | FPS | Mem.(G) | Time(s) | FPS |
| Restomer | 1.959 | 0.105 | 9.563 | 6.670 | 0.381 | 2.622 | 25.419 | 1.773 | 0.564 |
| AirNet | 1.039 | 0.194 | 5.159 | 3.480 | 0.738 | 1.355 | 11.244 | 20.499 | 0.049 |
| PromptIR | 2.544 | 0.111 | 8.981 | 7.255 | 0.399 | 2.508 | 26.005 | 1.845 | 0.542 |
| ProRes | 2.027 | 0.318 | 3.149 | 2.514 | 0.766 | 1.305 | 6.025 | 1.715 | 0.583 |
| IDR | 1.340 | 0.052 | 19.253 | 4.313 | 0.136 | 7.373 | 16.110 | 0.615 | 1.626 |
| AutoDIR | 7.023 | 6.266 | 0.160 | 11.021 | 11.986 | 0.083 | − | − | − |
| DA-CLIP | 2.119 | 2.585 | 0.387 | 6.775 | 7.937 | 0.126 | 58.548 | 60.893 | 0.016 |
| DiffUIR(n=3) | 1.563 | 0.118 | 8.450 | 3.528 | 0.206 | 4.862 | 18.060 | 0.911 | 1.098 |
| Ours-L(n=3) | 1.561 | 0.112 | 8.908 | 3.528 | 0.199 | 5.014 | 18.059 | 0.907 | 1.103 |
| **Naive (n=8), $k=1$, ×UPS** | | | | | | | | | |
| Ours-T | 0.777 | 0.277 | 3.605 | 2.291 | 0.385 | 2.594 | 15.306 | 1.705 | 0.587 |
| Ours-S | 0.787 | 0.290 | 3.450 | 2.300 | 0.401 | 2.494 | 15.316 | 1.755 | 0.570 |
| Ours-B | 0.942 | 0.291 | 3.431 | 2.907 | 0.492 | 2.033 | 17.438 | 2.280 | 0.439 |
| Ours-L | 1.562 | 0.293 | 3.407 | 3.527 | 0.529 | 1.889 | 18.058 | 2.402 | 0.416 |
| **Naive (n=8), $k=1$, ✓UPS** | | | | | | | | | |
| Ours-T | 1.764 | 0.500 | 1.998 | 5.793 | 0.829 | 1.206 | 32.581 | 4.454 | 0.225 |
| Ours-S | 1.905 | 0.502 | 1.993 | 6.265 | 0.865 | 1.156 | 34.237 | 4.545 | 0.220 |
| Ours-B | 2.762 | 0.520 | 1.923 | 9.645 | 1.078 | 0.928 | 41.357 | 5.770 | 0.173 |
| Ours-L | 3.613 | 0.535 | 1.868 | 10.593 | 1.144 | 0.874 | 43.815 | 6.022 | 0.166 |
| **Naive (n=8), $k=2$, ×UPS** | | | | | | | | | |
| Ours-T | 0.784 | 0.535 | 1.869 | 2.297 | 0.717 | 1.395 | 15.313 | 3.194 | 0.313 |
| Ours-S | 0.794 | 0.548 | 1.824 | 2.308 | 0.747 | 1.339 | 15.323 | 3.296 | 0.303 |
| Ours-B | 0.948 | 0.555 | 1.802 | 2.913 | 0.924 | 1.082 | 17.444 | 4.279 | 0.234 |
| Ours-L | 1.563 | 0.571 | 1.751 | 3.527 | 0.989 | 1.011 | 18.059 | 4.522 | 0.221 |
| **Naive (n=8), $k=2$, ✓UPS** | | | | | | | | | |
| Ours-T | 1.770 | 0.974 | 1.027 | 5.800 | 1.559 | 0.641 | 32.587 | 8.150 | 0.123 |
| Ours-S | 1.933 | 0.987 | 1.013 | 7.508 | 1.871 | 0.535 | 33.586 | 10.482 | 0.095 |
| Ours-B | 2.803 | 0.990 | 1.010 | 9.688 | 2.089 | 0.479 | 41.399 | 11.276 | 0.089 |
| Ours-L | 3.829 | 1.019 | 0.981 | 10.679 | 2.219 | 0.451 | 44.154 | 11.723 | 0.085 |
| **Queue (n=8), $k=2$, ×UPS** | | | | | | | | | |
| Ours-T | 0.780 | 0.303 | 3.301 | 2.294 | 0.431 | 2.321 | 15.310 | 1.905 | 0.524 |
| Ours-S | 0.791 | 0.321 | 3.115 | 2.304 | 0.460 | 2.174 | 15.320 | 1.980 | 0.505 |
| Ours-B | 0.946 | 0.323 | 3.096 | 2.911 | 0.547 | 1.823 | 17.442 | 2.573 | 0.388 |
| Ours-L | 1.563 | 0.324 | 3.086 | 3.527 | 0.598 | 1.673 | 18.059 | 2.694 | 0.371 |
| **Queue (n=8), $k=2$, ✓UPS** | | | | | | | | | |
| Ours-T | 1.771 | 0.557 | 1.795 | 5.800 | 0.925 | 1.081 | 32.587 | 5.012 | 0.200 |
| Ours-S | 1.920 | 0.561 | 1.782 | 6.279 | 0.961 | 1.041 | 34.250 | 5.101 | 0.196 |
| Ours-B | 2.892 | 0.590 | 1.695 | 9.677 | 1.201 | 0.833 | 41.388 | 6.472 | 0.155 |
| Ours-L | 3.816 | 0.600 | 1.667 | 10.601 | 1.273 | 0.786 | 44.074 | 6.741 | 0.148 |
| **Naive (n=8), $k=3$, ×UPS** | | | | | | | | | |
| Ours-T | 0.790 | 0.759 | 1.317 | 2.303 | 1.053 | 0.950 | 15.319 | 4.687 | 0.213 |
| Ours-S | 0.799 | 0.775 | 1.290 | 2.312 | 1.099 | 0.910 | 15.328 | 4.836 | 0.207 |
| Ours-B | 0.954 | 0.780 | 1.283 | 2.919 | 1.347 | 0.742 | 17.450 | 6.253 | 0.160 |
| Ours-L | 1.562 | 0.801 | 1.248 | 3.527 | 1.446 | 0.692 | 18.058 | 6.586 | 0.152 |
| **Naive (n=8), $k=3$, ✓UPS** | | | | | | | | | |
| Ours-T | 1.782 | 1.414 | 0.707 | 5.812 | 2.338 | 0.428 | 32.599 | 12.384 | 0.081 |
| Ours-S | 1.958 | 1.428 | 0.700 | 6.318 | 2.474 | 0.404 | 34.288 | 13.238 | 0.076 |
| Ours-B | 2.909 | 1.433 | 0.698 | 9.694 | 3.100 | 0.323 | 41.405 | 16.755 | 0.060 |
| Ours-L | 4.048 | 1.496 | 0.668 | 11.050 | 3.302 | 0.303 | 48.273 | 17.461 | 0.057 |
| **Queue (n=8), $k=3$, ×UPS** | | | | | | | | | |
| Ours-T | 0.777 | 0.338 | 2.959 | 2.290 | 0.481 | 2.077 | 15.306 | 2.132 | 0.469 |
| Ours-S | 0.787 | 0.364 | 2.744 | 2.301 | 0.502 | 1.991 | 15.316 | 2.195 | 0.455 |
| Ours-B | 0.942 | 0.366 | 2.730 | 2.907 | 0.615 | 1.626 | 17.438 | 2.845 | 0.351 |
| Ours-L | 1.562 | 0.369 | 2.710 | 3.527 | 0.657 | 1.522 | 18.058 | 2.999 | 0.333 |
| **Queue (n=8), $k=3$, ✓UPS** | | | | | | | | | |
| Ours-T | 1.765 | 0.624 | 1.602 | 5.794 | 1.060 | 0.943 | 32.581 | 5.664 | 0.177 |
| Ours-S | 1.907 | 0.646 | 1.547 | 6.267 | 1.090 | 0.917 | 34.238 | 5.789 | 0.173 |
| Ours-B | 2.765 | 0.661 | 1.512 | 9.648 | 1.369 | 0.730 | 41.359 | 7.339 | 0.136 |
| Ours-L | 3.613 | 0.695 | 1.440 | 10.593 | 1.470 | 0.680 | 43.814 | 7.656 | 0.131 |

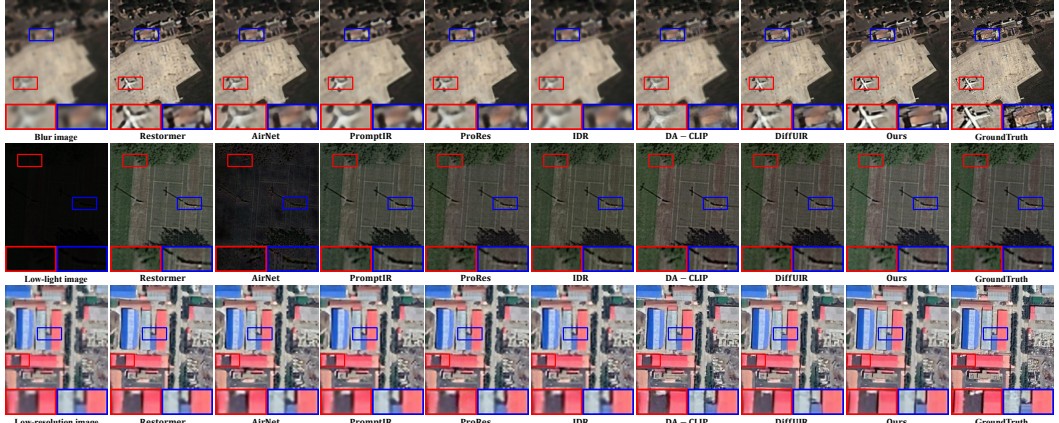

Figure A6: Visual results of remote sensing image restoration.

## E.6 Efficiency Comparisons

For fairness, we collect and mix the datasets used by comparison methods, with image resolution ranging from $256$ to $1024$ pixels. Accordingly, we evaluate model efficiency under three representative resolution settings, as summarized in Tab. A3. Let $k$ denote the solver order and $n$ be sampling steps. Obviously, our method (n = 3) and baseline DiffUIR remain competitive in computational cost and efficiency. As $n$ increases, memory consumption remains stable while time cost grows proportionally. Activating UPS that requires gradient backpropagation introduces per-step computational overhead and additional memory usage. Hence, we adopt the queue-based sampling strategy that significantly improves efficiency by reducing the computational complexity from $O(nk)$ to $O(k + n - 1)$. Tab. A4 also proves that queue-based solver achieves approximately a 2× efficiency improvement over naive solvers when $k = 2$, and around a 3× improvement when $k = 3$.

## F Discussions, Limitations, and Future Work

**Limitations and broader impact.** The main challenge lies in improving the precision and robustness of a unified restoration model when handling highly mixed representations. While we have theoretically proposed a general formulation for a k-order solver, our experiments are limited to first-, second-, and third-order solvers due to GPU memory constraints. This is because unified posterior sampling requires storing the gradient map during inference to compute $\nabla_{I_t} \log q(I_{in}|\widehat{I}_0, \widehat{I}_{res}^{\theta})$ and more time points are needed to compute derivatives. In addition, it is observed that the second order solver with universal posterior sampling reach the performance bottleneck. However, it remains unknown whether it will be saturated in more complex scenes or using larger models. Besides, our solver and strategy are training-free and can potentially be extended to other methods after appropriate modifications. Future work could explore more efficient network architectures to reduce memory usage, enabling larger batch sizes, higher-order solvers, and better task-specific training strategies, such as adaptive learning rate schedules. Despite current limitations, we believe our unified model offers a strong foundation for advancing image restoration.

**Future Work.** There are several promising directions to enhance our method: (1) With the rise of high-resolution imagery (e.g., 4K, 8K) driven by advances in sensor technology, developing an interpretable multi-dimensional latent diffusion model is crucial to address the computational demands that current architectures struggle with. (2) Apply model distillation to reduce memory overhead and parameter size. (3) Introduce additional single or compound degradation types, expand parameter ranges, and enhance model versatility. (4) Design adaptive learning rate schedules to reduce sampling steps and improve restoration quality.

