# OpenReview forum: "DGSolver: Diffusion Generalist Solver with Universal Posterior Sampling for Image Restoration"
_NeurIPS.cc/2025/Conference — NeurIPS 2025 poster_

### Official Review · Reviewer_92Km · 2025-06-21

**Clarity:** 3
**Significance:** 3
**Originality:** 3
**Rating:** 5
**Confidence:** 4

**Summary:**

The paper proposes a training-free solver for diffusion generalist models, termed as DGSolver. First, it reformulates the diffusion process as ODEs and exploits the semi-linear integral structure to better approximate the solutions. This avoids the computational overhead of training a series of networks for quality refinement. Second, it integrates the estimated residual term I_{res} into diffusion posterior sampling to construct the universal sampling mechanism, which guides the sampling process universally without the prior knowledge of degradation formulation. These two components enhance the restoration accuracy and efficiency across a variety of degradation tasks. Overall, the proposed method demonstrates notable theoretical novelty, practical applicability, and broad adaptability.

**Questions:**

1.How do you choose the sampling steps or variable $t$ in the reverse process? What are the relationships between time step $t$ and predefined parameters $\alpha(t)$, $\beta(t)$, and $\gamma(t)$ in Eq.(4)? I think it is important for readers to understand the paper and implementation.

2.Please provide the computational complexity analysis before and after applying the queue-based sampling strategy.

3.When dealing with compound degradation, the model can restore degraded images in a specific order, which seems to be determined spontaneously by the unified model. Could the authors share some insights on achieving controllable restoration in the given order or handling all degradation types in one execution for compound degraded images based on the proposed method?

**Ethical Concerns:**

["NO or VERY MINOR ethics concerns only"]

**Final Justification:**

After further discussion with the author, I still maintain my acceptance decision.

**Limitations:**

Yes, the authors have discussed the related limitations.

**Quality:**

3

**Strengths And Weaknesses:**

**Strengths**

1.This paper presents a rigorously derived and technically sound ODE integrator with accelerated sampling strategy for accurate sampling from diffusion models to achieve quality refinement, which can be widely exploited and benefited by other related methods.

2.The proposed UPS is novel and clever, and does not require the prior knowledge or estimation of degradation formulation. It can cope with various types of degradation without complicated designs with high potential.

3.All the framework is well motivated and the explanations are clear. The authors effectively provide a theoretical derivation and error analysis of the proposed method.

4.Comprehensive experiments are carried out on multiple datasets, various tasks, and different domains, fully demonstrating the proposed method’s superior performance over state-of-the-art methods both quantitatively and qualitatively.

**Weaknesses**

1.The expressions for $\alpha(t)$, $\beta(t)$, and $\gamma(t)$ in Eq.(4) lack clarity.

2.Some of the notations in the appendix lack definitions and some typos need to be corrected (e.g., $I_\text{res}^{\theta}$ is misspelled in Appendix F). It would be worthwhile to spend some effort to make that better.

---

> ### Author Rebuttal · Authors · 2025-07-30
>
> ### W1&Q1: Experssion and selection of noise schedule
>
> Following the linear noise schedule, the formulations of $\bar{\alpha}_t,\bar{\beta}_t,\bar{\delta}_t$ are:
>
> $\beta_t = \beta_{min} + (\beta_{max} - \beta_{min})\frac{t}{T},\beta_{min} = 0.0001,\beta_{max} = 0.02$,
>
> $\delta_t = \delta_{min} + (\delta_{max} - \delta_{min})\frac{t}{T},\delta_{min} = 1\times 10^{-6},\delta_{max}=0.002$
>
> $\bar{\beta}_t = \sum_0^t \beta_i$, $\bar{\alpha}_t = 1 - \sqrt{\prod_0^t (1 - \beta_i)}$, $\bar{\delta}_t = \sum_0^t \delta_i$
>
> Notably, the total timesteps $T=1000, t\in\mathcal{Z}, 0\le t < T$. Since $\beta_t$ and $\delta_t$ are non-negative within the range of [0,1], three variables $\bar{\beta}_t,\bar{\alpha}_t,\bar{\delta}_t$ exhibit strict monotonic increase.
>
> ### W2: Typos
>
> We sincerely appreciate your meticulous review, and we will correct the misspelled $I_{res}^{\theta}$ in revision. We promise to revise other typos for readability.
>
> ### Q2: Complexity of naive and queue-based sampling strategies
>
> We utilize neural function evaluations (NFEs) to evaluate the computational complexity. Let $k$ denote the solver order and $n$ be sampling steps. For a $k$-order solver, naive sampling from time $s$ to $t$ requires interpolating $k-1$ intermediate points within the interval, resulting in $k$ NFEs per sampling step. Consequently, the overall computational complexity for $n$ steps is $O(nk)$. In contrast, the queue-based sampling strategy precomputes the values at (k−1) intermediate time points and caches them for reuse in subsequent steps. This reduces the total complexity to $O(k−1+n)$, offering a clear computational advantage for multi-step sampling. For quantitative results, please refer to Tab. r1 in Reviewer (9bRU-W1) and Tab. r3 in Reviewer (1zc3-W2&Q3).
>
>
> ### Q3: Insights about controllable restoration.
>
> Our core idea is to map different degradations into a shared representation space, and then leverage high-order solvers to enhance reconstruction accuracy, the queue-based sampling to improve efficiency, and the UPS mechanism to provide effective guidance for restoration. Hence, our model handles compound degradation through a unified framework, but the restoration order is indeed implicitly learned during training rather than explicitly controlled. Below we clarify some insights and potential extensions: (i) Add task-specific prompts as controllable factors to condition the solver. For example, we can mine degradation priors $c$ either from the image itself or from foundation models, and train model $I_{res}^{\theta}(I_t,I_{in},t,c)$ using paired data with restoration order. With learnable parameters, image restoration becomes controllable. (ii) Project the gradient guidance $\nabla \log p(I_{in}|I_t)$ onto orthogonal subspaces for degradation-specific restoration. For instance, we assign orthogonal bases $ v_i, i=1,..,k $ for $k$ degradation types during training, and utilize the UPS term to project gradients $\sum_i^k \theta_i P_{v_i}\nabla \log p(I_{in}|I_t)$, where $\theta_i$ is a switch for activating the $i$-th degradation subspace and $P_{v_i}$ is a projection operator. This facilitates controllable and interpretable restoration. We sincerely appreciate your insightful comments, and we will conduct further research in our future work.

---

> > ### Comment · Reviewer_92Km · 2025-08-03
> >
> > I appreciate the authors’ thoughtful reply, which has resolved my concerns. Having considered the other reviewers’ feedback, I continue to support acceptance of this paper.

---

> > > ### Author Response · Authors · 2025-08-03
> > >
> > > We sincerely appreciate your valuable review and kind support for the acceptance of our paper. Your constructive comments have significantly contributed to improving our work.

---

### Official Review · Reviewer_iqQ1 · 2025-06-23

**Clarity:** 2
**Significance:** 3
**Originality:** 3
**Rating:** 4
**Confidence:** 3

**Summary:**

This paper presents DGSolver, a novel framework designed to balance the commonality of degradation representations with restoration quality in diffusion models. DGSolver mitigates accumulated discretization errors through tailored high-order solvers with a universal posterior sampling strategy, and further enhances sampling efficiency by introducing a queue-based accelerated sampling strategy.

**Questions:**

1.The Universal Posterior Sampling (UPS) strategy demonstrates clear performance benefits in this work. Could UPS also be applied to other existing methods?

2.Is the queue-based sampling strategy affecting model performance while improving efficiency?

3.Does increasing the order of the high-order solvers influence the efficiency of inference?

**Ethical Concerns:**

["NO or VERY MINOR ethics concerns only"]

**Final Justification:**

I will retain my original recommendation.

**Limitations:**

The limitations have been discussed in the supplementary materials.

**Quality:**

3

**Strengths And Weaknesses:**

Strengths:

1. The paper provides a thorough and rigorous derivation of the proposed method, demonstrating a solid theoretical foundation.

2. The proposed high-order solvers and universal posterior sampling strategy are intuitively illustrated in the figure, effectively clarifying their fundamental principle.

3. Experimental results demonstrate the effectiveness of the proposed method from qualitative, quantitative perspectives.

Weaknesses:

1.The ablation study should quantitatively compare model parameters, sampling steps, and inference time with baseline DiffUIR[1] to better demonstrate the superiority of the proposed method.

2.Lack of quantitative comparison with the latest state-of-the-art methods proposed in 2025.

3.Table 4 shows that increasing the number of sampling steps beyond a certain point leads to a decline in PSNR/SSIM. Could the authors provide insight into this counterintuitive result?

4.Why were the experimental settings (datasets and training parameters) from the baseline DiffUIR not adopted?

5.The motivation for designing the forward process as a stochastic differential equation (SDE) is unclear. Its practical significance remains questionable, as no explicit analysis or justification is provided in the experiments.

6.The expression “total M+1 time steps” in line 146 is inconsistent with the notation presented in Figure 2.

7.In Figure 5, the position of blue boxes is inconsistent.

8.In Tables 2, 3, and 4, the best results should be highlighted in bold.

---

> ### Author Rebuttal · Authors · 2025-07-30
>
> ### W1: Ablation and comparison with DiffUIR
>
>
> We sincerely appreciate this constructive suggestion. We perform a detailed efficiency comparison with baseline DiffUIR, as presented in Tab. r7. Both methods maintain identical model parameters (7.73M), confirming our enhancements require no additional network capacity. Our method outperforms the baseline in terms of performance, but the overall time consumption increases. For a fair comparison, we apply our solver to DiffUIR for inference process, and the results show a significant improvement in restoration quality, as presented in Tab. r2 in Reviewer (9bRU-W3). This verifies our solvers can be integrated into related methods to enhance restoration accuracy without retraining. For detailed efficiency comparison, please refer to Tab. r1 in Reviewer (9bRU-W1).
>
>
> |Metric|DiffUIR|Ours (k=1)|Ours (k=1+UPS)|Ours (k=2)|Ours (k=2+UPS)|Ours (k=3)|Ours (k=3+UPS)|
> |-|-|-|-|-|-|-|-|
> |Param.(M)|7.73|7.73|7.73|7.73|7.73|7.73|7.73|7.73|
> |Steps|3|8|8|8|8|8|8|8|8|
> |Time (s)|0.206|0.529|1.144| 0.598 |1.273|0.657|1.470|
> |Average PSNR (dB)|29.93|30.70|31.01|31.20|31.36|31.23|31.36|
> |Average SSIM |0.907|0.913|0.918|0.919|0.920|0.919|0.920|
>
> *Table r7: Ablation and comparison with DiffUIR*
>
>
> ### W2: Comparisons with 2025 SOTAs
>
> For a fair comparison, we reimplemented and retrained several representative universal image restoration models, namely AwRaCLe[1], MaIR[2], and DeepSNNet[3], on our collected dataset using their released code and official settings. The quantitative results are presented in Tab. r8. Evidently, our method outperforms others. Thanks for your constructive comments, and we will include these results in revision.
>
> |Method|Year|Deraining||Enhancement||Desnowing||Dehazing||Deblurring||Average||Complexity||
> |-|-|-|-|-|-|-|-|-|-|-|-|-|-|-|-|
> |||PSNR|SSIM|PSNR|SSIM|PSNR|SSIM|PSNR|SSIM|PSNR|SSIM|PSNR|SSIM|Params(M)|FLOPs(G)|
> |DA-CLIP|2024|28.63|0.854|19.50|0.730|28.23|0.934|27.26|0.941|26.47|0.818|27.54|0.881|32.96|158.14|
> |DiffUIR|2024|30.67|0.887|21.21|0.769|30.70|0.943|30.29|0.944|29.00|0.877|29.93|0.907|7.73|32.93|
> |AwRaCLe[1]|2025|29.42|0.868|22.22|0.773|28.90|0.944|24.60|0.883|27.54|0.846|27.94|0.888|94.18|165.42|
> |MaIR[2]|2025|29.17|0.859|20.09|0.726|28.85|0.931|26.79|0.936|26.74|0.812|27.91|0.880|20.71|110.44|
> |DeepSNNet[3]|2025|28.97|0.847|17.84|0.659|30.17|0.927|28.00|0.939|25.80|0.768|28.20|0.865|17.31|71.79|
> |Ours|-|31.46|0.896|23.84|0.801|32.69|0.955|31.68|0.946|30.15|0.899|31.36|0.920|7.73|32.93|
>
> *Table r8: Quantitative comparisons with latest methods*
>
>
> [1]Rajagopalan S, Patel V M. AWRaCLe: All-weather image restoration using visual in-context learning[C]. AAAI, 2025.
>
>
> [2]Li B, Zhao H, Wang W, et al. Mair: A locality-and continuity-preserving mamba for image restoration[C]. CVPR, 2025.
>
>
> [3]Deng X, Zhang C, Jiang L, et al. DeepSN-Net: Deep semi-smooth Newton driven network for blind image restoration[J]. TPAMI, 2025.
>
>
> ### W3: Counter-intuitive phenomenon between performance and sampling steps
>
> Thanks for your insightful observation. We agree that the non-monotonic performance trend in Tab. 4 appears unintuitive at first glance. This counter-intuitive phenomenon may stem from two main factors. (1) Our approach to image restoration is motivated by the principle of commonality, aiming to handle diverse degradation types within a unified framework. In Appendix E.2, we discuss the model's behavior under compound degradation scenarios. It can be observed that across multiple sampling steps, the model tends to first remove the primary degradation before addressing secondary ones. Consequently, in cases where samples from the deraining or deblurring datasets also suffer from additional degradations (e.g., low-light conditions), the restored output may diverge from the available reference that typically reflects only the removal of the primary degradation, leading to reduced evaluation metrics despite better perceptual quality. (2) The stochasticity in both the forward and reverse processes, combined with network estimated errors, can cause slight influence on performance within an acceptable range.
>
> ### W4: Datasets setting
>
> Our task setting is closely aligned with that of DiffUIR. However, since each method adopts different dataset configurations for image restoration, we attempt to collect and standardize all the datasets used by these methods for fairness, enabling direct and meaningful comparisons. Besides, we strictly follow their open-source code and retrain all of them except autodir.
>
> ### W5: SDEs motivation
>
> The motivation behind adopting forward SDEs is that SDEs offer a broader perspective for analyzing the transition of probability distributions in a continuous manner. As discussed in Appendices A and B, both the perturbed process and deterministic implicit sampling can be viewed as first-order instances of the reverse-time SDE and ODE, respectively. Therefore, defining the diffusion process from the perspective of probability distribution in Eq. (4) is fundamentally equivalent to describing it through continuous SDEs in Eq. (5). Their performance comparisons can be found in Tab. 2, where the results for $k=1$ without UPS correspond to deterministic implicit sampling, while our high-order solvers demonstrate improved performance. Besides, SDEs formulation offers higher scalability, enabling the integration of our high-order solver and UPS, which are tightly coupled with the dynamics of the SDEs.
>
> ### W6,7,8: Figure and table errors
>
> Thank you for your careful observations regarding inconsistencies in notation, figures, and formatting. We will correct the inconsistency in the description of Figure 2, as well as the misaligned highlighted regions in Figure 5. In addition, Tables 2, 3, and 4 will be revised to include clearly marked highlights for improved readability. We will thoroughly check and address all similar issues in the revision for clarity and accuracy.
>
> ### Q1: Generalization on other methods
>
> Thank you for the thoughtful question. Within the domain of image restoration, UPS can indeed be generalized to related frameworks such as DiffUIR. We apply UPS to DiffUIR, which yields performance gains, as shown in Tab. r2 in Reviewer (9bRU-W3). Broadly speaking, UPS depends on residual information from given distribution to provide gradient guidance for diffusion reverse process. It can be seamlessly extended to other tasks wherein both prior and target data distributions are accessible, such as image translation. Therefore, we believe UPS exhibits strong scalability and generalizability. We leave a more comprehensive exploration of UPS in other vision tasks as promising future work.
>
> ### Q2: Ablating sampling strategies on performance
>
> In Appendix B.2, we theoretically show that cumulative error is positively correlated with the sampling interval $\Delta t$ . For a k-order solver, naive sampling from time $s$ to $t$ requires interpolating $k−1$ intermediate points, whereas queue-based sampling leverages precomputed points cached in a queue. As a result, the effective sampling interval in the queue-based strategy is larger than that of naive sampling, which may lead to higher theoretical error. To validate this, we conduct experiments, and the results are shown in Tab. r9. Performance of queue-based sampling is slightly lower than naive sampling. However, queue-based approach significantly reduces neural function evaluations (NFEs), achieving 2–3× higher efficiency than naive sampling (please refer to Tab. r1 in Reviewer (9bRU-W1) ).
>
>
> |Sampling|Order|UPS|Deraining||Enhancement||Desnowing||Dehazing||Deblurring||Average||
> |-|-|-|-|-|-|-|-|-|-|-|-|-|-|-|
> |-|-|-|PSNR|SSIM|PSNR|SSIM|PSNR|SSIM|PSNR|SSIM|PSNR|SSIM|PSNR|SSIM|
> |Naive|k=2|×|31.33|0.894|23.10|0.799|32.52|0.956|31.64|0.948|30.09|0.898|31.22|0.919|
> |Queue|k=2|×|31.31|0.894|23.08|0.798|32.51|0.955|31.62|0.948|30.07|0.897|31.20|0.919|
> |Naive|k=2|√|31.47|0.897|23.85|0.802|32.70|0.955|31.69|0.946|30.17|0.899|31.37|0.920|
> |Queue|k=2|√|31.46|0.896|23.84|0.801|32.69|0.955|31.68|0.946|30.15|0.899|31.36|0.920|
> |Naive|k=3|×|31.35|0.895|23.12|0.800|32.55|0.956|31.65|0.948|30.11|0.898|31.24|0.920|
> |Queue|k=3|×|31.33|0.894|23.13|0.799|32.52|0.955|31.71|0.948|30.09|0.898|31.23|0.919|
> |Naive|k=3|√|31.50|0.897|23.87|0.803|32.72|0.955|31.70|0.946|30.19|0.900|31.39|0.920|
> |Queue|k=3|√|31.45|0.896|23.84|0.801|32.69|0.955|31.67|0.946|30.15|0.899|31.36|0.920|
>
> *Table r9: Performance for differnent $k$ with different strategies*
>
>
> ### Q3: Solver order and inference efficiency
>
> We use neural function evaluations (NFEs) to measure inference efficiency. Let $k$ denote the solver order and $n$ be sampling steps. For a $k$-order solver, naive sampling from time $s$ to $t$ requires interpolating $k-1$ intermediate points within the interval, resulting in $k$ NFEs per step. Consequently, the overall computational complexity for $n$ steps is $O(nk)$. In contrast, the queue-based sampling strategy precomputes the values at $k−1$ time points and caches them for reuse in subsequent steps, offering high efficiency by reducing total complexity to $O(k−1+n)$. Therefore, Both naive solvers and queue-based solvers have the same time complexity $O(k)$ only when $n=1$. When $n > 1$, the naive solvers are increasingly less efficient than queue-based solvers because $nk \ge k-1+n$. Under the queue-based sampling strategy, the difference in NFEs among different solvers equals the difference in their orders. As a result, their efficiency gap scales linearly. For quantitative results, please refer to Tab. r1 in Reviewer (9bRU-W1) and Tab. r3 in Reviewer (1zc3-W2&Q3).

---

> ### Comment · Reviewer_iqQ1 · 2025-08-06
>
> I would like to thank the authors for their comprehensive rebuttal. I will retain my original recommendation.

---

> > ### Author Response · Authors · 2025-08-07
> >
> > We sincerely appreciate your constructive feedback and acceptance recommendation. Your insights have greatly strengthened our work. Best regards.

---

### Official Review · Reviewer_Y9z3 · 2025-07-03

**Clarity:** 2
**Significance:** 3
**Originality:** 3
**Rating:** 4
**Confidence:** 4

**Summary:**

The paper proposes DGSolver, a training-free diffusion-based solver for image restoration that unifies a custom reverse-time ODE, high-order solvers, and universal posterior correction. It introduces a semi-linear forward process using a residual-based degradation model and derives tailored solvers to efficiently recover clean images.

**Questions:**

1.	Section 3.3 appears heavily influenced by Diffusion Posterior Sampling (DPS), yet the original paper is not cited or discussed in this section. Can the authors clarify what is novel in their Universal Posterior Sampling (UPS) approach beyond the omission of an explicit measurement operator? Additionally, why is the original DPS work not acknowledged in the relevant discussion?
2.	The results in Table 4 feel a bit unintuitive to me. It’s not clear why using exactly 8 steps leads to better performance than 9 or 10. I would normally expect performance to improve as the number of sampling steps increases. Is there a natural explanation for this behavior?
3.	Can the authors comment on sample variety?

**Ethical Concerns:**

["NO or VERY MINOR ethics concerns only"]

**Final Justification:**

The authors have satisfactorily addressed my concerns, and I find the paper to offer a meaningful and well-supported contribution that warrants acceptance.

**Limitations:**

Yes, the authors have addressed the limitations in the Appendix F.

**Paper Formatting Concerns:**

None.

**Quality:**

3

**Strengths And Weaknesses:**

**Strengths:**
1. The use of high-order solvers tailored to the semi-linear structure of the reverse process introduces a novel and principled way to reduce the number of sampling steps in diffusion-based restoration.
2. The solver design is clearly presented, with a queue-based implementation that enables efficient computation of higher-order derivatives without redundant evaluations.
3. Evaluation is very thorough and shows improvement over comparison methods.
4. Ablation studies are well-designed and support the impact of key components such as solver order and universal posterior correction (UPC).

**Weaknesses:**
1. The main ideas are valuable but somewhat difficult to follow due to dense presentation and unconventional notation. A more streamlined explanation with clearer separation of key components would improve accessibility.
2. The forward process relies on the definition $I_{res}=I_{in}-I_0$, which implies additive degradations in aligned domains. While appropriate for denoising-like tasks, this formulation does not generalize to inverse problems such as super-resolution, MRI, or operator-unknown deblurring, where the degradation is either non-additive or defined in a different domain. Given that $I_{res}$ plays a central role in both the forward and reverse processes, this assumption limits the applicability of DGSolver to a narrow set of problems, despite its claim to universality.

Minor comments:
- The blue zoomed-in insets in Figure 5 appear inconsistent as the regions shown for the rainy input and ground truth differ from those in the restored results.
- Line 94, “…computationally intractability...” -> computationally intractable
- Line 111, “when” -> When

---

> ### Author Rebuttal · Authors · 2025-07-30
>
> ### W1: Steamlined explanation of our key components
>
> We thank the reviewer for pointing out the issue with presentation clarity. We will add a more streamlined method description for readability in Sec. 3. Simplified contents can be concluded as: _The overview of our DGSolver is illustrated in Fig. 2. In the forward process, diffusion generalist SDEs map degradations into a shared, degradation-agnostic latent space. In the reverse process, our DGSolver integrates diffusion generalist solvers and universal posterior sampling to jointly enhance inference accuracy and stability, with respect to red and blue trajectories in Fig. 2. The former component reduces cumulative error by solving a semi-linear ODE, thereby guiding the solution toward the ground truth; the latter component further optimizes the solutions through gradient guidance along the learned data manifold._"
>
> ### W2: Residual Formulation and Degradation
>
> From a degradation modeling perspective, residual-based and kernel-based approaches emphasize additive degradation restoration and inverse problem solving, respectively, with each carrying inherent limitations. However, we respectfully clarify that our use of residual modeling is not tied to a strict pixel-wise alignment assumption, but instead rooted in a signal decomposition perspective. From this broader view, residual decomposition offers a more general and widely applicable framework, extensively adopted across various image restoration tasks, such as super-resolution, deblurring, and denoising [1,2,3]. In contrast, kernel estimation often suffers from ill-posedness and increased complexity, making the decomposition less stable and kernel estimation harder to solve. In our DGSolver, residual modeling is seamlessly embedded into the diffusion generalist SDEs, enabling the unified design of both high-order solvers and the UPS mechanism. As a result, UPS becomes a plug-and-play component for diffusion-based methods that adopt residual modeling, whereas kernel-based approaches still depend on explicit kernel priors or estimation. That said, we acknowledge the limited capacity of residual modeling to generalize across all inverse problems. For instance, as shown in Tab. 2, UPS yields more substantial improvements in deraining (i.e., 0.15 dB (k=2), 0.12 dB (k=3)) than in deblurring (i.e., 0.07 dB (k=2), 0.06dB (k=3)). We believe that kernel-based modeling may perform well in specific inverse problems, but its applicability to broader restoration scenarios remains somewhat constrained, especially in the compound restoration tasks.
>
> In a broad sense, our residual-based modeling exhibits the potential to be applied beyond the realm of image restoration. It can be seamlessly extended to other tasks wherein both prior and target data distributions are accessible, such as image translation.
>
> [1] Zhang Y, Li K, Li K, et al. Residual non-local attention networks for image restoration[J]. ICLR, 2019.
>
> [2] Liang J, Cao J, Sun G, et al. Swinir: Image restoration using swin transformer[C]. CVPR, 2021.
>
> [3] Cui Y, Ren W, Cao X, et al. Revitalizing convolutional network for image restoration[J]. TPAMI, 2024.
>
>
> ### W3: Typos and figure errors
>
> We appreciate your attention to detail regarding typos and figure inconsistencies. We will revise "computationally intractability" to “…computationally intractable” in line 94 and capitalize “when” to “When” in line 111. Besides, we will ensure that all zoomed-in regions are spatially aligned across input, ground truth, and restored images in Fig. 5. All these issues will be thoroughly checked and addressed in the revision to ensure clarity and accuracy.
>
> ### Q1: Disccusion about UPS and DPS
>
> We thank the reviewer for pointing this out. We sincerely apologize for overlooking the discussion and connection to DPS in Section 3.3. We fully acknowledge and appreciate that UPS is partially inspired by DPS, and in fact, we have dedicated Section 2.2 to introducing the theoretical foundations and related methods of DPS. We will explicitly include a discussion in Section 3.3. The simplified addition is as follows: “_DPS[1] circumvents the intractability of posterior sampling in diffusion models via a novel approximation, which is generally applicable to noisy inverse problems. Inspired by this, we propose universal posterior sampling from the perspective of residual modeling._”
>
> Both UPS and DPS aim to incorporate gradient guidance into the diffusion sampling process. By omitting the measurement operator, our residual modeling is generalizable across compound degradations. Beyond omission of an explicit measurement operator, we believe UPS can be a plug-and-play component for diffusion models in field of image restoration. Given that the prior distribution (e.g. degraded images) and data distribution (e.g. high-quality images) are both available, we believe future variants of diffusion model will increasingly integrate residual prior into their formulation. In this context, UPS can be seamlessly coupled into their reverse process. Moreover, UPS can be seamlessly extended to other tasks wherein both prior and target data distributions are accessible, such as image translation. In contrast, kernel-based models are often limited by their formulation. In future work, we plan to explore the adaptability of UPS in other tasks.
>
> [1] Chung H, Kim J, Mccann M T, et al. Diffusion Posterior Sampling for General Noisy Inverse Problems[C]. ICLR, 2023.
>
> ### Q2: Performance and sampling steps
>
> Thanks for your insightful observation. We agree that the non-monotonic performance trend in Tab. 4 appears unintuitive at first glance. This counter-intuitive phenomenon may stem from two main factors. First, the stochasticity in both the forward and reverse processes, combined with network approximation errors, can cause slight performance fluctuations across neighboring sampling steps, where the variations are within an acceptable range. Second, since we project all degradations into a shared degradation-agnostic representation and train the model on mixed degradation types, the model implicitly learns to handle compound degradations, as discussed in Appendix E.2. Consequently, when the number of sampling steps increases, the model may begin to address secondary degradations present in the image. For example, some samples in the deraining or deblurring datasets, also exhibit low-light conditions. After removing the primary degradation, the model may perform low-light enhancements, where the available ground truth typically reflects only the removal of the primary degradation. This mismatch means that additional sampling steps do not necessarily bring the output closer to the reference, and may even result in a slight drop in evaluation metrics.
>
> ### Q3: Sample variety
>
> Thanks for raising the concerns about sample variety. In field of image restoration, the primary focus is on achieving high-fidelity and consistent restoration results, rather than promoting sample diversity. Our work is therefore motivated by fully leveraging diffusion models to recover a unique, high-quality target, as opposed to sampling from a distribution of plausible outputs. As a result, sample variety is expected to be low. However, we still consider that our sample variety may arise from two sources: (i) the randomness of initial states sampled from the Gaussian prior, and (ii) the model’s capacity to handle compound degradations. For (i), we conduct a thorough evaluation across 10 random seeds, reporting the average and variances of metrics in Tab. r6. The results indicate strong robustness, with highly consistent outcomes across seeds. Notably, the use of UPS further stabilizes the inference process and reduces variance. For (ii), we conduct generalization experiments on compound degradations in Appendix E.2. Since the initial states are degradation-agnostic representations, more sampling steps allow the model to progressively eliminate secondary degradations, resulting in diverse yet plausible restorations.
>
> |DGSolver(k=2)|Deraining||Enhancement||Desnowing||Dehazing||Deblurring||Average||
> |-|-|-|-|-|-|-|-|-|-|-|-|-|
> |UPS|PSNR|SSIM|PSNR|SSIM|PSNR|SSIM|PSNR|SSIM|PSNR|SSIM|PSNR|SSIM|
> |x|31.31 $\pm$ 0.00118|0.894 $\pm$ 0.000004|23.08 $\pm$ 0.00180|0.798 $\pm$ 0.00008|32.51 $\pm$ 0.00068|0.955 $\pm$ 0.00001|31.62 $\pm$ 0.00655|0.948 $\pm$ 0.000012|30.07 $\pm$ 0.000291|0.897 $\pm$ 0.000014|31.20 $\pm$ 0.00118|0.919 $\pm$ 0.000001|
> |√|31.46 $\pm$ 0.00008|0.896 $\pm$ 0.000002|23.84 $\pm$ 0.00053|0.801 $\pm$ 0.00002|32.69 $\pm$ 0.00011|0.955 $\pm$ 0.00001|31.68 $\pm$ 0.00187|0.946 $\pm$ 0.000002|30.15 $\pm$ 0.000182|0.899 $\pm$ 0.000002|31.36 $\pm$ 0.000125|0.920 $\pm$ 0.000001|
>
> *Table r6: Performance of our DGSolver with different seeds*

---

> > ### Comment · Reviewer_Y9z3 · 2025-08-05
> > **Official Comment by Reviewer Y9z3**
> >
> > I would like to thank the authors’ for their clear and well-structured rebuttal, which effectively addressed my concerns. After reading the other reviews as well as the clarifications and additional experiments provided, I believe the paper makes a meaningful contribution and merits acceptance. I will therefore retain my original recommendation.

---

> > > ### Author Response · Authors · 2025-08-05
> > >
> > > We are truly grateful for your insightful review and the generous support for our paper's acceptance. Your constructive and valuable feedback has greatly enhanced the quality of our work.

---

### Official Review · Reviewer_1zc3 · 2025-07-03

**Clarity:** 2
**Significance:** 2
**Originality:** 2
**Rating:** 4
**Confidence:** 4

**Summary:**

The paper introduces DGSolver, a novel diffusion generalist solver with universal posterior sampling for image restoration. It addresses the limitations of existing diffusion models in universal image restoration, which suffer from cumulative errors in reverse inference and the trade-off between degradation representation commonality and restoration quality. The authors derive exact ODEs solution for diffusion generalist models and develop high-order solvers with a queue-based accelerated sampling strategy to enhance accuracy and efficiency. Additionally, they integrate universal posterior sampling to improve noise estimation and correct errors in inverse inference. Extensive experiments demonstrate that DGSolver outperforms state-of-the-art methods in restoration accuracy, stability, and scalability across various tasks and datasets.

**Questions:**

1. What challenges or opportunities arise from applying the generalist diffusion process reformulated with ODEs and customized high-order solvers, which are common in image generation, to the field of image restoration?
2. Given that k=2 with UPS already achieves near-optimal performance, what is the practical benefit of higher-order solvers (k=3) in terms of restoration quality versus computational cost? Can you provide a trade-off analysis (e.g., FLOPs vs. PSNR) for different k?
3. The paper mentions the use of a queue-based accelerated sampling strategy. Could this method remain effective when using fewer sampling steps? Typically, high-order solvers are designed to accelerate sampling with fewer steps.
4. The optimal $\bar{\delta}_T=1$ is used for complex degradations, but does this generalize to all tasks? For example, does deraining benefit from a different $\bar{\delta}_T$ than deblurring?

**Ethical Concerns:**

["NO or VERY MINOR ethics concerns only"]

**Final Justification:**

The authors' response during the rebuttal phase has addressed my concerns.

**Limitations:**

yes

**Quality:**

3

**Strengths And Weaknesses:**

### Strengths:
1. The reformulation of the generalist diffusion process using ODEs and the development of customized high-order solvers with a queue-based accelerated sampling strategy are novel contributions to the field of image restoration.
2. The integration of universal posterior sampling provides a versatile and effective way to enhance the accuracy of noise estimation and correct errors, leading to improved restoration quality.
3. The proposed method is validated through experiments on multiple image restoration tasks, such as deraining, low-light enhancement, etc., and shows superior performance over existing methods.

### Weaknesses:
1. The reformulation of the generalist diffusion process using ODEs and the development of customized high-order solvers have been extensively explored in image generation, and their application to image restoration is a natural extension.
2. The paper proposes a queue-based accelerated sampling strategy. However, its effectiveness might be limited in few-step sampling scenarios, and the paper lacks ablation studies to verify its performance in such cases.
3. While $\bar{\delta}_T$ controls degradation commonality, the paper does not systematically explore its effect on specific degradation types.
4. Although zero-shot results on real-world datasets are presented, the evaluation lacks quantitative metrics for unpaired real data, relying solely on visual comparisons

---

> ### Author Rebuttal · Authors · 2025-07-30
>
> ### W1&Q1: Challenges and opportunities arise from image restoration ODEs solvers
>
> To the best of our knowledge, in the field of image generation, high-order solvers are primarily employed to accelerate the sampling process from random noise to image samples, with the only requirement being that the generated samples conform to the target distribution. There is no strict constraint enforcing a one-to-one correspondence between samples from the prior and those from the data distribution. Consequently, various solvers can be employed to approximate the data distribution without explicit constraints, and the potential of high-order terms remains largely underexplored due to the weak coupling between the prior and data. In contrast, image restoration involves a strong dependency between the prior (i.e., the degraded image) and the target data (i.e., the high-quality image). Therefore, we argue that the key challenge and opportunity for designing solvers in image restoration lies in effectively exploiting the relationship between these two distributions and embedding this relationship throughout the diffusion process. To this end, we propose DGSolver. Our key insights are: (1) We fully incorporate the residuals derived from both the prior and data distributions into the high-order solver. (2) The potential of residuals in designed solvers is further exploited through a Universal Posterior Sampling (UPS), which reinforces the connection between the prior and data distributions, thereby enhancing restoration performance, inference stability, and sample fidelity.
>
> ### W2&Q3: Disscusion about queue-based sampling
>
> High-order solvers can reduce the sampling steps, but they do not necessarily reduce the number of neural function evaluations (NFEs) in few-step sampling scenarios. NFEs directly measure the number of neural network calls with respect to total runtime. For a $k$-order solver, naive sampling from time $s$ to $t$ requires interpolating $k-1$ intermediate points within the interval, resulting in $k$ NFEs per step. Consequently, the overall computational complexity for $n$ steps is $O(nk)$. In contrast, the queue-based sampling strategy precomputes the values at $k−1$ time points and caches them for reuse in subsequent steps, offering high efficiency by reducing total complexity to $O(k−1+n)$. Therefore, only when $n=1$ do both naive solvers and queue-based solvers have the same time complexity $O(k)$. When $n > 1$, the naive solvers are increasingly less efficient than queue-based solvers for $nk \ge k-1+n$.
>
> To verify that, we report runtime comparisons in Tab. r3 with a fixed image size of 512 and UPS deactivated. Apparently, queue-based strategy is more efficient than naive sampling. Additionally, as shown in Tab. r1 in Reviewer (9bRU-W1), enabling UPS further increases the computational cost, making the efficiency gap between the two strategies even more pronounced. Moreover, Tab. r9 in Reviewer (iqQ1-Q2) presents a performance comparison between the queue-based and naive strategies. Queue-based solvers possess a 2–3× speed advantage with comparable restoration quality to naive solvers.
>
> In conclusion, queue-based accelerated sampling still remains effective when using fewer sampling steps
>
>
> |Steps|1||2||3||4||5||6||7||8||9||10||
> |-|-|-|-|-|-|-|-|-|-|-|-|-|-|-|-|-|-|-|-|-|
> ||NFE|Time(s)|NFE|Time(s)|NFE|Time(s)|NFE|Time(s)|NFE|Time(s)|NFE|Time(s)|NFE|Time(s)|NFE|Time(s)|NFE|Time(s)|NFE|Time(s)|
> |(k=1)|1|0.063|2|0.129|3|0.194|4|0.263|5|0.322|6|0.401|7|0.466|8|0.529|9|0.596|10|0.661|
> |Naive(k=2)|2|0.120|4|0.247|6|0.393|8|0.523|10|0.644|12|0.769|14|0.910|16|0.989|18|1.147|20|1.251|
> |Queue(k=2)|2|0.127|3|0.191|4|0.262|5|0.320|6|0.399|7|0.476|8|0.533|9|0.598|10|0.659|11|0.728|
> |Naive(k=3)|3|0.201|6|0.401|9|0.578|12|0.769|15|0.977|18|1.169|21|1.322|24|1.446|27|1.585|30|1.814|
> |Queue(k=3)|3|0.197|4|0.258|5|0.318|6|0.410|7|0.479|8|0.528|9|0.594|10|0.657|11|0.719|12|0.806|
>
> *Table r3: Time efficiency of differnent strategies*
>
> ### W3&Q4: Discussion about $\delta_T$
>
> We appreciate your constructive suggestion. We respectfully clarify that performance comparisons under different values of $\overline{\delta}_T$ are presented in Tab. 3 (Page 8). As observed, the choice of $\overline{\delta}_T$ significantly influences restoration performance across different tasks, with higher values generally being more suitable. Specifically, the model achieves the best performance on deraining, low-light enhancement, and desnowing tasks with $\overline{\delta}_T = 1.0$; for dehazing, the optimal value is $\overline{\delta}_T = 0.9$; and for deblurring, $\overline{\delta}_T = 0.75$ yields the best results. Nevertheless, considering overall performance across tasks, $\overline{\delta}_T = 1.0$ emerges as the most favorable setting within our framework.
>
> ### W4: Real world quantitative evaluation
>
> To systematically evaluate the zero-shot capability of each method on real-world datasets (Appendix E, Tab. A1), we adopt PSNR for paired data and NIQE [1] for all data to assess perceptual quality, as reported in Tab. r4. Our method consistently achieves superior PSNR across various tasks. However, the non-reference metric (NIQE) exhibits patterns that diverge from the objective PSNR trend, where our performance is at a moderate level. This observation is consistent with prior studies [2,3], which highlight the inherent instability of NIQE. NIQE is highly sensitive to content variations and often fails to align with human perception, particularly under complex degradations. In conclusion, our method still remains highly competitive, as evidenced by both quantitative metrics and visual results (Appendix E, Fig. A6).
>
>
> |Method|Deraining||Enhancement||Desnowing||Dehazing||Deblurring||Average||
> |-|-|-|-|-|-|-|-|-|-|-|-|-|
> ||PSNR|NIQE|PSNR|NIQE|PSNR|NIQE|PSNR|NIQE|PSNR|NIQE|PSNR|NIQE|
> |Restomer|21.19|47.45|-|13.66|-|14.13|10.46|12.30|20.19|11.00|20.60|30.34|
> |AirNet|19.57|50.49|-|13.63|-|14.34|10.21|12.20|13.55|14.13|17.67|32.54|
> |Prompt-IR|21.38|51.34|-|12.98|-|14.29|10.46|13.06|24.83|12.56|21.98|32.66|
> |ProRes|16.26|45.57|-|13.12|-|15.99|10.54|13.08|21.38|12.19|17.47|30.12|
> |IDR|21.06|49.09|-|13.09|-|13.82|10.28|13.02|22.28|11.81|21.06|31.25|
> |AutoDIR|19.60|54.68|-|13.02|-|14.47|10.41|15.38|25.43|12.71|20.89|34.48|
> |DA-CLIP|20.93|51.90|-|12.76|-|14.45|10.40|14.11|22.25|12.52|20.96|32.99|
> |DiffUIR|21.38|51.85|-|12.57|-|14.34|8.57|12.79|24.77|12.21|21.91|32.85|
> |Ours|21.49|53.23|-|12.63|-|14.28|10.61|13.25|25.67|12.34|22.29|33.57|
>
> *Table r4: Evaluation on Real-world image restoration tasks*
>
>
> [1] Mittal A, Soundararajan R, Bovik A C. Making a “completely blind” image quality analyzer[J]. IEEE Signal processing letters, 2012.
>
> [2] Liu Y, Zhao H, Gu J, et al. Evaluating the generalization ability of super-resolution networks[J]. TPAMI, 2023.
>
> [3] Zhang L, Zhang L, Bovik A C. A feature-enriched completely blind image quality evaluator[J]. TIP, 2015.
>
>
> ### Q2: Disscusion about solver order
>
> The performance and computational cost of different solvers are summarized in Tab. r5. Notably, UPS increases computational cost by ~2× due to additional backward propagation[1]. In our configuration, the solver with $ k=2 $ and UPS yields the best performance and is thus used as the default setting in the paper. The case of $k=3$ demonstrates the solver’s scalability, although performance reaches a saturation point within our framework. Nevertheless, when generalized to related frameworks such as DiffUIR, both $k=3$ and UPS bring additional improvements, as shown in Tab. r2 in Reviewer (9bRU-W3). In terms of efficiency, due to the queued sampling strategy yielding neural function evaluations (NFEs) of $n+k−1$ (as proved in W2&Q3), the computational difference between solvers of different orders is determined solely by the order gap. Consequently, FLOPs increase approximately linearly with $k$. In conclusion, the solver complexity grows linearly with order, while performance varies depending on the target framework.
>
> |Order|UPS|Deraining||Enhancement||Desnowing||Dehazing||Deblurring||Average||NFEs|Forward FLops(G) per NFE|Backward FLOPs (G) per NFE|TotalFlops(G)|
> |-|-|-|-|-|-|-|-|-|-|-|-|-|-|-|-|-|-|
> |||PSNR|SSIM|PSNR|SSIM|PSNR|SSIM|PSNR|SSIM|PSNR|SSIM|PSNR|SSIM|||||
> |k=1|×|30.69|0.886|23.04|0.795|32.06|0.952|31.47|0.947|29.33|0.887|30.70|0.913|8|32.93|-|263.50||
> |k=2|×|31.31|0.894|23.08|0.798|32.51|0.955|31.62|0.948|30.07|0.897|31.20|0.919|9|32.93|-|296.44||
> |k=3|×|31.33|0.894|23.13|0.799|32.52|0.955|31.71|0.948|30.09|0.898|31.23|0.919|10|32.93|-|329.37||
> |k=1|√|31.10|0.892|23.07|0.798|32.32|0.954|31.54|0.947|29.80|0.895|31.01|0.918|8|32.93|≈65.86|≈790.32||
> |k=2|√|31.46|0.896|23.84|0.801|32.69|0.955|31.68|0.946|30.15|0.899|31.36|0.920|9|32.93|≈65.86|≈889.11||
> |k=3|√|31.46|0.896|23.84|0.801|32.69|0.955|31.68|0.946|30.15|0.899|31.36|0.920|10|32.93|≈65.86|≈987.91||
>
> *Table r5: FLOPs vs. PSNR for differnent $k$ with **queue-based sampling** strategy (Steps=8)*
>
>
> [1] Hobbhahn M, Sevilla J. What’s the backward-forward flop ratio for neural networks?[J]. Published online at epochai. org, 2021.

---

> > ### Comment · Reviewer_1zc3 · 2025-08-02
> >
> > Thank you for your reply, but  about q1 I still have two concerns:
> > 1. The authors' motivation to leverage the strong dependency between the degraded prior and the target image in restoration tasks is well-founded. However, the central claims of novelty appear to overlook significant and highly relevant prior art, most notably **ResShift** (Yue et al., 2023).
> >
> > 2.  The contribution of a "high-order solver" is also weakly positioned. This is an active research area. For instance, **DoSSR** (Cui et al., 2024) also proposes a "domain shift" approach with custom solvers to enhance efficiency. Furthermore, dedicated high-order solvers like **DPM-Solver** and **GENIE** are well-established methods for accelerating diffusion models.
> >
> > To be considered, the paper must rigorously differentiate its contributions from these foundational works (ResShift, DoSSR) and the broader field of high-order diffusion solvers. If you can address these two concerns of mine, I will consider raising the score.

---

> > > ### Author Response · Authors · 2025-08-03
> > >
> > > We appreciate your thoughtful feedback and would like to address the two points raised regarding prior art and the novelty of our proposed "high-order solver."
> > >
> > > **Regarding prior art, particularly ResShift (Yue et al., 2023) and DoSSR (Cui et al., 2024):** In image restoration, low- and high-quality images distributions serve as critical priors, which may be inevitably incorporated into diffusion process. Broadly speaking, these diffusion processes are governed by a probability distribution of the form:
> > >
> > >   **$p(I_t\vert I_0,I_{in}) \sim N(w^1_t I_0 + w^2_t I_{in}, (w^3_t)^2 \boldsymbol{I}) = w^1_t I_0 + w^2_t I_{in} + w^3_t \epsilon$**
> > >
> > > In Tab. d1, we summarize the forward processes of ResShift, DoSSR, and our DGSolver. ResShift modifies the standard diffusion process by using a Markov chain to shift the residual between LR and HR images. Building upon this, DoSSR retains the default diffusion settings (e.g., noise schedule) akin to DDPM(Ho et al., 2020) to fully utilize the pre-trained diffusion prior, and integrates DPMSolver framework to obtain high-order solvers. However, the key distinctions between our work and related methods are:
> > >
> > > (1) **General diffusion framework for image restoration:** In the forward process, we decouple the coefficients of each term (i.e., $w_t^i,i=1,2,3$) into independent variables, enabling the application of distinct noise schedules to each. This design enhances the flexibility and generalizability of our forward process. By employing different noise schedule settings, our framework can yield different solvers, whereas DoSSR is limited to specific schedules and can be regarded as a special case of DGSolver. Furthermore, we introduce a queue-based sampling strategy, which acts as a plug-and-play strategy to enhance solvers efficiency, potentially benefiting the performance of DoSSR and other solvers.
> > >
> > > (2) **Reuse the solver component for Universal Posterior Sampling (UPS):** Our approach not only integrates residual component into the forward process, but also reuses it during the reverse inference process to implement UPS. This ensures the stability and robustness of the solver, marking a key contribution that differentiates our work from previous methods.
> > >
> > > In summary, though the diffusion components are similar to those in ResShift and DoSSR, and may be inevitably applied in future work, our SDE solver stands out by integrating the residual component into both the forward and reverse SDEs, and embedding it into the reverse process for UPS with a queue-based sampling strategy, which are generalizable and efficient. Critically, our framework subsumes these methods as special cases under different noise schedule configurations.
> > >
> > >
> > > **Regarding the high-order solver and its position in the broader research field (e.g., DPMSolver (Lu et al., 2022) , GENIE (Dockhorn et al., 2022)):** We agree that solvers explored in DPM-Solver and GENIE are key advances in the field. Both our work and these methods fundamentally improve solver accuracy using Taylor expansions at different orders. Apart from that, we incorporate further innovations:
> > >
> > > (1) **Queue-based accelerated sampling:** DPMSolver employs a naive sampling strategy, while GENIE accelerates sampling using Gradient Distillation, which incurs additional computational overhead. In contrast, we introduce a queue-based sampling strategy, serving as a plug-and-play option to enhance efficiency for different solvers.
> > >
> > > (2) **Utilize solver component for Universal Posterior Sampling (UPS):** In addition to applying Taylor expansions to the solver component for more accurate approximations, residual component of solvers is further explored and utilized for UPS. This enables us to enhance the quality of restored images by reinforcing the coupling between prior and target data distributions, which is particularly critical for restoration tasks.
> > >
> > > In summary, though these solvers share a similar core principle, our method stands out by highly efficient sampling strategies and the full exploitation of solver components to enhance both efficiency and performance.
> > >
> > > We sincerely appreciate your constructive feedback, which provides us with valuable insights and significantly enhances the quality and depth of our manuscript. These valuable insights enable us to make clearer comparisons between different methods, allowing us to better position our work within the existing literature. Besides, it helps present a comprehensive overview of the field to the readers. In the revision, we will properly cite and discuss these relevant works, and include a detailed comparative analysis in the appendix.
> > >
> > > |Method|$w^1_t$|$w^2_t$|$(w^3_t)^2$|
> > > |-|-|-|-|
> > > |DDPM(Ho et al. 2020)|$\alpha_t$|0|$1-\alpha_t^2$|
> > > |Resshift|$1-\eta_t$| $\eta_t$|$\kappa\eta_t^2$|
> > > |DoSSR|$\alpha_t (1-\eta_t)$|$\alpha_t \eta_t$|$1-\alpha_t^2$|
> > > |Ours|$1-\alpha_t^*$|$(\alpha_t^* -\delta_t^* )$|$(\beta_t^*)^2$|
> > >
> > > *Table d1: Formulation of different diffusion process for image restoration. *

---

> ### Comment · Reviewer_1zc3 · 2025-08-06
>
> Thank you for the authors' substantial efforts in the rebuttal. It has addressed my concerns, and I will be raising my score from 3 to 4.

---

> > ### Author Response · Authors · 2025-08-07
> >
> > Thank you very much for your insightful and constructive comments throughout the review process. Your feedback has significantly improved the quality of our manuscript, and we deeply appreciate the time and expertise you dedicated to this work.

---

### Official Review · Reviewer_9bRU · 2025-07-03

**Clarity:** 3
**Significance:** 3
**Originality:** 3
**Rating:** 5
**Confidence:** 3

**Summary:**

Overall, this work is a strong paper with both theoretical depth and clear empirical improvements over previous universal methods.
The authors reformulate the restoration process as a semi-linear ODE that can be solved analytically, and they derive a closed form integral solution for the ODE. This derivation uses an inverse transformation for the nonlinear parts, assuming those parts are monotonic and invertible functions. They expand these nonlinear integrals with Taylor series and retain terms up to a chosen order k, and obtain a family of k-th order update formulas. Authors show that earlier diffusion restoration methods can be considered as the first order case of their solver.
Overall, the proofs around proposition 1 are solid and novel in the context of diffusion restoration.

Next, authors provide a theoretical upper bound for the approximation error (Jensen's gap where the outer expectation over the posterior is replaced by inner expectations with respect to the data distribution). They show the error depends on the variance of the measurement noise and the accuracy of the model’s predictions, guaranteeing that the error is controlled (for linear inverse problems with Gaussian noise) in Theorem 1.

Authors' claims are well supported by the empirical results, showing strong SoTA performance among universal methods and on par performance compared to task specific methods.

**Questions:**

1. Equation (4) has a typo; $q_{0t}$ should be $q_{0}$?

2. Equation (21), is this $l_{2}$ norm $∥⋅∥^2 $?

3. Could you show a direct ablation against another method like DiffUIR using their own solver vs. DGSolver to isolate the solvers contribution?

**Ethical Concerns:**

["NO or VERY MINOR ethics concerns only"]

**Final Justification:**

Authors have addressed my concerns and I continue to support acceptance of this work.

**Limitations:**

Theorem 1 provides a rigorous bound for the posterior approximation error only for linear measurement models, although many degradations ops are inherently nonlinear and hence not covered.

**Quality:**

3

**Strengths And Weaknesses:**

Strengths
1. The paper proposes the novel idea to combine exact ODE solution + universal posterior sampling, generalizing prior diffusion restoration methods.
2. Theoretical derivations are clear.
3. Empirically, they show strong SOTA results across tasks and datasets.


Weaknesses:
1. Authors don't provide FPS/runtime throughput data on full-resolution images. Also, they don't provide data on actual inference time or memory usage.

---

> ### Author Rebuttal · Authors · 2025-07-30
>
> ### W1: Efficiency comparisons among universal methods
>
> We appreciate the reviewer’s concern regarding inference efficiency. For fairness, we collect and mix the datasets used by comparison methods, with image resolution ranging from 256 to 1024 pixels. Accordingly, we evaluate model efficiency under three representative resolution settings, as summarized in Tab. r1. Let $k$ denote the solver order and $n$ be sampling steps. Obviously, our method (n = 3) and baseline DiffUIR remain competitive in computational cost and efficiency. When increasing $n$, memory consumption remains stable while time cost grows proportionally. Activating UPS that requires gradients backpropagation introduces per-step computational overhead and additional memory usage. To alleviate these issues, we employ a queue-based sampling strategy that significantly improves efficiency by reducing the computational complexity. Specifically, we use the number of neural function evaluations (NFEs) to evaluate the complexity. For a $k$-order solver, naive sampling from time $s$ to $t$ requires interpolating $k-1$ intermediate points within the interval, resulting in $k$ NFEs per step. Consequently, the overall computational complexity for $n$ steps is $O(nk)$. In contrast, the queue-based sampling strategy precomputes the values at $k−1$ time points and caches them for reuse in subsequent steps, offering high efficiency by reducing total complexity to $O(k+n−1)$. Tab. r1 also demonstrates that the queue-based solver achieves approximately a 2× efficiency improvement over naive solvers when $k=2$, and around a 3× improvement when $k=3$.
>
>
> |Size||256x256|||512x512|||1024x1024||
> |-|-|-|-|-|-|-|-|-|-|
> |Method|Mem.(G)|Time(s)|FPS|Mem.(G)|Time(s)|FPS|Mem.(G)|Time(s)|FPS|
> |Restomer|1.959|0.105|9.563|6.670|0.381|2.622|25.419|1.773|0.564|
> |AirNet|1.039|0.194|5.159|3.480|0.738|1.355|11.244|20.499|0.049|
> |PromptIR|2.544|0.111|8.981|7.255|0.399|2.508|26.005|1.845|0.542|
> |ProRes|2.027|0.318|3.149|2.514|0.766|1.305|6.025|1.715|0.583|
> |IDR|1.340|0.052|19.253|4.313|0.136|7.373|16.110|0.615|1.626|
> |AutoDIR|7.023|6.266|0.160|11.021|11.986|0.083|-|-|-|
> |DA-CLIP|2.119|2.585|0.387|6.775|7.937|0.126|58.548|60.893|0.016|
> |DiffUIR(n=3)|1.563|0.118|8.450|3.528|0.206|4.862|18.060|0.911|1.098|
> |Ours-L(n=3)|1.561|0.112|8.908|3.528|0.199|5.014|18.059|0.907|1.103|
> |Naive(n=8)|k=1|×UPS|
> |Ours-T|0.777|0.277|3.605|2.291|0.385|2.594|15.306|1.705|0.587|
> |Ours-S|0.787|0.290|3.450|2.300|0.401|2.494|15.316|1.755|0.570|
> |Ours-B|0.942|0.291|3.431|2.907|0.492|2.033|17.438|2.280|0.439|
> |Ours-L|1.562|0.293|3.407|3.527|0.529|1.889|18.058|2.402|0.416|
> |Naive(n=8)|k=1|√UPS|
> |Ours-T|1.764|0.500|1.998|5.793|0.829|1.206|32.581|4.454|0.225|
> |Ours-S|1.905|0.502|1.993|6.265|0.865|1.156|34.237|4.545|0.220|
> |Ours-B|2.762|0.520|1.923|9.645|1.078|0.928|41.357|5.770|0.173|
> |Ours-L|3.613|0.535|1.868|10.593|1.144|0.874|43.815|6.022|0.166|
> |Naive(n=8)|k=2|×UPS|
> |Ours-T|0.784|0.535|1.869|2.297|0.717|1.395|15.313|3.194|0.313|
> |Ours-S|0.794|0.548|1.824|2.308|0.747|1.339|15.323|3.296|0.303|
> |Ours-B|0.948|0.555|1.802|2.913|0.924|1.082|17.444|4.279|0.234|
> |Ours-L|1.563|0.571|1.751|3.527|0.989|1.011|18.059|4.522|0.221|
> |Naive(n=8)|k=2|√UPS|
> |Ours-T|1.770|0.974|1.027|5.800|1.559|0.641|32.587|8.150|0.123|
> |Ours-S|1.933|0.987|1.013|7.508|1.871|0.535|33.586|10.482|0.095|
> |Ours-B|2.803|0.990|1.010|9.688|2.089|0.479|41.399|11.276|0.089|
> |Ours-L|3.829|1.019|0.981|10.679|2.219|0.451|44.154|11.723|0.085|
> |Queue(n=8)|k=2|×UPS|
> |Ours-T|0.780|0.303|3.301|2.294|0.431|2.321|15.310|1.905|0.524|
> |Ours-S|0.791|0.321|3.115|2.304|0.460|2.174|15.320|1.980|0.505|
> |Ours-B|0.946|0.323|3.096|2.911|0.547|1.823|17.442|2.573|0.388|
> |Ours-L|1.563|0.324|3.086|3.527|0.598|1.673|18.059|2.694|0.371|
> |Queue(n=8)|k=2|√UPS|
> |Ours-T|1.771|0.557|1.795|5.800|0.925|1.081|32.587|5.012|0.200|
> |Ours-S|1.920|0.561|1.782|6.279|0.961|1.041|34.250|5.101|0.196|
> |Ours-B|2.892|0.590|1.695|9.677|1.201|0.833|41.388|6.472|0.155|
> |Ours-L|3.816|0.600|1.667|10.601|1.273|0.786|44.074|6.741|0.148|
> |Naive(n=8)|k=3|×UPS|
> |Ours-T|0.790|0.759|1.317|2.303|1.053|0.950|15.319|4.687|0.213|
> |Ours-S|0.799|0.775|1.290|2.312|1.099|0.910|15.328|4.836|0.207|
> |Ours-B|0.954|0.780|1.283|2.919|1.347|0.742|17.450|6.253|0.160|
> |Ours-L|1.562|0.801|1.248|3.527|1.446|0.692|18.058|6.586|0.152|
> |Naive(n=8)|k=3|√UPS|
> |Ours-T|1.782|1.414|0.707|5.812|2.338|0.428|32.599|12.384|0.081|
> |Ours-S|1.958|1.428|0.700|6.318|2.474|0.404|34.288|13.238|0.076|
> |Ours-B|2.909|1.433|0.698|9.694|3.100|0.323|41.405|16.755|0.060|
> |Ours-L|4.048|1.496|0.668|11.050|3.302|0.303|48.273|17.461|0.057|
> |Queue(n=8)|k=3|×UPS|
> |Ours-T|0.777|0.338|2.959|2.290|0.481|2.077|15.306|2.132|0.469|
> |Ours-S|0.787|0.364|2.744|2.301|0.502|1.991|15.316|2.195|0.455|
> |Ours-B|0.942|0.366|2.730|2.907|0.615|1.626|17.438|2.845|0.351|
> |Ours-L|1.562|0.369|2.710|3.527|0.657|1.522|18.058|2.999|0.333|
> |Queue(n=8)|k=3|√UPS|
> |Ours-T|1.765|0.624|1.602|5.794|1.060|0.943|32.581|5.664|0.177|
> |Ours-S|1.907|0.646|1.547|6.267|1.090|0.917|34.238|5.789|0.173|
> |Ours-B|2.765|0.661|1.512|9.648|1.369|0.730|41.359|7.339|0.136|
> |Ours-L|3.613|0.695|1.440|10.593|1.470|0.680|43.814|7.656|0.131|
>
> *Table r1: Efficiency comparisons among universal methods. '-' means out of memmory.*
>
> ### Q1&Q2: Definitions about $q_{0t}$ and norm $\|\cdot\|$
>
> We consider $q_0$ as the data distribution of $I_0$. In Eq. (4), $q_{0t}:=q(I_t\vert I_0,I_{res},I_{in})$ denotes a conditional probability distribution of $I_t$ conditioned on $I_{0},I_{res},I_{in}$ for simplicity. In Eq. (21), $\|\cdot\|$ is $l_2$-norm. We appreciate your comments and promise to refine the notations to avoid ambiguity in the revision.
>
> ### Q3: Isolated contribution of our Solver
>
> We appreciate your insightful suggestion. Our solver is built upon the prediction of residuals and noise, and is thus applicable to any diffusion-based models that adopt similar formulations. To validate its generality, we adapt our solver to DiffUIR. As shown in Tab. r2, DiffUIR consistently benefits from our solver across various configurations, leading to notable performance improvements.
>
> |DiffUIR|n=3|Deraining||Enhancement||Desnowing||Dehazing||Deblurring||Average||
> |-|-|-|-|-|-|-|-|-|-|-|-|-|-|
> |Order|UPS|PSNR|SSIM|PSNR|SSIM|PSNR|SSIM|PSNR|SSIM|PSNR|SSIM|PSNR|SSIM|
> |k=1|x|30.67|0.887|21.21|0.769|30.70|0.943|30.29|0.944|29.00|0.877|29.93|0.907|
> |k=2|x|30.76|0.888|21.28|0.771|30.82|0.946|30.58|0.949|29.04|0.879|30.05|0.910|
> |k=3|x|30.81|0.890|21.30|0.772|30.85|0.948|30.61|0.951|29.05|0.880|30.08|0.911|
> |k=1|√|30.72|0.887|21.24|0.770|30.81|0.945|30.43|0.946|29.02|0.878|30.01|0.908|
> |k=2|√|31.17|0.899|21.52|0.786|31.01|0.951|30.76|0.954|29.34|0.885|30.32|0.917|
> |k=3|√|31.21|0.901|21.54|0.787|31.08|0.953|30.80|0.955|29.35|0.886|30.37|0.918|
>
> *Table r2: Adapt our solvers to DiffUIR*

---

> > ### Comment · Reviewer_9bRU · 2025-08-06
> >
> > I appreciate authors' efforts in addressing the comments. This is a strong work and I continue to support its acceptance.

---

> > > ### Author Response · Authors · 2025-08-07
> > >
> > > Thank you for your supportive feedback and acceptance recommendation. We truly appreciate your time and valuable insights. Best wishes.

---

### Decision · Program_Chairs · 2025-09-17

**Decision:**

Accept (poster)

**Comment:**

The paper improves diffusion-based image restoration by proposing a training-free solver for diffusion models, treating them as ODE. They demonstrate enhanced restoration quality and efficiency across a wide range of degradations and benchmarks and compared to several existing baselines.

Five expert reviewers refereed the work and found the approach sound, well-motivated, and the empirical evidence generally convincing. On the other hand, the main concerns were regarding ablation studies, some similar prior work, experimental setup, clarity, and computational complexity.

The authors provided a thorough rebuttal which addressed all the main concerns and led to all reviewers eventually suggesting acceptance.

The AC does not see any major flaw with the paper and on the other hand finds the contribution novel with significant empirical evidence. Therefore, the AC suggests acceptance.